# A long-term estimation of biogenic volatile organic compound (BVOC) emission in China during 2001-2016: the roles of land cover change and climate variability

Hui Wang[1, 2], Qizhong Wu[1], Alex B. Guenther[2], Xiaochun Yang[1], Lanning Wang[1], Tang Xiao[3], Jie Li[3], Jinming Feng[4], Qi Xu[1], Huaqiong Cheng[1]

[1]College of Global Change and Earth System Science, Beijing Normal University, Beijing 100875, China
[2]Department of Earth System Science, University of California, Irvine, CA 92697, USA
[3]State Key Laboratory of Atmospheric Boundary Layer Physics and Atmospheric Chemistry, Institute of Atmospheric Physics, Chinese Academy of Sciences, Beijing 100029, China
[4]Key Laboratory of Regional Climate-Environment for Temperate East Asia, Institute of Atmospheric Physics, Chinese Academy of Sciences, Beijing 100029, China
*Correspondence to*: Qizhong Wu (wqizhong@bnu.edu.cn) & Lanning Wang (wangln@bnu.edu.cn)

**Abstract.** Satellite observations reveal that China has been leading the global greening trend in the past two decades. We assessed the impact of land cover change as well as climate variability on total BVOC emission in China during 2001-2016. We found the greening trend in China is leading a national scale increase of BVOC emission. The BVOC emission level in 2016 can be 11.7% higher than that in 2001 because of higher tree cover fraction and vegetation biomass. In the regional scale, and the BVOC emission level during 2013-2016 could be 8.6%~19.3% higher than that during 2001-2004 in the hotspots including 1) northeastern China, 2) Beijing and its surrounding areas, 3) the Qinling Mountains, 4) Yunnan province, 5) Guangxi-Guangdong provinces and 6) Hainan island because of the land cover change without considering the impact of climate variability. The comparison among different scenarios showed that vegetation changes resulting from land cover management is the main driver of BVOC emission change in China. Climate variability contributed significantly to interannual variations but not much to the changing trend during the study period. In the standard scenario, that considers both land cover change and climate variability, a statistic significant increasing trend still can be found in the regions including Beijing and its surroundings, Yunnan provinces and Hainan island, and BVOC emission total amount in these regions during 2013-2016 is 11.0%-17.2% higher that during 2001-2004. We compared the long-term HCHO vertical columns (VC) from the satellite-based Ozone Monitoring Instrument (OMI) with the estimation of isoprene emission in summer. The results showed statistically significant positive correlation coefficients over the regions with high vegetation cover fractions. In addition, the isoprene emission and HCHO VC both showed statistically significant increasing trends in the south of China where these two variables have high positive correlation coefficients. This result may support our estimation of the variability and trends of BVOC emission in this region, however, the comparison still has large uncertainties since the absence of chemical and physical processes. Our results suggest that the continued increase of BVOC will enhance the importance of considering BVOC when making policies for controlling ozone pollution in China along with ongoing efforts to increase the forest cover fraction.

## 1 Introduction

Biogenic Volatile Organic Compounds (BVOCs) play an important role for air quality and the climate system due to their large emission amount and reactivity (Guenther et al., 1995; Guenther, 2006). BVOCs are important precursors of ozone and secondary organic aerosols (SOAs) (Kavouras et al., 1998; Claeys et al., 2004), therefore, it is important to understand the variability of BVOC emission and its impact on air quality

and the climate system. The emission of BVOC is controlled by multiple environmental factors like temperature, radiation, $CO_2$ concentration and other stresses, therefore it is affected by climate changes (Guenther et al., 1995; Arneth et al., 2007; Penuelas and Staudt, 2010). Besides the climatic factors, the land cover change also plays a key role in the variability of BVOC emission (Stavrakou et al., 2014; Unger, 2013; Chen et al., 2018). For instance, the global cropland expansion has been estimated to dominate the reduction of isoprene, the dominant BVOC species, in last century (Lathière et al., 2010; Unger, 2013) although there are large uncertainties associated with these estimates.

China has been greening in recent decades (Piao et al., 2015). A recent study points out that China accounts for 25% of the net increase of global leaf area during 2000-2017 (Chen et al., 2019). The increase of forest area plays a dominant role in greening in China with multiple programs to maintain and expand forests (Zhang et al., 2016; Bryan et al., 2018; Chen et al., 2019). The enhancement of vegetation cover rate and biomass can lead to the increase of BVOC emission and induce changes on local air quality and the climate system. Previous studies have investigated the long-term emission trend of dominant BVOC species like isoprene in China (Fu and Liao, 2012; Li and Xie, 2014; Stavrakou et al., 2014; Chen et al., 2019). Li and Xie (2014) estimated the historical BVOC emissions during 1981-2003 in China using the national forest inventory records and reported that the BVOC emission increased at a rate of 1.27% $y^{-1}$. Another estimation by Stavrakou et al. (2014) showed an upward trend of 0.42% $y^{-1}$ of isoprene emission in China during 1979-2005 driven by the increasing temperature and solar radiation, moreover, the upward trend of isoprene emission reached 0.7% $y^{-1}$ when considering the replacement of cropland with forest. A recent study by Chen et al. (2018) concluded that the global isoprene emission decreased by 1.5% because of the tree cover change during 2000-2015, but in China, the isoprene emitted by broadleaf trees and non-trees increased by 3.6% and 5.4%, respectively. However, these studies have limitations in representing annual changes of vegetation, e.g., Li and Xie (2014) used fixed LAI input of year 2003 over the whole study period of 1981-2003.

Considering the significant land cover change and greening trend in China, it is necessary to thoroughly investigate the impact of intense reforestation on BVOC emission in China. In this study, we used the latest annually continuous land cover products Version 6 by the MODerate-resolution Imaging Spectroradiometer (MODIS) sensors as well as the Model of Emissions of Gases and Aerosols from Nature (MEGAN, Guenther et al. 2012) model to investigate BVOC emission in China from 2001 to 2016. By annually updating the vegetation information of MODIS observations, we could accurately estimate interannual variability of BVOC emission to assess the impact of greening trend on BVOC in China during 2001-2016.

A long-term in-situ observation of BVOC is not available in China currently to investigate interannual variability of BVOC emission, however, satellite formaldehyde (HCHO) observations provide an opportunity to validate the interannual variability of isoprene, the dominant compound among BVOC species that

accounts for almost half of total BVOC emission in China (Li et al., 2013). Since HCHO is an important proxy of isoprene in forest regions with no significant anthropogenic impact, satellite HCHO columns are widely used to derive isoprene emission at regional to global scales (Palmer et al., 2003; Marais et al., 2012; Stavrakou et al. 2009; Stavrakou et al., 2015; Kaiser et al. 2018). Zhu et al. (2017b) reported an increasing trend of HCHO vertical columns (VC) detected by the Ozone Monitoring Instrument (OMI) driven by increasing cover rate of local forest in the northwestern United States. Stavrakou et al. (2018) also used the long-term HCHO VC to investigate the annual variability of BVOC induced by climate variability. Here we used the long-term OMI 2005-2016 record to evaluate the interannual isoprene variability we estimated in China.

## 2 Data and Method

### 2.1 MEGAN Model

MEGAN (Guenther et al., 2006; Guenther et al., 2012) is the most widely used model for calculating BVOC emission from regional to global scales (Müller et al., 2008; Li et al., 2013; Sindelarova et al., 2014; Chen et al., 2018; Bauwens et al. 2018; Messina et al. 2016). The offline version of the MEGAN v2.1 (Guenther et al., 2012) model, available at https://bai.ess.uci.edu/megan, was used to estimate the BVOC emission in China from 2001 to 2016. MEGAN v2.1 calculates emissions for 19 major compound categories uses the fundamental algorithm:

$$F_i = \varepsilon_i \gamma_i \qquad (1)$$

where Fi, εi and γi represent the emission amount, the standard emissions factor, and emission activity factor of chemical species i. The standard emission factor in this study is based on the plant functional type (PFT) distribution from the Community Land Model 4.0 (Lawrence et al., 2011). The emission activity factor γi accounts for the impact of multiple environmental factors and can be written as:

$$\gamma_i = C_{CE} LAI \gamma_{p,i} \gamma_{T,i} \gamma_{A,i} \gamma_{SM,i} \gamma_{C,i} \qquad (2)$$

where $\gamma_{p,i}$, $\gamma_{T,i}$ $\gamma_{A,i}$ $\gamma_{SM,i}$ and $\gamma_{C,i}$ represent the activity factors for light, temperature, leaf age, soil moisture and $CO_2$ inhibition impact. The Cce (=0.57) is a factor to set the γi equal to 1 at standard conditions (Guenther et al., 2006). The LAI is the leaf area index, and it is used to define the amount of foliage and the leaf age response function as described in Guenther et al. (2012). The light and temperature response algorithms in MEGAN v2.1 are from Guenther et al. (1991, 1993, 2012), which described enzymatic activities controlled by temperature and light conditions. The $CO_2$ inhibition algorithm is from Heald et al. (2009), and only the

estimation of isoprene emission considers the impacts of soil moisture and $CO_2$ concentration. The detailed descriptions of these algorithms can be found in Guenther et al. (2012) and Sakulyanontvittaya et al. (2008).

**2.2 Land Cover Datasets.**

The land cover parameters for driving MEGAN including LAI, PFT and vegetation cover fraction (VCF) were provided by satellite datasets. The MODIS MOD15A2H for 2001 (https://lpdaac.usgs.gov/products/mod15a2hv006/, Myneni et al., 2015a) and MCD15A2H for 2002-2016 LAI (https://lpdaac.usgs.gov/products/mcd15a2hv006/, Myneni et al., 2015b) datasets were used in this study. The parameter LAIv in MEGAN is calculated as:

$$LAIv = \frac{LAI}{VCF} \tag{3}$$

where VCF is provided by MODIS MOD44B datasets (https://lpdaac.usgs.gov/products/mod44bv006/, Dimiceli et al., 2015).

The PFT was used to determine the canopy structure and standard emission factors in MEGAN (Guenther et al., 2012). We adopted the default emission factors for PFTs described in Guenther et al. (2012), which have been presented in the Table S3 in the supplement. The PFT dataset in this study is obtained from the MODIS MCD12C1 land cover product (https://lpdaac.usgs.gov/products/mcd12c1v006/, Friedl and Sulla-Menashe, 2015). MODIS IGBP classification were mapped to the PFT classification of MEGAN or the Community Land Model (CLM) (Lawrence et al., 2011) based on the description of the legends in the user guide (Sulla-Menashe and Friedl, 2018) and the climatic criteria described in Bonan et al. (2002). The spatial distribution of percentage of PFTs in model grids is presented in Figure 1. According to the description of the legends, we firstly mapped the IGBP classification to eight main vegetation categories: 1) needleleaf evergreen forests, 2) broadleaf evergreen forests, 3) needleleaf deciduous forests, 4) broadleaf deciduous forests, 5) mixed forests, 6) shrub, 7) grass and 8) crop. The mapping method is described in Table S1 in the supplement. Eight main categories then were mapped to the classification of MEGAN/CLM for boreal, temperate and boreal climatic zones using the definition in Bonan et al. (2002). Table S2 in the supplement presents the climatic criteria for mapping, and the climatic information for mapping was from the ERA Interim climatology (https://www.ecmwf.int/en/forecasts/datasets/reanalysis-datasets/era-interim, Berrisford et al., 2011) Reanalysis dataset over 2001-2016.

**2.3 Meteorological Datasets**

The hourly meteorological fields including temperature, downward shortwave radiation (DSW), wind speed, surface pressure, precipitation and water vapor mixing ratio were provided by the Weather Research and Forecast (WRF) Model V3.9 (Skamarock et al., 2008) simulations. The model was driven by ERA-Interim reanalysis data (Berrisford et al., 2011) with 27 km horizontal spatial resolution and 39 vertical layers. The physical schemes were presented in supplemental Table S4.

Since light and temperature conditions are the main environmental drivers of BVOC emission (Guenther et al., 1993; Sakulyanontvittaya et al., 2008), we assessed the reliability of the WRF simulated DSW and 2-meter temperature (T2) using in-situ observations from 98 radiation observation sites and 697 meteorology observation sites in China. The in-situ observations are from the National Meteorological Information Center (http://data.cma.cn/). We converted the hourly model outputs and daily observations to monthly averaged values from 2001 to 2016 for comparison. For DSW, the average mean bias (MB), mean error (ME) and root mean square error (RMSE) are 40.37 (± 20.81), 43.55 (± 17.52) and 49.79 (± 17.70) W m$^{-2}$ for 98 studied sites. The overestimation of DSW simulation is a common issue in multiple simulation studies and may be induced by the lack of physical processes for aerosol radiation effect (Wang et al., 2011; Situ et al., 2013; Wang et al., 2018) and misrepresentation of the radiative effect of the sub-grid scale cumulus cloud (Ruiz-Arias et al., 2016). For T2, the average MB, ME and RMSE are -1.19 (± 2.87), 2.40 (± 2.14) and 2.65 (± 2.11) °C among 697 sites over China. The regional scale validation was also conducted in five main regions (Figure S1) by comparing the averaged values of the observation and the simulation among the sites in the regions we defined, and the results (Figure S2 and Figure S3) and statistical parameters (Table S5 and Table S6) can be found in the supplement.

We also compared the monthly anomalies of DSW and T2 from the model simulation and observation to validate the interannual variability of meteorological fields simulated by WRF. As shown in Figure 2, the results indicate that the model accurately reproduced the interannual variability of DSW and T2, and the correlation coefficients of DSW and T2 anomaly between the simulation and observation reached 0.77 and 0.88, respectively. The trends of growing season averaged T2 and DSW from model results as well as in-situ measurements are presented in Figure 3. The model and the in-situ measurements show similar patterns of T2. For instance, the model and observations both show an increasing trend in regions like the Tibetan Plateau and southern China as well as a decreasing trend in eastern and northeastern China. For DSW, the model presented a dimming trend in northeastern and eastern China and a brightening trend in southeastern and central China, and the limited number of radiation observation sites show a similar pattern of trend with model results. In general, the WRF simulation successfully captured the long-term meteorological variabilities and is reasonable to use for estimating the impact of climatic variability on BVOC emission in China for this study.

### 2.4 Satellite Formaldehyde (HCHO) Observations

The satellite HCHO VC used in this study is from the Belgian Institute for Space Aeronomy (BIRA-IASB) and was retrieved using the differential optical absorption spectroscopy (DOAS) algorithm (De Smedt et al., 2012; De Smedt et al., 2015). We used the monthly Level-3 HCHO VC product with 0.25° × 0.25° spatial resolution, and the rows affected by the row anomaly since June 2007 have been filtered in this product (De

Smedt et al., 2015; Jin and Holloway, 2015). Since the OMI instrument is temporally stable (Dobber et al., 2008; De Smedt et al., 2015), the OMI HCHO VC product is suitable for long-term analysis (Jin and Holloway, 2015) and was used to primarily validate our estimation of isoprene emission variability. The major sources of tropospheric HCHO are biogenic VOC, anthropogenic VOC and open fires (Zhu et al., 2017a). Since biogenic isoprene is the dominant source of HCHO over forests in summertime (Palmer et al., 2003), we used HCHO as the proxy of isoprene to validate the interannual variability of isoprene estimates.

**2.5 Scenarios and Analysis Method**

We designed five scenarios (S1-S5) to investigate the impact of land cover change and climatic conditions on BVOC emission. The configurations of the five scenarios are shown in Table 1:

1) S1 was considered as the standard or full scenario;

2) S2 was used to investigate the impact of the ecosystem and land cover variability on BVOC emission;

3) S3 and S4 characterize the effect of climate variability and compare the difference of BVOC emission induced by vegetation change between 2001 and 2016;

4) S5 investigate the contribution of LAI trend to BVOC emission trend solely.

The climatic variability can affect the growth of vegetation and then affect LAI values (Piao et al., 2015). In this study, the interaction between climate and ecosystem is not considered in the offline MEGAN model, which means that the meteorological conditions, e.g. precipitation, will not affect the LAI values as well as phenology of vegetations.

The chemical species emissions estimated by MEGAN were grouped into four major categories including isoprene, monoterpene, sesquiterpene and other VOCs since the terpenoids account for the majority of total BVOC emission and have known impacts on atmospheric oxidants and SOA (Wang et al., 2011). The trend analysis in this study was done following the Theil-Sen trend estimation method and the results were tested by the Mann-Kendall non-parametric trend test (MK test) (Gilbert, 1987). The trend analysis and the MK tests in this study were implemented using the trend_manken (https://www.ncl.ucar.edu/Document/Functions/Built-in/trend_manken.shtml) function of the NCAR Command Language (NCL, https://www.ncl.ucar.edu/).

**3 Results and Discussion**

**3.1 The Variability of BVOC Emission in China During 2001-2016**

As shown in Table 2, the average annual emission during 2001-2016 of isoprene, monoterpene, sesquiterpene and other VOCs estimated from S1 are 15.94 (±1.12), 3.99 (±0.17), 0.50 (±0.03) and 13.84 (±0.78) Tg, respectively. Isoprene is the dominant species and accounts for about half of the total BVOC emission in

China. As shown in Figure 4, the estimated BVOC emission in S2 has a statistically significant increasing trend without considering the annual variability of meteorological conditions. The increasing rates of isoprene, monoterpenes, sesquiterpenes and total BVOC emission in S2 scenarios are 0.64, 0.44, 0.39 and 0.50 % $y^{-1}$, respectively. The S1 scenario considers the impact of annual meteorological variability as well as the surface vegetation change, and the BVOC emission in S1 is in an upward trend but did not pass the significance test of $p < 0.1$. There's no significant trend of BVOC emission for both S3 and S4, with fixed landcover and annually updated meteorological conditions, demonstrates that meteorology was not the direct driver of BVOC emission trend in China during this period. Climatic conditions could affect the BVOC emission indirectly by affecting the growth of vegetation and controlling BVOC emission (Peñuelas et al., 2009), which is not considered in the model used in this study. The estimated total BVOC emission in S5 also has a statistically significant increasing trend of 0.26% $y^{-1}$ ($p<0.05$) without considering the annual variability of meteorological conditions, which is purely caused by the increase of LAI during 2001-2016. The surface vegetation change had a significant influence on BVOC emissions in China during 2001-2016 according to our estimation. In S2, the interannual variability of total BVOC emission is primarily determined by the surface vegetation change resulting in a nearly linear increasing trend of BVOC emission. The average annual emission of total BVOC during 2009-2016 is 3.9% (1.29 Tg) higher than that during 2001-2008 in S2, and the average annual emissions of isoprene, monoterpene and sesquiterpene during the previous eight years are by 5.0% (0.75 Tg), 3.5% (0.13 Tg) and 3.1% (0.02 Tg) higher than those during next eight years, respectively. The comparison of S3 and S4 results further demonstrates the importance of vegetation development on BVOC emission considering the interannual variability of meteorological conditions. S3 and S4 adopted the same annually updated meteorological field but the fixed land cover information of the year 2001 and 2016, respectively. The fluctuation of meteorological factors leads to an interannual variability of BVOC emission in S3 and S4, but the increase of vegetation cover rate in 2016 results in BVOC emissions that are much higher than that in 2001. As presented in Table 2, the average total BVOC emissions are 31.77 (±1.54) and 35.48 (±1.76) Tg in S3 and S4, respectively, and the total BVOC emission in S4 is by 11.7% (3.71 Tg) higher than that in S3. The emissions of isoprene, monoterpene and sesquiterpene with the land cover information of the year 2016 are by 14.1% (2.07 Tg), 9.0% (0.34 Tg) and 8.5% (0.04 Tg) higher than those estimated based on the land cover information of the year 2001, respectively.

### 3.2 The Regional Variability of BVOC Emission in China

The hotspots of BVOC emission are mainly located in the northeast, central and south of China where the forest is widely distributed and the climate is warm and favorable for emitting BVOC as shown in Figure 5. The Changbai Mountains, the Qinling Mountains, the southeast and southwest China forest regions, southeast Tibet, Hainan and Taiwan islands are the regions with highest BVOC emission in 2001. The spatial patterns

of statistically significant (p < 0.1) changing trends in S1-S5 are also presented for individual categories in Figure 5. The spatial distributions of trends of different species in S2 all shows a nationwide significantly increasing trend since the vegetation development is the main driver of the increasing trend of BVOC emission (c, i, o and u in Figure 5). In the full scenario of S1, the area with statistically significant trend is less than that in S2 considering the impact of meteorological variability. S5 also shows a nationwide significantly increasing trend of BVOC emission but with smaller rates comparing to S2 (f, l, r and x in Figure 5). While a positive increasing trend induced by meteorology is also found in Tibet, western Sichuan and southeastern Yunnan province in S3 and S4, which is induced by the warming climate and stronger DSW as presented in Figure 3.

The spatial patterns of changing trends of total BVOC emission and landcover parameters are presented in Figure 6. The cover fraction of broadleaf trees shows a strong increasing trend in the regions including northeastern, central and southern China. Meanwhile, the grass and crop cover fractions show a decreasing trend in these regions. The crop cover rate also shows an increasing trend in northeastern China, Shan Xi, Gansu and Xinjiang Provinces by replacing the grass there. Besides the change of PFTs, a nationwide increasing trend of LAIv was also found for most regions in China.

In order to understand the regional discrepancies of changing trend of BVOC emission and its drivers, we chose six regions of interest to further analyze. As shown in (a) of Figure 6, the six regions includes 1) northeastern China (orange frame in Figure 6a, 45.5-54N, 118-130E), 2) Beijing and its surrounding areas (black frame in Figure 6a, 39-42.5N, 114-120E), 3) Qinling Mountains (red frame in Figure 6a, 30-34N, 105.5-112E), 4) Yunnan Province (blue frame in Figure 6a, 21-27N, 97.5-106E), 5) Guangxi-Guangdong provinces (purple frame in Figure 6a, 21-25N, 106-117E) and 6) Hainan island (green frame in Figure 6a, 17.5-20.5N, 108-112E). The annual changes of vegetation conditions (PFTs and LAIv), the annual emission flux and growing season averaged temperature and DSW are presented in Figure 7 and Figure 8, and the averaged values and trends of above variables are listed in Table 3 and Table 4. In general, six regions all show that the woody vegetations replaced the herbaceous vegetations with a significantly increasing trend of annual LAIv. Since the broadleaf trees tend to have a higher emission potential than grass or crop (Guenther et al., 2012), the transformation of land cover from grass or crop to broadleaf tree is expected to enhance the emission of BVOC by increasing the landscape average emission factor. As shown in Table 3 and Table 4, the broadleaf tree cover fraction increased at a rate of 0.15~0.32 % y$^{-1}$, and the grass cover fraction decreased at a rate of 0.11~0.37% y$^{-1}$ among the six regions during 2001-2016. Except for northeastern China we defined, other five regions all show a decreasing trend of 0.04~0.26% y$^{-1}$ for the crop cover fraction. As a result, the total tree cover fraction during the last four years (2013-2016) is 11.0, 82.5, 6.1, 5.7, 5.9 and 8.0 % higher than that during first four years (2001-2004) for northeastern China, Beijing and its surroundings,

Qinling Mountains, Yunnan Province, Guangxi-Guangdong provinces and Hainan Island, respectively, and the LAIv for these regions also increased by 14.8 ~ 26.4 %. Correspondingly, the annual BVOC emission flux in six regions all show a significantly increasing trend without considering the variability of meteorology in S2. The mean annual BVOC emission flux for the last four years (2013-2016) is 8.6%~9.8% higher than that for the first four years (2001-2004) in the regions defined above except for Beijing and its surrounding areas, where the change of the annual BVOC emission flux reached 19.3% with the tree cover fraction increased by 82.5%. If we only consider the contribution of LAI change, as described in the scenario S5, above sub-regions except for Guangxi-Guangdong provinces still show a statistically significant increasing trend of BVOC emission without considering the variability of meteorology, and the contributions of the LAIv change to BVOC emission increasing trend is about 25%-66% in these regions.

The changing trend of the annual BVOC emission flux is different in S1 when the impact of meteorological variability is taken into account. The simulated T2 and DSW during the growing season do not show a significant trend in most regions we chose. As shown in Figure 7 and Figure 8, the variabilities of the temperature and DSW during the growing season controlled the variability of BVOC flux in S1. When the meteorological variability is considered, there are still three regions we defined above that show a significantly increasing trend of BVOC emission: 1) Beijing and its surrounding areas, 2) Guangxi-Guangdong Provinces and 3) Hainan island. In Beijing and its soundings, the changing trend of the annual BVOC emission flux is 0.03 g m$^{-1}$ y$^{-1}$ in S1, and the mean annual BVOC emission flux in last four years still shows a large increase of 16.6% compared to that in first four years in this region. A significantly increasing trend of temperature of 0.03 °C y$^{-1}$ were found in southwestern China region, therefore, the increasing trend of the annual BVOC emission flux is 0.1 g m$^{-1}$ y$^{-1}$ in S1, which is higher than that in S2 of 0.04 g m$^{-1}$ y$^{-1}$. The BVOC flux in last four years is about 17.2% higher than that in first four years in southwestern China. In Hainan island, the changing trend of the annual BVOC emission flux is 0.12 g m$^{-1}$ y$^{-1}$ in S1, and the annual BVOC emission flux in last four years is 11.0% higher than that in first four years.

The estimated increase of BVOC in the regions like the Qinling Mountains and southern China are expected to affect regional air quality. For the Qinling Mountains and surrounding areas, as estimated by Li et al. (2018) using the WRF-chem model, the average contribution of BVOC to O$_3$ could reach 16.8 ppb for the daily peak concentration and 8.2 ppb for the 24h concentration in the urban region of Xi'an, one of the biggest cities near the Qinling Mountains suffering from poor air quality in recent years (Yang et al., 2019). For Guangxi-Guangdong Provinces, Situ et al. (2013) reported that BVOC emission could contribute an average 7.9 ppb surface peak O$_3$ concentration for the urban area in the Pearl River Delta region, and the contribution from BVOC even reached 24.8 ppb over PRD in November. Since BVOC plays an important role in local air quality, the change of BVOC emission may have an even greater effect on the local ozone pollution. For

instance, the simulation study by Li et al. (2018) also found that the urban region of Xi'an is VOC-limited because of the abundant NOx emission there. Therefore, the increase of BVOC emission in the Qinling Mountains would further favor the formation of $O_3$ in the urban region of Xi'an.

**3.3 Comparison of Estimates of Isoprene Emission and Satellite Derived Formaldehyde Column Concentration**

The OMI HCHO VC product from 2005-2016 developed by BIRA-IASB (De Smedt et al., 2015) was used in this study. The interannual variability of isoprene emission estimated in this study was evaluated by comparing the summer (June-August) averaged isoprene emission with the summer averaged HCHO VC. The annually averaged LAI during 2005-2016 presented in Figure 9 indicates the spatial distribution of vegetation in China. However, the spatial pattern of estimated isoprene emission (Figure 9b) differs from the
spatial distribution of vegetation because of the variability of emission potentials among different PFTs in the MEGAN model as well as the climatic conditions. The spatial pattern of average summertime HCHO VC observed by the OMI sensor during 2005-2016 is also presented in Figure 9c. The highest summer HCHO concentrations in the US are mainly distributed in rural forest regions dominated by biogenic emission (Palmer et al., 2003), while the highest summer HCHO concentrations in China are mainly distributed in
developed regions like North China Plain where HCHO concentration is dominated by anthropogenic sources (Smedt et al., 2010). There is a moderate HCHO VC of about 6-10 $\times 10^{15}$ molec cm$^{-2}$ in the vegetation dominated regions of China.

The grid level correlation coefficients between the average summer HCHO VC and isoprene emission estimated in our study are shown in Figure 9d, and the grids with statistically significant correlations (p <
0.1, N=12) grids are marked with black dots. A correlation is found in the northeast, central and south of China where there are relatively high vegetation cover rates and low anthropogenic influence. In contrast, there's almost no statistically significant correlation in the high HCHO VC regions like the North China Plain which is dominated by anthropogenic emissions. However, the distribution of statistically significant positive correlated points is not completely consistent with the vegetation distribution indicated by LAI due to the
absence of consideration of physical and chemical processes, including transportation, diffusion, and chemical reactions. The grids with significant correlation are mostly distrusted in or near rural regions with high vegetation biomass indicating that our estimations can represent the annual variation of isoprene emission.

The increasing trends of isoprene and HCHO VC during 2005-2016 are presented in (e) and (f) of Figure 9,
and the statistically significant (p < 0.1) grids are marked with black dots. The increasing trend pattern of isoprene emission during 2005-2016 is basically consistent with that during 2001-2016, which has been described in the Section 3.2, and it is clear that southern China is the region with the strongest positive trend. For HCHO, developed regions such as the North China Plain have an increasing trend because of the increase

of human activities (Smedt et al., 2010), there is also an obvious increasing trend of HCHO VC at Yunnan and Guangxi provinces in the south of China. Moreover, these regions, especially Guangxi province also show a statistically significant positive correlation between isoprene emission and HCHO VC as presented in Figure 9d. This implies that biogenic emissions might be the main driver of the increased HCHO in Guangxi province, however, the absence of the physical and chemical processes like transport led to a large uncertainty to this conclusion. Here we conducted a primary comparison between HCHO VC and isoprene emission, and a more thorough study by using chemical transport model may help to further explain the relationship between the variability of HCHO VC and the isoprene emission in the future.

### 3.4 Comparison with other studies and uncertainties discussion

The comparison of isoprene and monoterpenes emission estimations between our estimations and previous studies is presented in Table 5. The estimations of isoprene emission range from 4.65 Tg to 33.21 Tg, and the estimations of monoterpenes emission range from 3.16 Tg to 5.6 Tg in China. Multiple factors including emission factor, meteorological and land cover inputs can lead to the discrepancy of these estimations. We listed the inputs of these estimations in Table 6 to fully understand the discrepancies between our results and other estimations.

The setting of inputs in this study is relatively close to the study by Stavrakou et al. (2014) and CAMS-GLOB-BIO biogenic emission inventories (https://eccad3.sedoo.fr/#CAMS-GLOB-BIO) that adopted the method described by Sindelarova et al. (2014). However, the estimation of isoprene emission in this study is about 86.6%-122.3% higher than their estimations, and the estimation of monoterpene emission is about 23.5% and 31.3% higher than that from CAMS-GLOB-BIO v3.1 and v1.1, respectively. We further compared our results with two versions of CAMS-GLOB-BIO inventories. Figure 10 and Figure 11 present the trends of isoprene emission and monoterpenes emission respectively from S1 and S3 in this study, CAMS-GLOB-BIO inventory v 1.1 and v 3.1 during 2001-2016. As shown in Figure 10 and Figure 11, S3 shows similar spatial patterns and magnitude of changing trend of isoprene and monoterpenes emissions with CAMS-GLOB-BIO v 1.1 and CAMS-GLOB-BIO v3.1, e.g. three datasets all showed a strong increasing trend in Yunnan province, and S1 shows much more stronger changing trends comparing with other three datasets with annually updated LAI and PFT datasets. The meteorological inputs for CAMS-GLOB-BIO v1.1 and v3.1 are ERA-Interim and ERA-5 reanalysis data, respectively, and the WRF model used in this study was also driven by ERA-Interim reanalysis data. Therefore, the four datasets have the similar source of meteorological inputs. In addition, these estimations all adopted the same PFT level emission factors from Guenther et al. (2012). Therefore, the potential reason for the differences of isoprene and monoterpenes emission among the datasets in Figure 10 and Figure 11 is the discrepancies of PFT and LAI inputs. CAMS-GLOB-BIO also adopted the annually updated LAI inputs developed by Yuan et al. (2011) based on MODIS

MOD15A v5 LAI product, but the two versions of CAMS-GLOB-BIO inventory did not show a same level strong increasing trend with S1. The increasing trend of LAI in China is agreed by multiple LAI products but with different rates (Piao et al., 2015; Chen et al., 2020). In this study, we adopted the latest MODIS LAI product of version 6, and a strong increasing trend of LAI in China has been found by using this product (Chen et al., 2019). Therefore, an increasing trend of BVOC emission induced by LAI should be seen in the estimation with annually updated LAI inputs, but the magnitude of this trend is also affected by the magnitude of changing trend of LAI products. The PFT map used in this study is coming from MODIS land cover product, which is a mesoscale satellite product with the highest resolution of 500m. Besides the quality of the product, the method for converting the original land cover classification system to PFT classification system is also important. Hartley et al. (2017) illustrated that the cross-walking table for converting land cover class maps to PFT fractional maps can lead to 20%-90% uncertainties for gross primary production estimation in land surface model by using different vegetation fractions for mixed pixels, and the BVOC emission estimation has the same issue. In this study, we assumed that the pixels that were assigned as vegetation is 100% covered by that kind of vegetation (Table S1 in the supplement). Therefore, it will lead to an overestimation of vegetation cover rate for mixed pixels, which can lead to higher BVOC emission. The emission factor is also an important source of uncertainties, and it decided the spatial patterns of emission rates together with the PFT distribution. In order to understand the role of emission factor, the flux measurements of isoprene and monoterpenes from the campaigns conducted during 2010 to 2016 in China (Bai et al., 2015; Bai et al., 2016; Bai et al., 2017) were collected and compared with model results in this study. The details of these campaigns are provided in Table 7, and the emission factors that were retrieved from the observations are also listed for these sites. Most samples were collected during the daytime every 3 hours according to the descriptions of the measurements (Bai et al., 2015;Bai et al., 2016;Bai et al., 2017), therefore, we averaged the model results during 8:00 A.M. to 20 A.M in local time with a three-hour interval for comparison. As shown in the (a) and (b) of Figure 12, the modeled fluxes of isoprene and monoterpenes with the default emission factors in this study did not capture the variability of the observations. The ME, MB and RMSE are 1.60, 1.59 and 2.31 mg m$^{-2}$ h$^{-1}$ for isoprene and 0.21, -0.003 and 0.32 mg m$^{-2}$ h$^{-1}$ for monoterpenes. When we adopted the emission factors retrieved from observations (Bai et al., 2015;Bai et al., 2016;Bai et al., 2017), the simulated isoprene and monoterpenes fluxes showed relatively good consistence with the observations by using the same activity factor from the model ($\gamma$ in equation (1)) as shown in (c) and (b) of Figure 12. The ME, MB and RMSE are 0.44, 0.41 and 0.57 mg m$^{-2}$ h$^{-1}$ for isoprene and 0.32, 0.14 and 0.49 mg m$^{-2}$ h$^{-1}$ for monoterpenes after adopting the observation-based emission factors, and the statistic parameters for isoprene simulation are largely improved. Although the MB and ME of monoterpenes simulation are increased, but the simulated monoterpenes flux show better agreement with observations

(Figure 12). Therefore, it is clear that our calculation of activity factors is in a reasonable range, but the emission factor is the main source of uncertainties. The PFT level emission factors used in this study from Guenther et al. (2012) represents the globally averaged emission factor for PFTs, and it is relatively easy to use them with the satellite PFT products. Therefore, the most studies listed in Table 6 adopted the PFT/landuse level emission factors. Our validation showed that the accurate emission factor based on observations could largely improve the performance of the MEGAN model, but it also requires abundant efforts to conduct measurements. However, the measurements listed in Table 7 are still very limited for describing the spatial discrepancies of ecosystems in China, so we still used the default emission factors in MEGAN model for our national scale estimation. The estimations by Li et al. (2013, 2020) used the species level emission factors and Vegetation Atlas of China for 2007 to describe the spatial distribution of BVOC emission potentials, and they concluded the reason why their estimations were far higher than other studies is the high emission factors they adopted. Therefore, the same validations by using canopy-scale BVOC flux measurements are also needed for these studies to validate and constrain the emission factors they used.

Meteorological input is also a source of uncertainties for BVOC emission estimation. As shown in Figure 12, the modeled isoprene and monoterpenes fluxes are still generally higher than observations when observation-based emission factors were used. One potential reason for this phenomenon is the overestimation of temperature and radiation as described in Section 2.3. The sensitivity tests by Wang et al. (2011) showed that the about 1.89 °C discrepancy of temperature can result in -19.2 to 23.2% change of isoprene emission and -16.2 to 18.5% change of monoterpenes emission for Pearl River Delta region in July, where is also a hotspot for BVOC emission in this study. They also found that 115.8 W m$^{-2}$ discrepancy of DSW can result in -31.4 to 36.2% change of isoprene emission and -14.3 to 16.8% change of monoterpenes emission for the same region. The BVOC emission in this study might be overestimated because of the overestimated temperature and DSW in meteorological inputs. However, inaccurate emission factors could lead to over 100% uncertainties, which is more significant than the uncertainties induced by meteorological inputs.

**4. Conclusion**

Satellite observations have shown that China has led the global greening trend in recent decades (Chen et al., 2019). In this study, we investigated the impact of this greening trend on BVOC emission in China from 2001 to 2016. We used long-term satellite vegetation products as inputs in the MEGAN. According to the estimation of model, we found the greening trend of China is leading a national scale increase of BVOC emission. The BVOC emission level in 2016 can be 11.7% higher than that in 2001 because of higher tree cover fraction and biomass. The comparison among different scenarios showed that vegetation changes resulting from land cover management is the main driver of BVOC emission change in China. Climate

variability contributed significantly to interannual variations but not much to the long-term trend during the study period.

On regional scales, there are strong increasing trends in 1) northeastern China, 2) Beijing and its surrounding areas, 3) the Qinling Mountains, 4) Yunnan province, 5) Guangdong-Guangxi provinces and 6) Hainan island. A strong increasing trend of broadleaf tree cover fractions and LAIv were found in these regions. The mean total tree cover fraction during the last four years (2013-2016) is 5.7-82.5 % higher than that of the first four years (2001-2004) for these regions, and the LAIv during 2013-2016 increased by 14.8 ~ 26.4 % comparing to that during 2001-2004 in these regions. Consequently, the average BVOC emission flux for the last four years (2013-2016) is 8.6%~19.3% higher than that for the first four years (2001-2004) in the sub-regions we defined driven by the same meteorological inputs. In the standard scenario of S1, a statistic significant increasing trend still could be found in the sub-regions including Beijing and its surroundings, Yunnan province and Hainan island when the climate variability was considered.

We used the long-term record of satellite HCHO VC from the OMI sensor to assess our estimation of isoprene emission in China during 2005-2016. The results indicated statistically significant positive correlation coefficients between the isoprene emission estimate and satellite HCHO VC in summer over the regions with high vegetation cover fraction including the northeast, central and southern China. In addition, isoprene emission and HCHO VC both had a statistically significant increasing trend in the south of China, mainly Guangxi Province, where there was a statistically significant positive correlation supporting the estimated variability of BVOC emission in China. However, the absence of the physical and chemical processes, e.g. transport, led to a large uncertainty to this conclusion, and a more thorough study by using chemical transport model may help to further explain the relationship between the variability of HCHO VC and the isoprene emission.

We conclude that uncertainties of this study mainly come from the emission factor, PFT and LAI inputs through comparing our results with other studies and flux measurements during 2010-2016 in China. The validation with flux measurements suggested that using the observation-based emission factor could largely improve the performance of model, but it also requires more much more efforts. Our results suggest that the continued increase of BVOC will enhance the importance of considering BVOC when making policies for controlling ozone pollution in China along with ongoing efforts to increase the cover fraction of forest.

**Author Contribution**

QW, LW and HW planned and organized the project. HW, JF and QX prepared the input datasets. HW modelled and analyzed the data. HW and QW wrote the manuscript. HW, AG and QW revised the manuscript. AG, XY, LN, XT, JL, JF and HC reviewed and provided key comments on the paper.

**Data Availability**

The source code of MEGAN v2.1 is available at https://bai.ess.uci.edu/. The MODIS MCD12C1 land cover product Version 6 and MODIS MCD15A2 LAI Version 6 and MODIS MOD44B VCF Version 6 datasets are available on the website of the Land Processes Distributed Active Archive Center (LP DAAC) at https://lpdaac.usgs.gov/dataset_discovery/modis/modis_products_table. The version 14 Level 3 OMI HCHO VC product were downloaded from the website of Tropospheric Emission Monitoring Internet Service (TEMIS) at http://h2co.aeronomy.be.

**Competing Interests**

The authors declare no competing financial interest.

**Acknowledgements**

The National Key R&D Program of China (2017YFC0209805 and 2016YFB0200800), the National Natural Science Foundation of China (41305121), the Fundamental Research Funds for the Central Universities and Beijing Advanced Innovation Program for Land Surface funded this work.

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

**Table 1. Description of different scenarios used to estimate the BVOC emission.**

|    | Land Cover | LAIv | Meteorological conditions |
|----|------------|------|---------------------------|
| S1 | Annually updated | Annually updated | Annually updated |
| S2 | Annually updated | Annually updated | Year 2001 |
| S3 | Year 2001 | Year 2001 | Annually updated |
| S4 | Year 2016 | Year 2016 | Annually updated |
| S5 | Year 2001 | Annually updated | Year 2001 |

**Table 2. The mean annual emission (Tg) of different species in China during 2001 to 2016. The scenarios S1 to S5 are described in Table 1.**

|  | S1 | S2 | S3 | S4 | S5 |
|--|----|----|----|----|----|
| Isoprene | 15.94 (±1.12) | 15.40 (±0.66) | 14.63 (±0.76) | 16.70 (±0.89) | 15.29 (±0.54) |
| Monoterpenes | 3.99 (±0.17) | 3.91 (±0.10) | 3.78 (±0.12) | 4.12 (±0.14) | 3.9 (±0.08) |
| Sesquiterpenes | 0.50 (±0.03) | 0.48 (±0.02) | 0.47 (±0.02) | 0.51 (±0.03) | 0.48 (±0.02) |
| Other VOCs | 13.84 (±0.78) | 13.95 (±0.34) | 12.89 (±0.66) | 14.15 (±0.73) | 13.95 (±0.34) |
| Total BVOCs | 34.27 (±2.06) | 33.74 (±1.10) | 31.77 (±1.54) | 35.48 (±1.76) | 33.63 (±0.95) |

**Table 3. The change and trend of annual emission flux (S1, S2 and S5), cover fractions of main PFTs, LAIv, growing season temperature and DSW in northeastern China, Beijing and its surrounding areas and the Qinling Mountains.**

Northeastern China

|  | BVOC Emission (S2, g m$^{-2}$) | BVOC Emission (S1, g m$^{-2}$) | BVOC Emission (S5, g m$^{-2}$) | LAIv (m$^{-2}$ m$^{-2}$) | BLT Cover Fraction (%) | NLT Cover Fraction (%) | Shrub Cover Fraction (%) | Grass Cover Fraction (%) | Crop Cover Fraction (%) | 2-m Temp (°C) | DSW (W m$^{-2}$) |
|--|--|--|--|--|--|--|--|--|--|--|--|
| **Average** | 3.37 (±0.13) | 3.04 (±0.36) | 3.25 (±0.06) | 1.45 (±0.1) | 21.37 (±1.56) | 13.56 (±0.12) | 5.97 (±0.16) | 30.86 (±1.8) | 25.85 (±0.3) | 13.74 (±0.67) | 224.5 (±6.08) |
| **Average (2001-2004)** | 3.21 (±0.05) | 2.9 (±0.32) | 3.19 (±0.06) | 1.34 (±0.05) | 19.37 (±0.51) | 13.57 (±0.04) | 6.06 (±0.15) | 33.1 (±0.61) | 25.71 (±0.21) | 13.89 (±0.41) | 227.54 (±5.22) |
| **Average (2013-2016)** | 3.52 (±0.06) | 3.07 (±0.36) | 3.3 (±0.05) | 1.55 (±0.12) | 23.1 (±0.04) | 13.45 (±0.13) | 5.87 (±0.04) | 28.7 (±0.18) | 26.22 (±0.25) | 13.42 (±0.59) | 218.39 (±4.74) |
| **Trend** | 0.02[***a] | 0.01 | 0.01[**] | 0.02[**] | 0.31[***] | -0.01 | -0.03[**] | -0.37[***] | 0.06[***] | -0.03 | -0.73[**] |

| | BVOC Emission (S2, g m⁻²) | BVOC Emission (S1, g m⁻²) | BVOC Emission (S5, g m⁻²) | LAIv (m⁻² m⁻²) | BLT Cover Fraction (%) | NLT Cover Fraction (%) | Shrub Cover Fraction (%) | Grass Cover Fraction (%) | Crop Cover Fraction (%) | 2-m Temp (°C) | DSW (W m⁻²) |
|---|---|---|---|---|---|---|---|---|---|---|---|
| **Average** | 2.94 (±0.21) | 2.58 (±0.25) | 2.76 (±0.08) | 1.24 (±0.1) | 4.96 (±1.17) | 0.61 (±0.25) | 2.74 (±0.63) | 58.18 (±1.58) | 27.48 (±1.33) | 17.68 (±0.65) | 251.09 (±3.22) |
| **Average (2001-2004)** | 2.7 (±0.14) | 2.41 (±0.09) | 2.67 (±0.12) | 1.13 (±0.11) | 3.83 (±0.16) | 0.35 (±0.03) | 1.99 (±0.11) | 58.73 (±0.07) | 29.44 (±0.33) | 17.87 (±0.6) | 250.47 (±4.56) |
| **Average (2013-2016)** | 3.22 (±0.08) | 2.81 (±0.29) | 2.81 (±0.04) | 1.3 (±0.08) | 6.66 (±0.44) | 0.97 (±0.14) | 3.6 (±0.27) | 55.84 (±1.26) | 26.51 (±0.41) | 17.52 (±0.75) | 250.42 (±1.95) |
| **Trend** | 0.04*** | 0.03* | 0.01** | 0.01* | 0.23*** | 0.04*** | 0.13*** | -0.18* | -0.26*** | -0.03 | 0.02 |

| | BVOC Emission (S2, g m⁻²) | BVOC Emission (S1, g m⁻²) | BVOC Emission (S5, g m⁻²) | LAIv (m⁻² m⁻²) | BLT Cover Fraction (%) | NLT Cover Fraction (%) | Shrub Cover Fraction (%) | Grass Cover Fraction (%) | Crop Cover Fraction (%) | 2-m Temp (°C) | DSW (W m⁻²) |
|---|---|---|---|---|---|---|---|---|---|---|---|
| **Average** | 9.25 (±0.38) | 9.29 (±0.93) | 9.10 (±0.28) | 1.8 (±0.19) | 44.08 (±1.52) | 12.25 (±0.17) | 14.05 (±0.64) | 14.67 (±0.67) | 12.15 (±0.25) | 20.78 (±0.58) | 219.93 (±9.01) |
| **Average (2001-2004)** | 8.84 (±0.25) | 8.91 (±0.38) | 8.85 (±0.25) | 1.59 (±0.17) | 42.18 (±0.32) | 12.48 (±0.11) | 14.84 (±0.29) | 15.51 (±0.26) | 12.31 (±0.32) | 20.83 (±0.25) | 220.28 (±9.41) |
| **Average (2013-2016)** | 9.71 (±0.22) | 9.75 (±1.64) | 9.39 (±0.22) | 2.01 (±0.12) | 45.91 (±0.27) | 12.07 (±0.03) | 13.26 (±0.14) | 13.84 (±0.16) | 11.95 (±0.10) | 20.75 (±0.91) | 221.26 (±12.30) |
| **Trend** | 0.06*** | 0.07 | 0.04** | 0.03*** | 0.32*** | -0.03*** | -0.13*** | -0.14*** | -0.04** | -0.01 | -0.11 |

a: *: p<0.1; **: p<0.05; ***: p<0.01;

**Table 4. The change and trend of annual emission flux (S1, S2 and S5), cover fractions of main PFTs, LAIv, growing season temperature and DSW in Yunnan province, Guangxi-Guangdong provinces and Hainan island.**

| | BVOC Emission (S2, g m⁻²) | BVOC Emission (S1, g m⁻²) | BVOC Emission (S5, g m⁻²) | LAIv (m⁻² m⁻²) | BLT Cover Fraction (%) | NLT Cover Fraction (%) | Shrub Cover Fraction (%) | Grass Cover Fraction (%) | Crop Cover Fraction (%) | 2-m Temp (°C) | DSW (W m⁻²) |
|---|---|---|---|---|---|---|---|---|---|---|---|
| **Average** | 6.79 (±0.26) | 7.28 (±0.54) | 6.67 (±0.21) | 2.23 (±0.17) | 32.7 (±0.83) | 14.92 (±0.32) | 17.25 (±0.12) | 21.83 (±0.52) | 9.86 (±0.71) | 18.54 (±0.31) | 224.71 (±5.64) |
| **Average (2001-2004)** | 6.53 (±0.28) | 6.76 (±0.45) | 6.57 (±0.30) | 2.02 (±0.19) | 32.1 (±0.19) | 14.51 (±0.04) | 17.22 (±0.14) | 22.45 (±0.20) | 10.34 (±0.53) | 18.35 (±0.30) | 219.18 (±6.70) |
| **Average (2013-2016)** | 7.09 (±0.09) | 7.92 (±0.35) | 6.94 (±0.04) | 2.4 (±0.02) | 33.93 (±0.58) | 15.33 (±0.1) | 17.2 (±0.17) | 21.12 (±0.30) | 8.92 (±0.25) | 18.7 (±0.47) | 227.49 (±2.65) |
| **Trend** | 0.04***a | 0.1*** | 0.02** | 0.03*** | 0.15*** | 0.07*** | 0 | -0.11*** | -0.17*** | 0.03** | 0.42 |

| | BVOC Emission (S2, g m⁻²) | BVOC Emission (S1, g m⁻²) | BVOC Emission (S5, g m⁻²) | LAIv (m⁻² m⁻²) | BLT Cover Fraction (%) | NLT Cover Fraction (%) | Shrub Cover Fraction (%) | Grass Cover Fraction (%) | Crop Cover Fraction (%) | 2-m Temp (°C) | DSW (W m⁻²) |
|---|---|---|---|---|---|---|---|---|---|---|---|
| **Average** | 15.53 (±0.79) | 16.23 (±1.59) | 15.57 (±0.67) | 2.24 (±0.22) | 32.92 (±1.6) | 9.08 (±0.27) | 19.13 (±0.38) | 20.47 (±0.60) | 9.89 (±0.70) | 26.32 (±0.67) | 258.72 (±7.32) |
| **Average (2001-2004)** | 15.06 (±1.09) | 15.84 (±1.70) | 15.23 (±1.23) | 2.1 (±0.35) | 32.2 (±0.57) | 9.3 (±0.02) | 19.41 (±0.04) | 21.02 (±0.03) | 9.89 (±0.57) | 26.36 (±0.25) | 258.74 (±9.25) |
| **Average (2013-2016)** | 16.36 (±0.37) | 17.03 (±1.99) | 15.92 (±0.29) | 2.44 (±0.09) | 35.24 (±0.88) | 8.69 (±0.19) | 18.57 (±0.31) | 19.62 (±0.32) | 9.03 (±0.16) | 26.31 (±0.99) | 256.36 (±4.26) |
| **Trend** | 0.13*** | 0.14 | 0.05 | 0.03** | 0.32*** | -0.05*** | -0.06*** | -0.12*** | -0.14** | 0.02 | -0.24 |

**Hainan Island**

| | BVOC Emission (S2, g m⁻²) | BVOC Emission (S1, g m⁻²) | BVOC Emission (S5, g m⁻²) | LAIv (m⁻² m⁻²) | BLT Cover Fraction (%) | NLT Cover Fraction (%) | Shrub Cover Fraction (%) | Grass Cover Fraction (%) | Crop Cover Fraction (%) | 2-m Temp (°C) | DSW (W m⁻²) |
|---|---|---|---|---|---|---|---|---|---|---|---|
| **Average** | 17.79 (±0.73) | 17.98 (±1.40) | 17.57 (±0.51) | 2.43 (±0.20) | 39.44 (±1.46) | 0 | 17.41 (±0.14) | 22.2 (±1.12) | 8.67 (±0.56) | 27.3 (±0.47) | 257.51 (±4.55) |
| **Average (2001-2004)** | 17.16 (±0.72) | 17.51 (±1.04) | 17.27 (±0.80) | 2.3 (±0.26) | 38.07 (±0.52) | 0 | 17.46 (±0.18) | 23.63 (±0.04) | 8.79 (±0.33) | 27.38 (±0.22) | 259.79 (±7.28) |
| **Average (2013-2016)** | 18.68 (±0.27) | 19.44 (±1.89) | 18.07 (±0.24) | 2.64 (±0.14) | 41.11 (±0.23) | 0 | 17.31 (±0.08) | 20.9 (±0.28) | 8.14 (±0.07) | 27.41 (±0.78) | 258.39 (±3.95) |
| **Trend** | 0.13*** | 0.12* | 0.06** | 0.03* | 0.27*** | **0** | -0.02 | -0.22*** | -0.07** | 0 | -0.13 |

a: *: p<0.1; **: p<0.05; ***: p<0.01;

**Table 5. Comparison of isoprene and monoterpene emissions (Tg) in China with previous studies.**

| Data Source | Isoprene | Monoterpene | Study period | Method or Model |
|---|---|---|---|---|
| This study | 15.94 (±1.12) | 3.99 (±0.17) | 2001-2016 | MEGAN |
| Stavrakou et al. (2014) | 7.17 (±0.30) | - | 2007-2012 | MEGAN-MOHYCAN |
| Li et al. (2013) | 23.4 | 5.6 | 2003 | MEGAN |
| Li et al. (2020) | 33.21 | 6.35 | 2008-2018 | MEGAN |
| CAMS-GLOB-BIO v1.1 (Sindelarova et al., 2014) | 7.67 | 3.04 | 2001-2016 | MEGEN |
| CAMS-GLOB-BIO v3.1 (Sindelarova et al., 2014) | 8.54 | 3.23 | 2001-2016 | MEGAN |

| | | | | |
|---|---|---|---|---|
| Fu and Liao (2012) | 10.87 | 3.21 | 2001-2006 | GEOS-Chem-MEGAN |
| Tie et al. (2006) | 7.7 | 3.16 | 2004 | Guenther et al. (1993) |
| Klinger et al. (2002) | 4.65 | 3.97 | 2000 | Guenther et al. (1995) |
| Guenther et al. (1995) | 17 | 4.87 | 1990 | Guenther et al. (1995) |

**Table 6. Comparison of inputs for BVOC estimation with previous studies.**

| Reference | Emission Factor Type | Emission Factor Reference | PFT/Land use | LAI/Biomass | Meteorology | Model/Algorithms |
|---|---|---|---|---|---|---|
| This study | PFT level emission factors | Guenther et al. (2012) | MODIS MCD12C1 v6 | MODIS MCD15A2H v5 | WRF Model v3.9 | MEGANv2.1 |
| Stavrakou et al. (2014) | PFT level emission factors | Guenther et al. (2006) | Ramankutty and Foley (1999) | MODIS MOD15A2 v5 | ERA-Interim Dataset | MEGAN-MOHYCAN |
| Li et al. (2013) | Vegetation genera/species level emission factors | Li et al. (2013) | Vegetation Atlas of China for year 2007 | MEGAN database for 2003 | MM5 Model v3.7 | MEGAN |
| Li et al. (2020) | Vegetation genera/species level emission factors | Li et al. (2013) | Vegetation Atlas of China for year 2007 | Estimations based on surveys and statistics | WRF Model v3.8 | MEGAN |
| CAMS-GLOB-BIO v1.1 (Sindelarova et al., 2014) | PFT level emission factors | Guenther et al. (2012) | 16 plant functional types consistent with the Community Land Model | MODIS MOD15A2 v5 | ERA-Interim Dataset | MEGAN |
| CAMS-GLOB-BIO v3.1 (Sindelarova et al., 2014) | PFT level emission factors | Guenther et al. (2012) | 16 plant functional types consistent with the Community Land Model | MODIS MOD15A2 v5 | ERA-5 Dataset | MEGAN |
| Fu and Liao (2012) | PFT level emission factors | Guenther et al. (1995) Lathière et al. (2006) Levis et al. (2003) Bai et al. (2006) | MODIS MCD12Q1 v5 | MODIS MOD15A2 v5 | GEOS-4 Meteorology | GEOS-Chem-MEGAN |
| Tie et al. (2006) | Landuse level emission factors | Landuse-based emission rates | USGS 1km land use data | / | WRF model | Guenther et al. (1993) |
| Klinger et al. (2002) | Vegetation genera/species level emission factors | Klinger et al. (2002) | Province-level Forest Inventory | / | Monthly meteorology database by (Leemans and Cramer, 1991) | Guenther et al. (1995) |

| | | | | | Monthly | |
| Guenther et al. (1995) | PFT level emission factors | Guenther et al. (1995) | Grided Global Ecosystem Types | Estimations from NPP | meteorology database by (Leemans and Cramer, 1991) | Guenther et al. (1995) |

**Table 7. Detailed descriptions of the flux measurements used in this study and corresponding campaigns.**

| Reference | Site Location | Sample Collection Periods | Ecosystem Type | Isoprene Emission Factor (mg m$^{-2}$ h$^{-1}$) | Monoterpenes Emission Factor (mg m$^{-2}$ h$^{-1}$) |
|---|---|---|---|---|---|
| Bai et al. (2015) | Changbai Mountain (42°24′ N, 128°6′) | 28 June -9 July 2010; 19 July -30 July 2010; 12 Aug.- 25 Aug. 2010; 19 June - 30 June 2011; 10 July -16 July 2011; 22 July - 29 July 2011; 5 Sep. - 8 Sep. 2011. | Mixed forest | 4.3 | 0.32 |
| Bai et al. (2016) | An Ji, Zhejiang (30°40′15″ N, 119°40′15″) | 7 July-13 July 2012; 20 Aug.-26 Aug. 2012; 25 Sep.-1 Oct. 2012; 28 Oct.- 5 Nov. 2012. | Moso bamboo forest | 3.3 | 0.008 |
| Bai et al. (2017) | Taihe, Jiangxi (26°44′48″ N, 115°04′13″) | 22 May -28 May 2013; 29 June - 6 July 2013; 6 Aug. -13 Aug. 2013; 7 Sep. -11 Sep. 2013; 18 Jan. -19 Jan. 2014; 23 July - 27 July 2014; 2 Nov. - 7 Nov. 2015; 31 Dec. 2015 -4 Jan. 2016. | Subtropical Pinus forest | 0.71 | 1.65 |

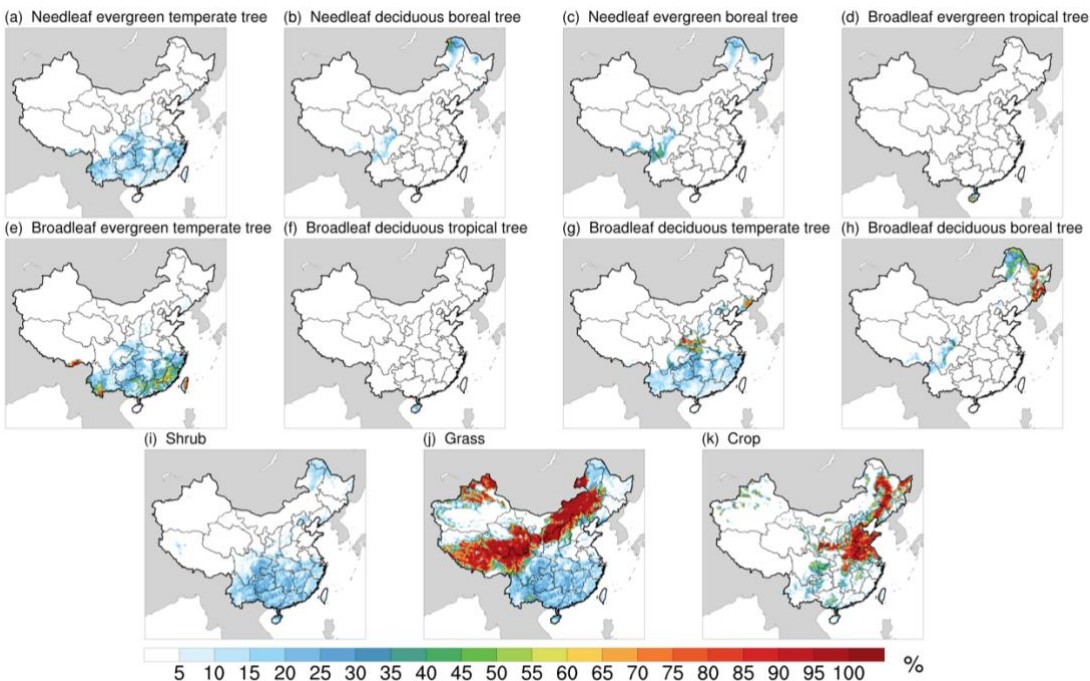

**Figure 1. The cover factions of different PFTs for the year 2016.**

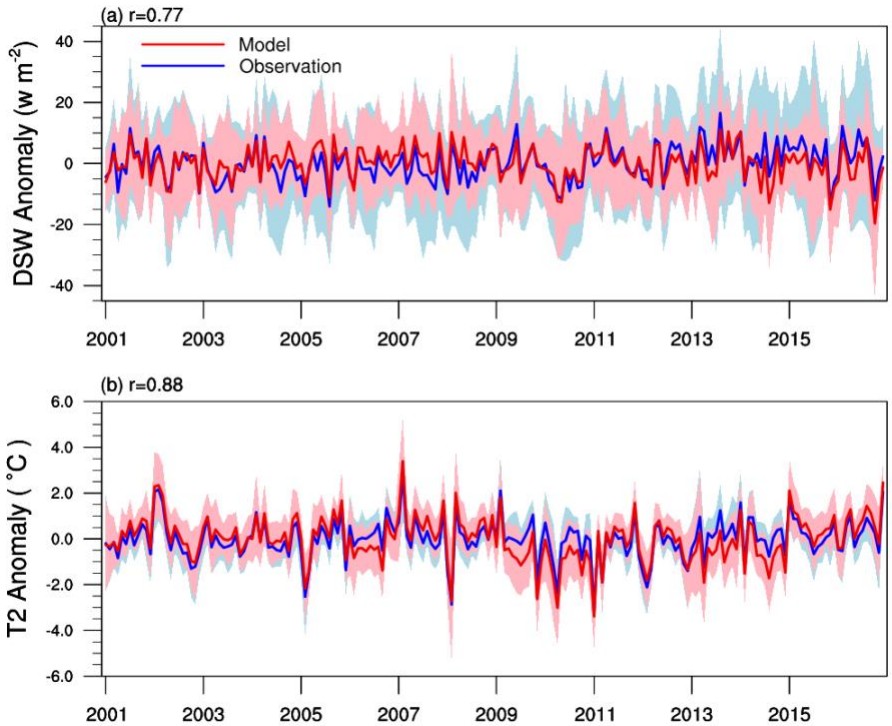

5   **Figure 2. The comparison of monthly anomaly of downward shortwave (DSW) radiation (a) and 2-meter temperature (T2) (b) for model simulation and in-situ observation and the filled areas present the standard deviations among 98 sites for DSW and 697 sites for T2.**

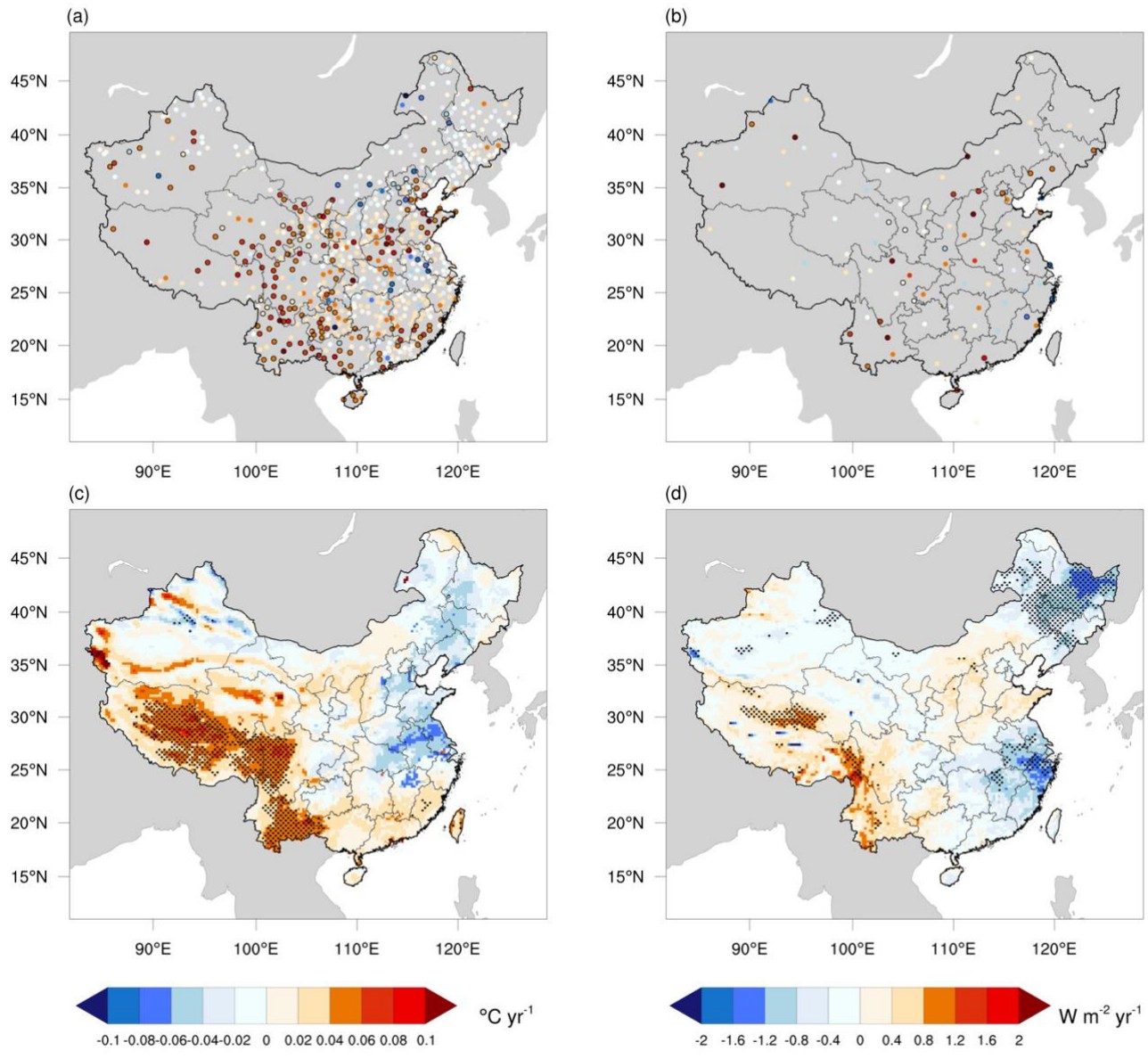

**Figure 3. The trend of growing season averaged 2-meter temperature (T2) and downward shortwave radiation (DSW). (a) and (b) are for in-situ T2 and DSW, respectively, and the sites with statistically significant trend are marked by black circles. (c) and (d) are for the WRF simulated T2 and DSW, respectively, and the regions with statistically significant trend are illustrated by shadow.**

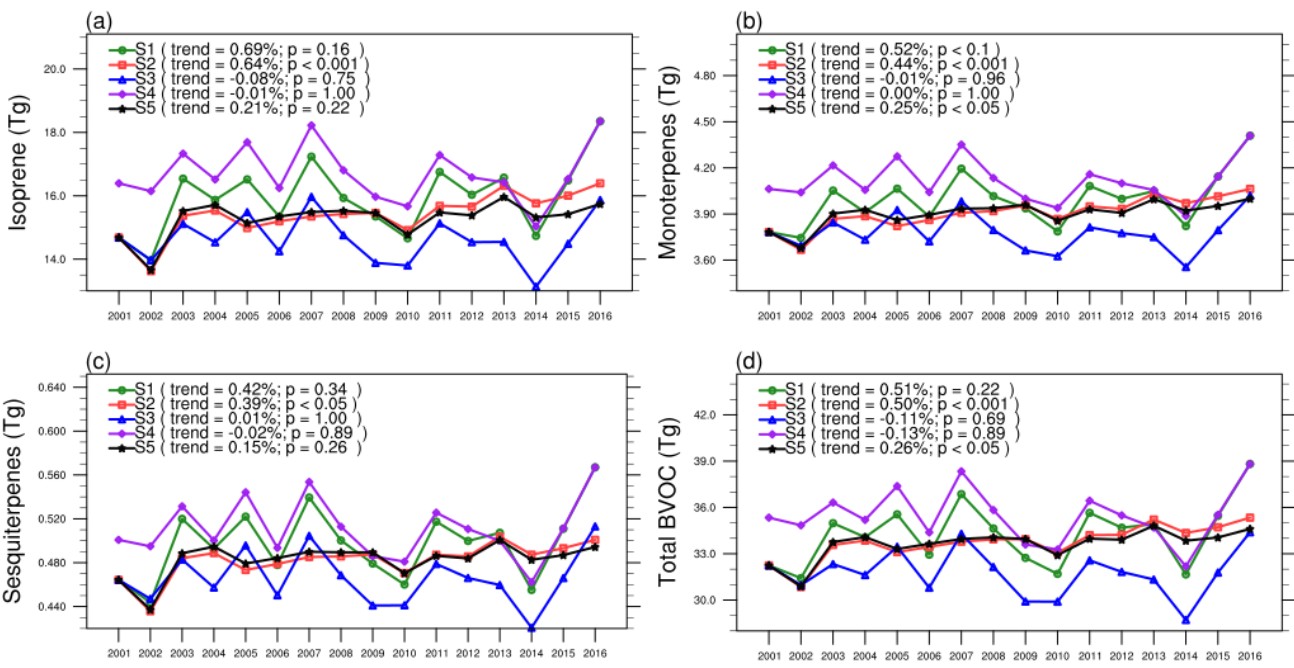

**Figure 4. Annual BVOC emissions in China during 2001 to 2016 for five scenarios (S1-S5) described in Table 1. The increasing trends and the probabilities (p) using the Mann-Kendall test are shown in the legend.**

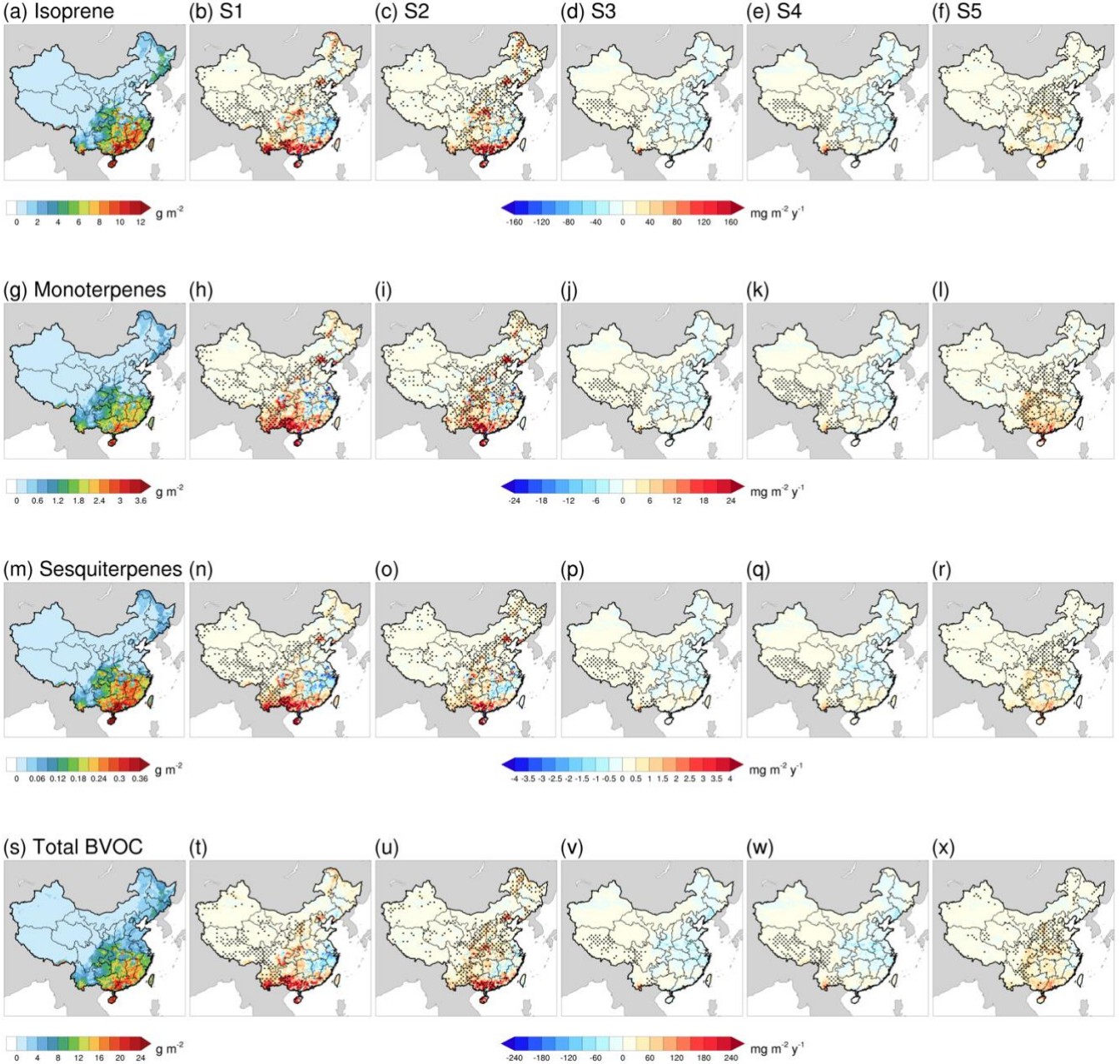

**Figure 5. The horizontal distributions of isoprene, monoterpenes, sesquiterpenes and total BVOCs emissions of China in 2001 are shown in figure (a), (g), (m) and (s), respectively. The rest of the columns of figures present the changing trend of isoprene (b-f), monoterpenes (h-l), sesquiterpenes (n-r) and total BVOCs (t-x) in S1, S2, S3, S4 and S5, respectively. The Mann-Kendall test was used to mark the grids where the p is smaller than 0.1.**

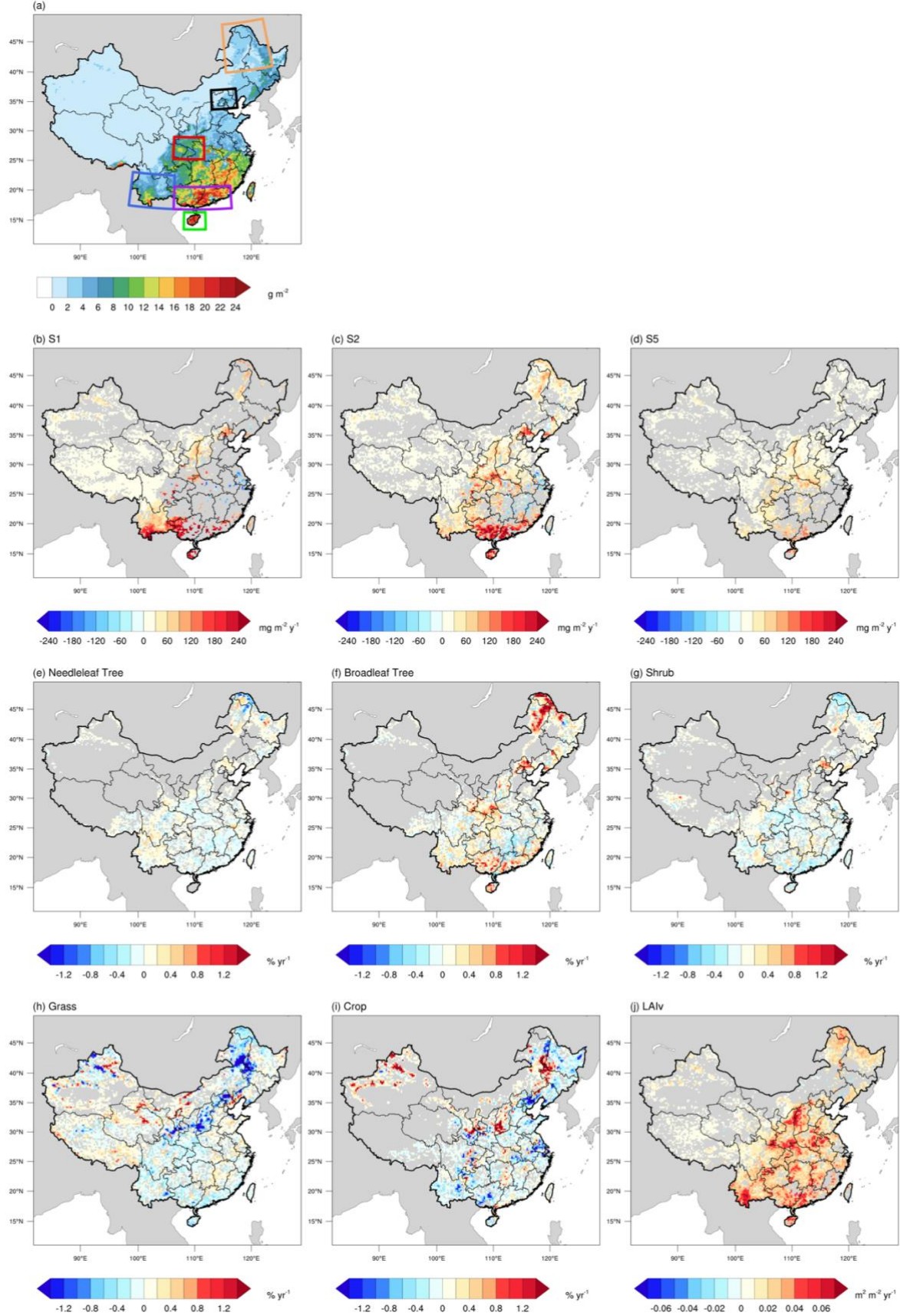

**Figure 6. Spatial distribution of BVOC emission in 2001 (a) and the changing trends of annual emission flux (S1, S2 and S5), cover fractions of main PFTs and LAIv. The Mann-Kendall test was used to filter the grids where the p is greater than 0.1.**

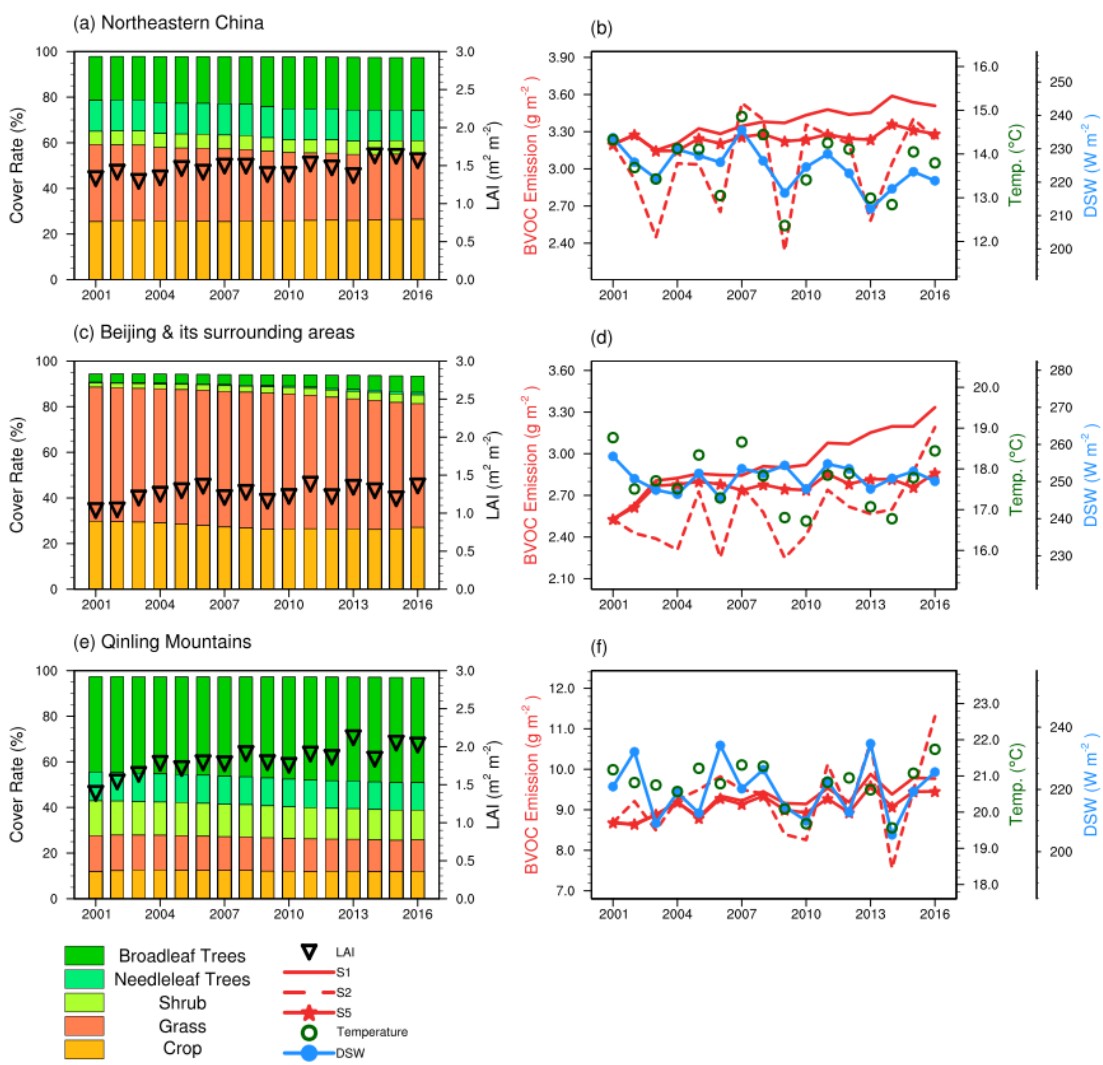

**Figure 7. The annual changes of PFTs, the annual emission amount of BVOC and LAI in (a) northeastern China, (b) Beijing and its surroundings, and the (c) Qinling mountains. The solid, dashed and marked line represents the mean emission flux rate of total BVOC in S1, S2 and S5, respectively.**

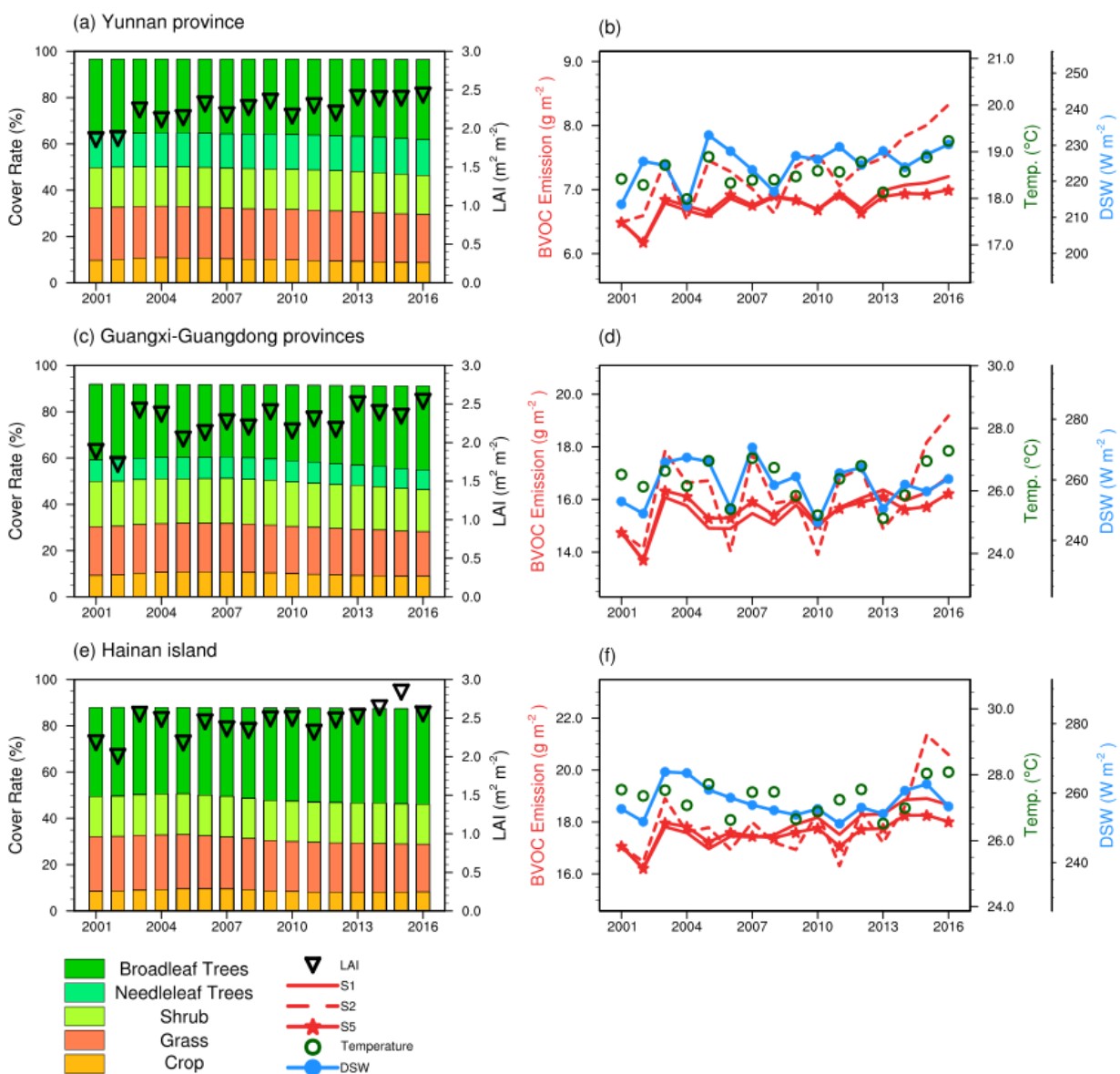

**Figure 8. The annual changes of PFTs, the annual emission amount of BVOC and LAI in (a) southwestern China, (b) southern, and (c) Hainan island. The solid, dashed and marked line represents the mean emission flux rate of total BVOC in S1, S2 and S5, respectively.**

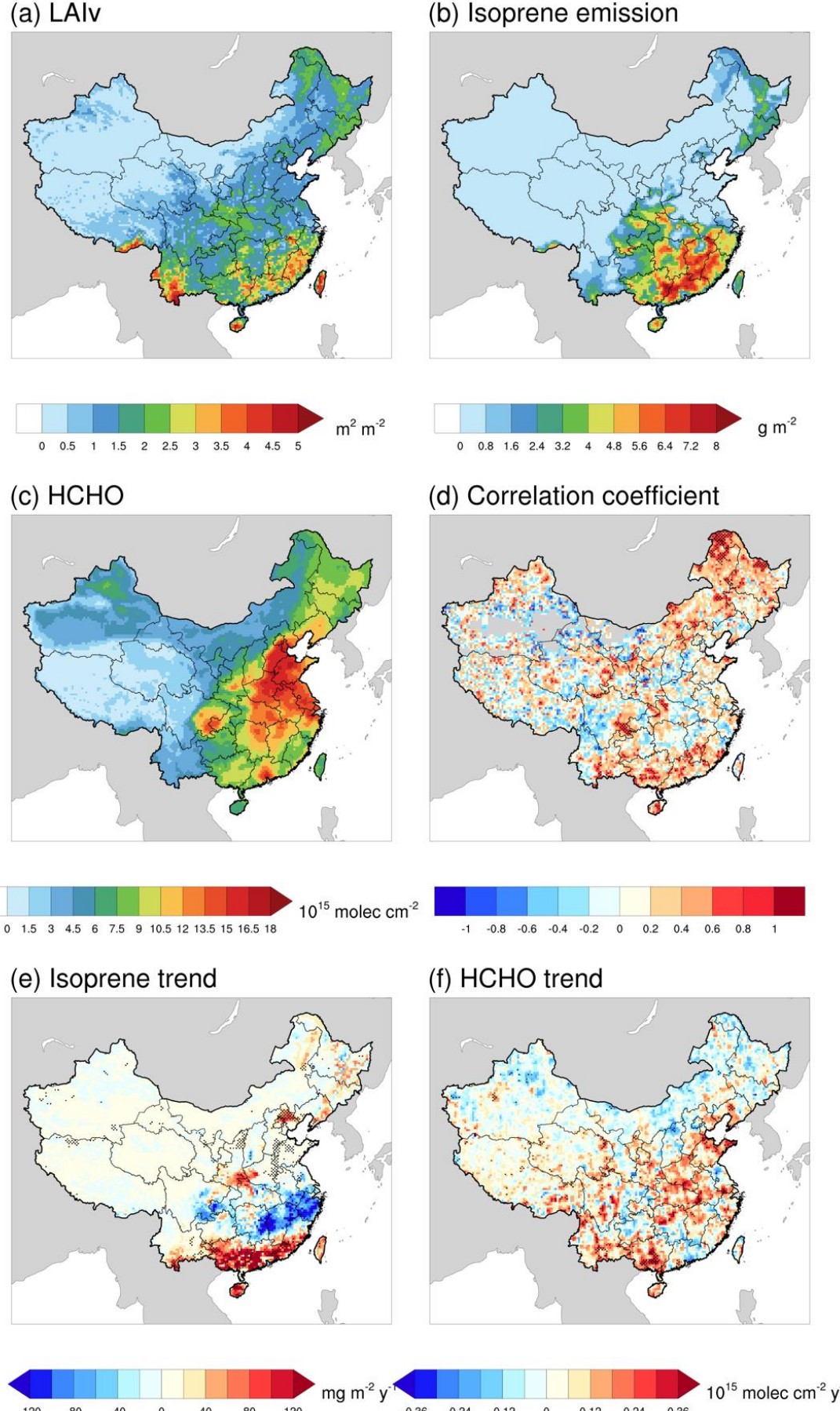

**Figure 9. Comparison of estimated isoprene annual emission with the satellite derived tropospheric HCHO vertical column concentration by OMI during 2005-2016. (a), (b) and (c) illustrate the spatial patterns of annual mean LAIv, isoprene emission and HCHO vertical columns (VC) by OMI respectively. (d) presents the spatial distribution of the correlation coefficient between summertime isoprene emission and HCHO VC. (e) and (f) shows the increasing trend of isoprene and HCHO VC during 2005-2016.**

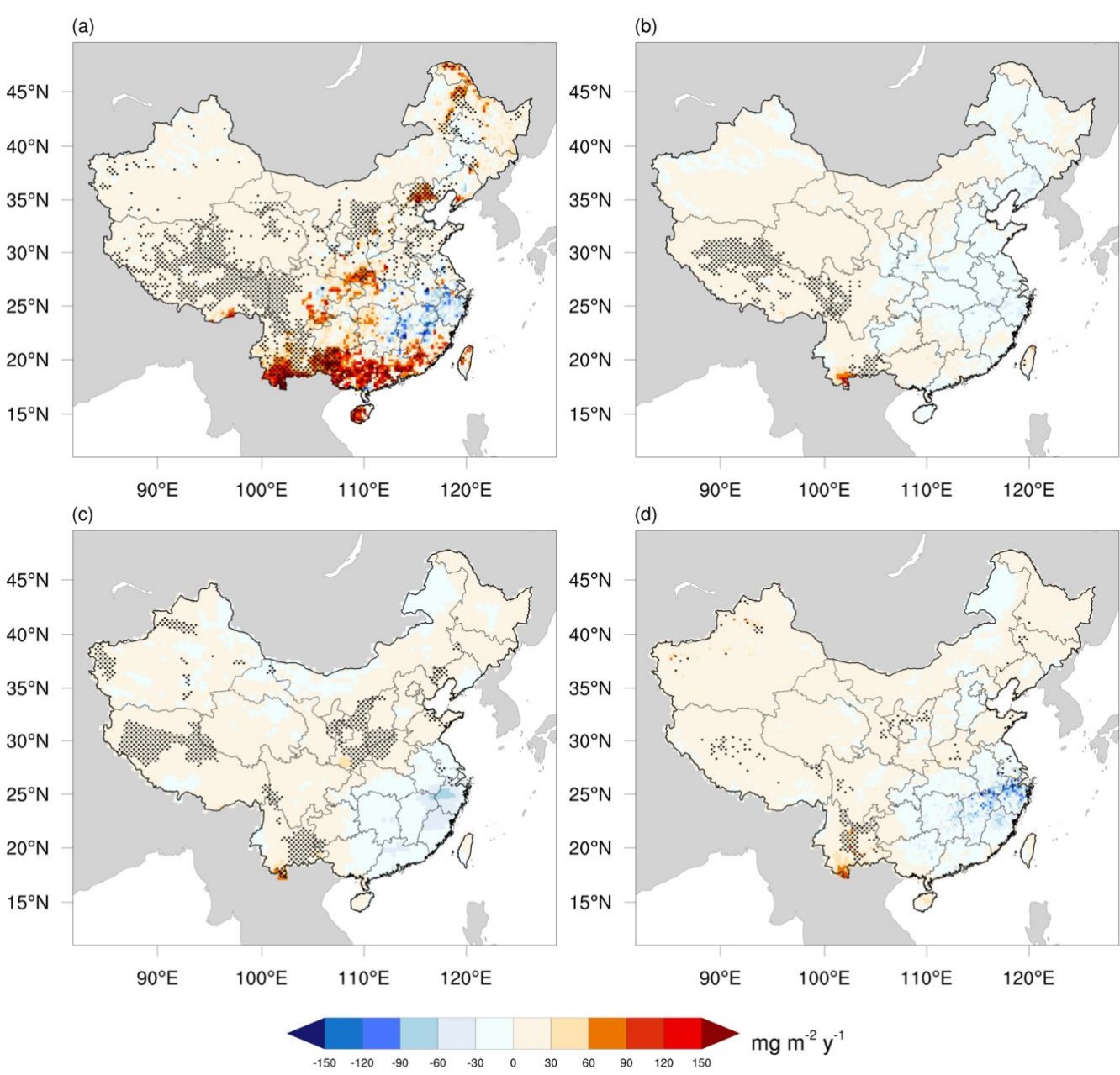

**Figure 10. Comparison of the trend of isoprene emission between this study (S1) and other estimations during 2001-2016. (a) and (b) is for S1 and S3 respectively in this study, and (c) and (d) are for CAMS-GLOB-BIO v 1.1 and CAM-GLOB-BIO v3.1, respectively. The Mann-Kendall test was used to mark the grids where the p is smaller than 0.1.**

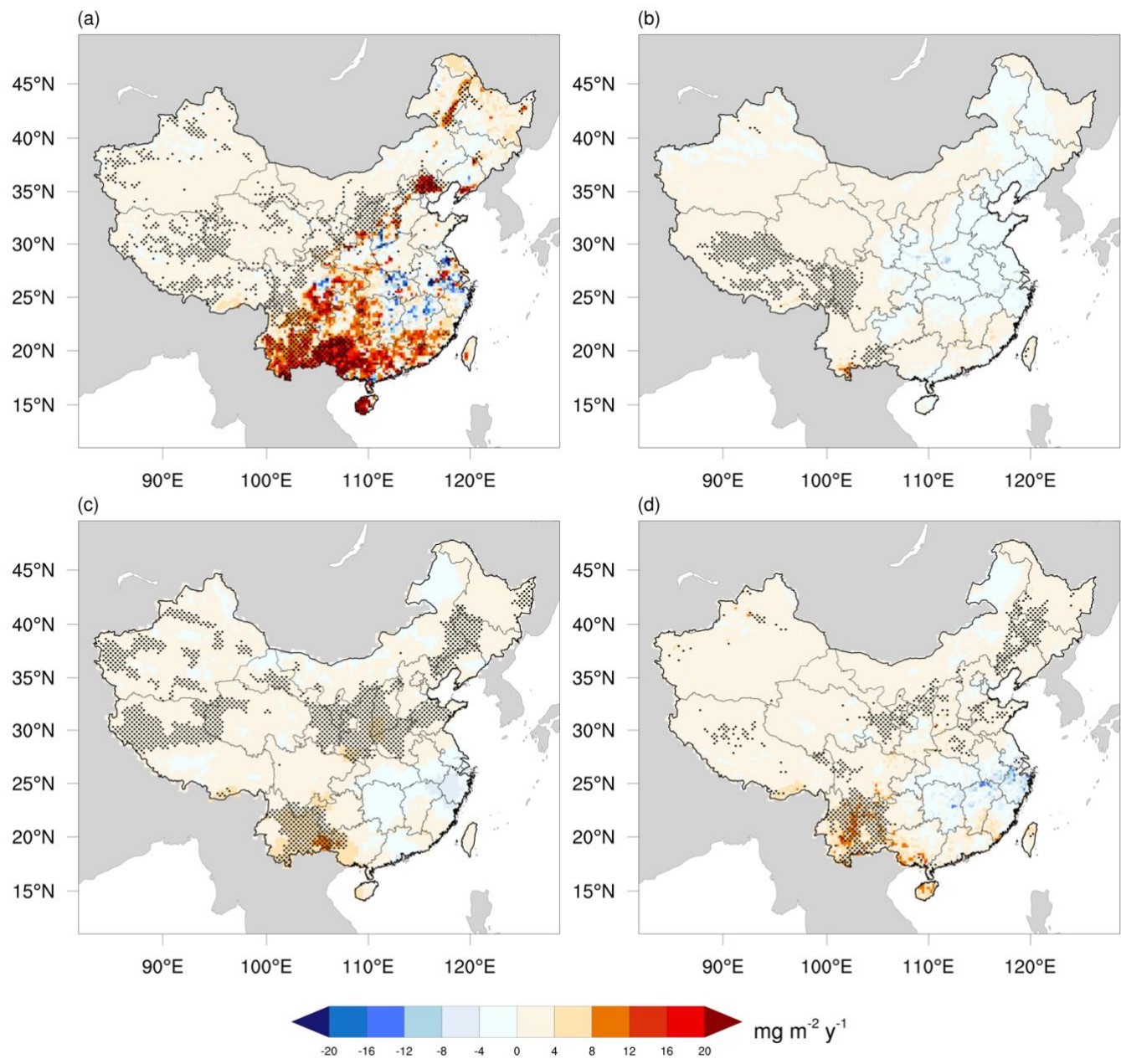

**Figure 11. Comparison of the trend of monoterpenes emission between this study (S1) and other estimations during 2001-2016. (a) and (b) is for S1 and S3, respectively, in this study, and (c) and (d) are for CAMS-GLOB-BIO v 1.1 and CAM-GLOB-BIO v3.1, respectively. The Mann-Kendall test was used to mark the grids where the p is smaller than 0.1.**

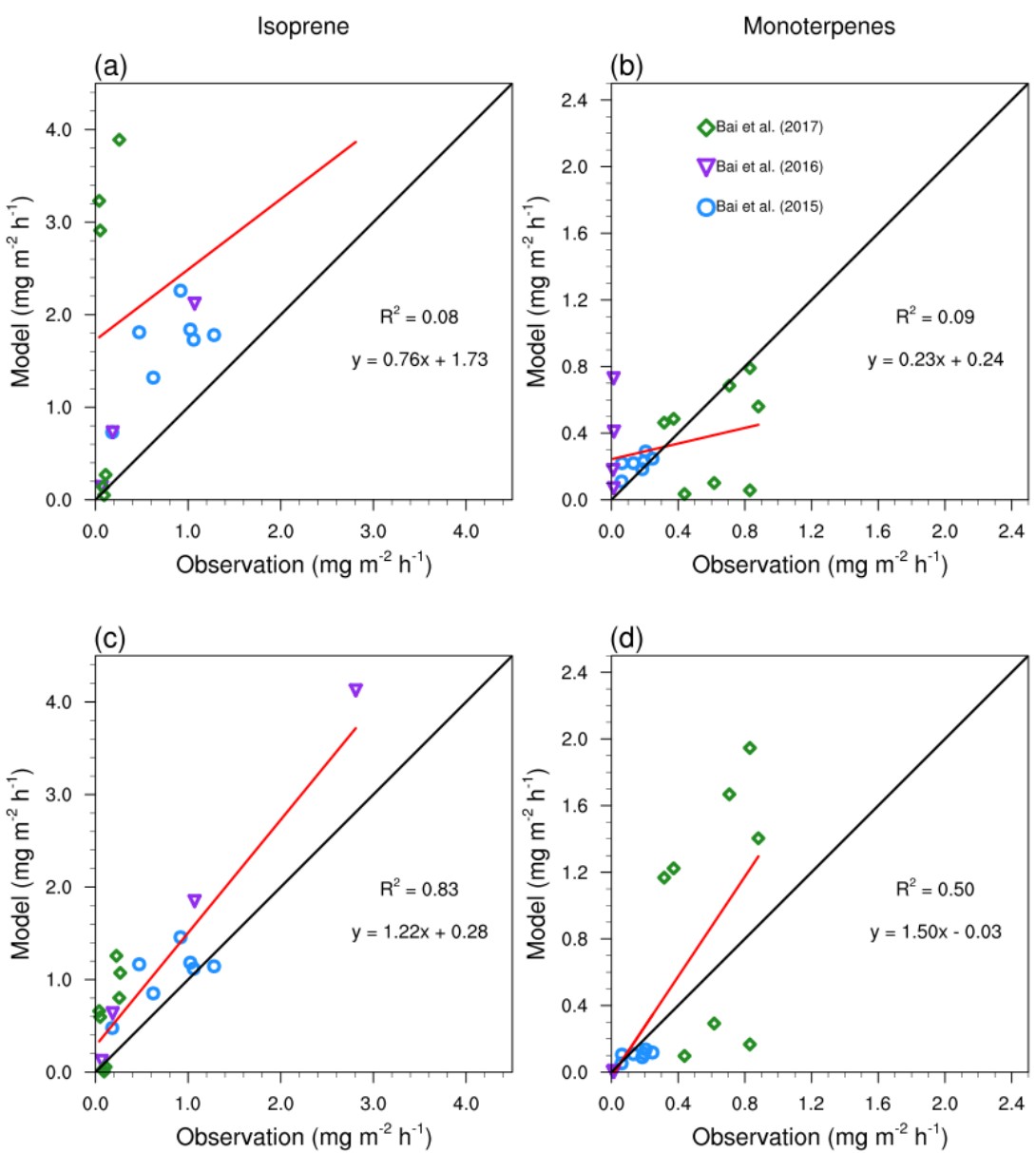

**Figure 12. Validation of the model with flux measurements in China. (a) and (b) show the performance of the MEGAN model with the default emission factors (N=19). (c) and (d) show the performance of the MEGAN model with the emission factors derived from observations (N=19).**