# Peer review of "A long-term estimation of biogenic volatile organic compound (BVOC) emission in China during 2001-2016: the roles of land cover change and climate variability"

_Atmospheric Chemistry and Physics, 2020_

## Referee Comment (RC1) · Anonymous Referee #1 · 14 Apr 2020

General comments: This paper presented the MEGAN-simulated biogenic volatile organic compound (BVOC) emissions in China and analysed the modelled contributions from changes in land cover and climate to the BVOC emissions. The modelled variations in isoprene emissions were further linked to the HCHO vertical column. The paper is well-written and has delivered the message about the potential importance of land cover changes in BVOC emissions in China. The current format of the manuscript has been much focused on analysing the patterns simulated from the four different scenarios, but rather limited in understanding the uncertainties (e.g., uncertainties from satellite products or assigned emission factor or missing PFT) associated with the model simulation. Then when the authors linked their simulated isoprene emission with the

[Figure]

HCHO vertical column, the disagreement of these two has been mainly attributed to the AVOC, but I would think there could be also contributions from the uncertainties in the simulated BVOCs. From the maps with simulated BVOCs, I am a bit surprised to see that the north part of China with high LAI showed very low simulated emissions, especially monoterpene. Could this be linked to the misclassification of forest type? Then in the east and/or at least North China Plain area, there is wide distribution of crops. Are crops specifically considered in MEGAN? In general, a map showing the spatial distribution of PFTs could be very useful for readers. I also think it is crucial to compare the modelled emissions with a few sites' measurement data to illustrate the performance of the model before digging into analysing the changes of the emission patterns at the national scale and further linking to the HCHO column data.

Specific comments:

P2 L5-6, please indicate at which spatial scale we can see cropland dominates the reduction of isoprene.

P2 L10, the authors mentioned that the greening in China has been linked to "maintain and expand forests". Did they change plant species when expanding forest? And can you see this level of land use change in the MODIS PFT product?

P3 L2, suggest to delete "accurately". You have not evaluated the modelled BVOC against the measurements.

P4L2-4, here you might need to specify where these emission factors are from? How much of these emission factors covered the measurements from China? I did a quick google search and could already see some measurement data available for different ecosystems in China. https://www.sciencedirect.com/science/article/pii/S1352231017302947 https://www.sciencedirect.com/science/article/pii/S1352231015305173 https://www.sciencedirect.com/science/article/pii/S0269749119346081?via%3Dihub

P4 L8, "The Cce(=0.57) is a factor to xx" what does this mean?

P4 L9, How can LAI define leaf age in MEGAN?

P4 L13, Is soil moisture used as inputs for model? If so, please clarify.

P4 L17-18, LAI is a 'modelled' product from other satellite products and potentially has large uncertainty in itself. I wonder if the LAI has been filtered by the quality flags before using as inputs for MEGAN and how the model deals with the LAI gap if there is no data for many 8-days? P4 L21-23, Could you list what PFTs you have in your simulations (or showing a map), and also how MODIS PFTs were reclassified to the CLM group? I think this information is important for readers to understand the spatial pattern.

P7 L18-19, the reasons why the simulated MT is so much lower than the previous estimations needs to dig in-depth. Like I mentioned early, could it be linked to the misclassification of PFTs or very different emission factors assigned? In Table 3, the modelled isoprene is very low than Li et al., 2013, can the authors describe a bit about why?

P9 L23, might need to add one or two sentence in the method section why p > 0.9 is statistically significant. I did not get it here.

P12 L11-12, "The lack of long-term in-situ observations of BVOC in China..." I think this might be the case for most of countries where we don't have dataset being representative at the whole country level, but I think the authors should definitely compare the modelled with in-situ data for a few representative sites to evaluate the model performance. In China, there are some sites where you can find the ecosystem-level BVOC measurement data for comparison, like some links I provided in the previous comments.

P12 L12-18, this part should be in the method section.

P13 L5, "... are marked with black dots" it is difficult to see these dots though.

[Figure]

Conclusion, it is rather lengthy at this moment and includes large section of discussion as well. Please make it more concise.

---

## Referee Comment (RC2) · Anonymous Referee #2 · 21 May 2020

The study by Wang and co-workers investigates the impact of satellite-based land use changes on biogenic VOC emissions in China over 16 years (2001-2016). They report positive emission trends of 1-1.5% per year over the whole country, which are attributed, for a major part, to changes in vegetation. The strongest BVOC trends are reported in Qianling mountains and in south China, where the BVOC emissions increased by more than ~60% in 2016 relative to 2001. Further comparison of BVOC interannual variability with HCHO columns from the OMI instrument over the studied period in summertime exhibited positive temporal correlation over forested regions.

This study addresses an interesting subject for Atmospheric Chemistry and Physics

journal. However, there are weaknesses and limitations in the present study, which raise doubts regarding the validity of the conclusions. Furthermore, the presentation is often difficult to follow, mostly due to insufficient mastery of the English language. To my view, the manuscript will need a major revision before it becomes suitable for publication. My main concerns are listed below:

(i) Important input datasets required for calculating BVOC emissions using MEGAN model (e.g. PFTs) are not shown. Annual maps of the MODIS PFTs and LAI should be provided, as well as their trends. Without such information, it is impossible to assess the driving factors for the changes and therefore for the validity of the claims. Furthermore, it is not clearly mentioned whether a unique emission factor per PFT has been used (Table 2 of Guenther et al. 2012) or if a map of standard emission factors has been used.

(ii) I have my doubts regarding the almost negligible isoprene trends due to meteorology suggested by Figure 3 (simulations S3 and S4). The scale in this figure does not allow to see any changes elsewhere than in the Tibetan Plateau. Elsewhere, the color (grey) corresponds to no value. In order to explain the emission trend in S3 and S4, trends of the main drivers of the BVOC emission trends, namely, air temperature, solar radiation and leaf area index should be analysed. In addition, the simulated trend in surface temperature and radiation should be compared to the corresponding trends of the in situ temperature and solar radiation data used for the evaluation of the WRF model simulation in Section 2.3.

(iii) There is not convincing evidence for the very low monoterpene emission derived in this study compared to previous work (Table 3). The invoked reasons, e.g. interannual variations, horizontal resolution, etc. (page 7, lines 81-20) are not convincing. The reasons of the discrepancy should be investigated through detailed comparisons e.g. with the MEGAN inventory and similar studies e.g. Sindelarova et al. (2014). These datasets are accesible via the ECCAD database (https://eccad.aeris-data.fr).

(iv) The strong trends inferred over the Qinling mountains and over Southern China need further discussion. Can you put compare this result to past studies? What is the respective roles played by LAI and PFT cover trends?

Specific comments/Language corrections

p.2, l.3-7: The sentence is too long, considering splitting into two and rephrasing

p.2, l.5: add space between '2014' and 'Chen'

p.2, l.12: 'a corresponding impact', replace by 'changes'

p.3, l.10: remove 'observed'

p.3, l.10: 'regional ecosystem isoprene emission', change to 'isoprene emission at regional to global scales'

p.3, l.11: 'reported the', change to 'reported an'

p.3, l.12: read 'detected by the Ozone'

p.3, l.14-15: rephares as follows: 'Here we used the long-term OMI 2005-2016 record to estimate the interannual isoprene variability in China'

p.3, l.19: add reference Guenther et al.(2012)

p.3, l.20: add more references, e.g. Bauwens et al.(2018) and Messina et al.(2016)

p.3, l.23: read 'uses the fundamental'

p.4, l.1: read 'the standard emisisons factor, and the emission activity factor for the chemical species i'

p.4, l.3: '(PFT) distribution from the Community Land...'

p.4, l.5: replace 'expresses it as' by can be written as'

p.4, l.8: 'equal to 1 at standard conditions (Guenther et al. (2006)'

p.4, l.9: please specify the source of the LAI dataset

p.4, l.9: replace 'and the leaf age in MEGAN' by a new sentence : 'It is used to define the leaf age response function as described in Guenther et al.(2012).'

p.4, l.10: the test should read 'Guenther et al. (1991, 1993, 2012)'

p.4, l.14: remove 'factor'

p.4, l.18: 'adopted', change to 'used'

p.4, l.18: 'in this study', missing reference for the LAI datasets used

p.4, l.20: missing reference for the dataset

p.4, l.22: change 'data' to 'dataset'

p.4, l.22: 'land cover product' missing reference.

p.4, l.24: 'described in'

p.4, l.24: using the climatology of ERA-interim dataset', change to 'using the ERA-Interim climatology'

p.5, l.1: 'during 2001-2016', change to 'over 2001-2016'

p.5, l.5: 'The meteorological simulation is', change to 'The model was'

p.5, l.10: 'using the in-situ', change to 'using in-situ'

p.5, l.13: 'monthly averaged'

p.5, l.15: -2 in Wm-2 should be superscript

p.5, l.15: among 98 sites, and the overestimations', change to 'for 98 studied sites. The overestimation'

p.5, l.17: 'the lack of aerosol radiation effect and cloud simulation', not clear what is meant here

p.5, l.23: 'Our', change to 'The'

p.6, l.1: 'Observations'

p.6, l.3: 'and was retrieved'

p.6, l.4-5: 'The detailed...De Smedt et al. (2015)'. Please remove sentence (repetition)

p.6, l.6: 'temporally stable', what about the row anomaly? This effect should be mentioned.

p.6, l.9: change 'anthropogenic source' to 'anthropogenic VOC'

p.6, l.10: 'in the forest regions without obvious anthropiogenic impact', replace by 'over forests in summertime'

p.6, l.21: 'between 2001 to 2016', change to 'between 2001 and 2016'

p.6, l.25: This has been already mentioned, please avoid repetitions

p.6, l.26-27: sentence not clear

p.7, l.6-8: what do you mean by 'results' and corresponding results'? State clearly what you did

p.7, l.15: 'S1...conditions', repetition

p.7, l.18: 'other estimations', missing references

p.7, l.24: 'increasing rates of these species', replace by 'trends'

p.7, l.25: 'despite the direct impact of meteorological conditions', not clear

p.8, l.11: Rewrite as 'The average annual total BVOC emission over 2009-2016 is by 50% higher than over 2001-2008.' Is that what you mean?

p.8, l.13: 'are by 11.3%'

p.8, l.21: 'S4 is 23.5%', change to 'S4 is by 23.5%

p.8, l.23: 'by 29.9%'

p.8, l.25-26: poor language

p.9, l.15: 'landcover', change to 'land cover'

p.9, l.15: read 'contribute up to 20%, and taken together more than 30% to the estimated...'

p.10, l.6: 'driven', change to 'driven'

p.10, l.10-12: Sentence could be removed

p.10, l.15: superscripts for m-2 y-1

p.10, l.20: read 'broadleaf trees, needleleaf trees and other vegetation'

p.10, l.25: 'percent', replace by 'percentage'

p.10, l.13-24: too many numbers in this paragraph make the reading difficult, consider removing some of the numbers and rewriting

p.11, l.4-7: too many numbers in the text, consider introducing them in a table

p.11, l.11: 'in (Figure 3)', change to 'in Figure 3'

p.11, l.23: 'dominate factor', read 'dominant factor'

p.12, l.2: 'suffering from poor air quality'

p.12, l.2: add space between 'years' and 'Yang'

p.12, 13: 'in rural regions with minimal anthropogenic influence', change to 'over forests'

p.12, l.18: 'summer-average isoprene emission estimated in our study to evaluate our estimation of interannual variability of isoprene emission', poor wording

p.13, l.1: 'anthropogenic sources', missing reference

p.13, l.5: 'correlation can be found', change to 'correlation is found'

p.13, l.10: 'anthropogenic sources', missing reference

p.13, l.20: 'greatest increasing trend', change to 'strongest positive trend'

p.14, l.4: 'the mega-city areas', read 'in megacities'

p.15, l.1: read 'from 2001 to 2016'

p.15, l.1: read 'as inputs in the MEGAN'

p.15, l.1: 'the long-term', remove 'the'

p.15, l.11: here and elsewhere in the manuscript, use one instead of two decimals

p.15, l.18: 'there'?

p.15, l.21: 'during 200-2010', missing reference

p.15, l.22: 'there has been in a increasing trend', do you mean 'showed an increasing trend'?

p.15, l.24: read 'assess'

p.16, l.6: remove the references (they are already mentioned before)

p.16, l.6-10: repetition of l.20-25 of page 14, not necessary

p.23: Table 3, the estimates reported in Li et al. are in TgC, not in Tg, please correct

p.26: Difficult to read, I suggest splitting into a figure with 4 panels (a, f, k, p) and another figure with the trends. The regions in panel (r) are barely visible. Plese improve.

p.27: It is very difficult to distinguish the colors corresponding to broadleaf and needleleaf trees, please adapt. In the caption, please correct typos for the names of provinces.

References:

Bauwens, M., Stavrakou, T., Müller, J.-F., Van Schaeybroeck, B., De Cruz, L., De Troch, R., Giot, O., Hamdi, R., Termonia, P., Laffineur, Q., Amelynck, C., Schoon, N., Heinesch, B., Holst, T., Arneth, A., Ceulemans, R., Sanchez-Lorenzo, A., and Guenther, A.: Recent past (1979-2014) and future (2070-2099) isoprene fluxes over Europe simulated with the MEGAN-MOHYCAN model, Biogeosciences, 15, 3673-3690, https://doi.org/10.5194/bg-15-3673-2018, 2018.

Messina, P., Lathière, J., Sindelarova, K., Vuichard, N., Granier, C., Ghattas, J., Cozic, A., and Hauglustaine, D. A.: Global biogenic volatile organic compound emissions in the ORCHIDEE and MEGAN models and sensitivity to key parameters, Atmos. Chem. Phys., 16, 14169-14202, https://doi.org/10.5194/acp-16-14169-2016, 2016.

---

## Author Comment (AC1) · 6 Jul 2020

**Response to Referee #1**

General comments: This paper presented the MEGAN-simulated biogenic volatile organic compound (BVOC) emissions in China and analysed the modelled contributions from changes in land cover and climate to the BVOC emissions. The modelled variations in isoprene emissions were further linked to the HCHO vertical column. The paper is well-written and has delivered the message about the potential importance of land cover changes in BVOC emissions in China. The current format of the manuscript has been much focused on analysing the patterns simulated from the four different scenarios, but rather limited in understanding the uncertainties (e.g., uncertainties from satellite products or assigned emission factor or missing PFT) associated with the model simulation. Then when the authors linked their simulated isoprene emission with the HCHO vertical column, the disagreement of these two has been mainly attributed to the AVOC, but I would think there could be also contributions from the uncertainties in the simulated BVOCs. From the maps with simulated BVOCs, I am a bit surprised to see that the north part of China with high LAI showed very low simulated emissions, especially monoterpene. Could this be linked to the misclassification of forest type? Then in the east and/or at least North China Plain area, there is wide distribution of crops. Are crops specifically considered in MEGAN? In general, a map showing the spatial distribution of PFTs could be very useful for readers. I also think it is crucial to compare the modelled emissions with a few sites' measurement data to illustrate the performance of the model before digging into analysing the changes of the emission patterns at the national scale and further linking to the HCHO column data.

Response: Thank you so much for your precious time and your comments. We will follow your suggestion and add more discussion about the uncertainties by comparing with other studies and adding extra experiments. In addition, some necessary information, e.g. PFT spatial distribution, will also be presented in the revised paper. We will also try to find some in-situ measurements from previous literatures to validate the model. Currently, we haven't finished the revision of paper, so we will only response the discussion of ACP and introduce our direction of revision. The final response will be submitted with the revised paper.

Specific comments:

P2 L5-6, please indicate at which spatial scale we can see cropland dominates the reduction of isoprene.

Response: Thank you for your comments. We will add this part in the revised paper.

P2 L10, the authors mentioned that the greening in China has been linked to "maintain and expand forests". Did they change plant species when expanding forest? And can you see this level of land use change in the MODIS PFT product?

Response: Thank you for your comments. Currently, distinguish the specific species of trees using MODIS since the spatial resolution of MODIS sensor is not high enough to do so. So, we can't see the species-level change through the MODIS PFT. Our estimation is mainly based on the PFT level change.

P3 L2, suggest to delete "accurately". You have not evaluated the modelled BVOC against the measurements.

Response: Thank you so much for your advice, and the word "accurately" will be deleted in the revised paper.

P4L2-4, here you might need to specify where these emission factors are from? How much of these emission factors covered the measurements from China? I did a quick google search and could already see some measurement data available for different ecosystems in China.
https://www.sciencedirect.com/science/article/pii/S1352231017302947
https://www.sciencedirect.com/science/article/pii/S1352231015305173
https://www.sciencedirect.com/science/article/pii/S0269749119346081?via%3Dihub

Response: Thank you so much for your comments. The emission factors in this study are the default values of the MEGAN 2.1 provided by Guenther et al. 2012. Since we didn't have an ability to distinguish the species of the trees using the MODIS images, we didn't consider using the species-based emission factors. It is true that this will induce the uncertainty of emission amount, and we will add some discussion for this in the revised paper. However, this work is focusing on the impact of land cover change and vegetation biomass change on BVOC emission in China, so using the default emission factor could help to discuss the change of BVOC.

P4 L8, "The Cce(=0.57) is a factor to xx" what does this mean?

Response: As described by Guenther et al. (2006), the $C_{ce}$ is a parameter in MEGAN model that sets the emission factor to unity at the standard conditions. It has no physical meaning and was used to normalize the emission factors. We will state it more clearly in the revised paper.

P4 L9, How can LAI define leaf age in MEGAN?

Response: Thank you so much for your comments. The leaf-age factor, $\gamma_{age}$, in MEGAN is described in detail in Guenther et al. (2006). For the evergreen canopies, $\gamma_{age}$ is constant. For the deciduous canopies, the leaves are divided into four stages of new leaf, growing leaf, mature leaf and old leaf since the emission capacity of leaf is diverse with leaf age(Guenther et al., 1991;Monson et al., 1994;Guenther et al., 2006). According to Guenther et al. (2006), the $\gamma_{age}$ is defined as:

$$\gamma_{age} = F_{new}A_{new} + F_{gro}A_{gro} + F_{mat}A_{mat} + F_{old}A_{old}$$

where $A_{new}$, $A_{gro}$, $A_{mat}$ and $A_{old}$ are the relative emission rates for new, growing, mature and old foliages. $F_{new}$, $F_{gro}$, $F_{mat}$ and $F_{old}$ are the fractions of different sorts of leaves and are defined by the change of LAI between the current time step (LAIc) and the previous time step (LAIp). $F_{new}$=0, $F_{gro}$=0.1, $F_{mat}$=0.8 and $F_{old}$=0.1 when LAIc equals LAIp. When LAIp> LAIc, the fractions in different stages are as:

$$\begin{cases} F_{new} = 0 \\ F_{gro} = 0 \\ F_{old} = [(LAIp - LAIc)/LAIp] \\ F_{old} = 1 - F_{old} \end{cases}$$

In the cases of LAIp<LAIc, the fractions are calculated as:

$$\begin{cases} F_{new} = 1 - (LAIp/LAIc) \\ F_{gro} = 1 - F_{new} - F_{mat} \\ F_{mat} = LAIp/LAIc \\ F_{old} = 0 \end{cases}$$

. More details can be found in Guenther et al. (2006), and we will add some brief introduction into the revised paper.

P4 L13, Is soil moisture used as inputs for model? If so, please clarify.

Response: Thanks for your comments. The soil moisture is simulated by the WRF model and will be considered in the calculation. We will follow your comments and clarify this part in the revised paper.

P4 L17-18, LAI is a 'modelled' product from other satellite products and potentially has large uncertainty in itself. I wonder if the LAI has been filtered by the quality flags before using as inputs for MEGAN and how the model deals with the LAI gap if there is no data for many 8-days?

Response: Thank you so much for your comments. We used all available values in MODIS LAI products, and we didn't use the quality filter at the first place to ensure the model can be driven by continued LAI field. The model didn't have ability to deal with the LAI gap, but this problem can be solved by using some interpolation technics when preparing the inputs.

P4 L21-23, Could you list what PFTs you have in your simulations (or showing a map), and also how MODIS PFTs were reclassified to the CLM group? I think this information is important for readers to understand the spatial pattern.

Response: Thank you so much for your comments. We will provide the spatial distribution of PFT as well as the method used to convert MODIS PFT to CLM group in this part. Also, we will do some sensitivity tests to discuss the impact of PFT classification system.

P7 L18-19, the reasons why the simulated MT is so much lower than the previous estimations needs to dig in-depth. Like I mentioned early, could it be linked to the misclassification of PFTs or very different emission factors assigned? In Table 3, the modelled isoprene is very low than Li et al., 2013, can the authors describe a bit about why?

Response: Thank so much for your comments. We agreed with your comments. The low estimated monoterpene emission may be induced by the misclassification of PFTs or emission

factors. We will double check the PFT classification system and add some sensitivity tests for further discussion. The emission factors used in this study are the default values in Guenther et al., 2012, which may be different from that in Li et al., 2013. We will also do some comparisons to explain the discrepancy between two studies.

P9 L23, might need to add one or two sentences in the method section why p > 0.9 is statistically significant. I did not get it here.

Response: Thank you so much for comments. Some corresponding explanation will be given in the revised paper.

P12 L11-12, "The lack of long-term in-situ observations of BVOC in China..." I think this might be the case for most of countries where we don't have dataset being representative at the whole country level, but I think the authors should definitely compare the modelled with in-situ data for a few representative sites to evaluate the model performance. In China, there are some sites where you can find the ecosystem-level BVOC measurement data for comparison, like some links I provided in the previous comments.

Response: Thank you so much for comments. The rare observations limit the validation of MEGAN model in this study. Furthermore, the outputs of MEGAN for BVOCs are canopy level fluxes not concentrations. Although there are some BVOC concentration observations in China, MEGAN model cannot be validated though these observations. But we will try our best to find some ecosystem-level BVOC measurements from the previous literatures to see if there's a chance to validate the model results.

P12 L12-18, this part should be in the method section.

Response: Thanks for your comments. We will find a better way to do that in the revised paper.

P13 L5, ". . . are marked with black dots" it is difficult to see these dots though.

Response: Thanks for your comments. We will find a better way to present the information in the revised paper.

Conclusion, it is rather lengthy at this moment and includes large section of discussion as well. Please make it more concise.

Response: Thanks for your comments. We will try to compress our current manuscript by removing unnecessary information and descriptions to make current contents relatively concise. On the other hand, we will also add more results to investigate the issues like PFT classification and discuss more uncertainties in this study.

**Reference**

Guenther, A., Karl, T., Harley, P., Wiedinmyer, C., Palmer, P., and Geron, C.: Estimates of global terrestrial isoprene emissions using MEGAN (Model of Emissions of Gases and Aerosols from Nature), Atmos. Chem. Phys, 6, 3181-3210, 2006.

Guenther, A. B., Monson, R. K., and Fall, R.: Isoprene and monoterpene emission rate variability: Observations with eucalyptus and emission rate algorithm development, Journal of Geophysical Research: Atmospheres, 96, 10799-10808, 10.1029/91JD00960, 1991.

Monson, R. K., Harley, P. C., Litvak, M. E., Wildermuth, M., Guenther, A. B., Zimmerman, P. R., and Fall, R.: Environmental and developmental controls over the seasonal pattern of isoprene emission from aspen leaves, Oecologia, 99, 260-270, 10.1007/BF00627738, 1994.

---

## Author Comment (AC2) · 7 Jul 2020

**Response to Referee #2**

The study by Wang and co-workers investigates the impact of satellite-based land use changes on biogenic VOC emissions in China over 16 years (2001-2016). They report positive emission trends of 1-1.5% per year over the whole country, which are attributed, for a major part, to changes in vegetation. The strongest BVOC trends are reported in Qinling mountains and in south China, where the BVOC emissions increased by more than ~60% in 2016 relative to 2001. Further comparison of BVOC interannual variability with HCHO columns from the OMI instrument over the studied period in summertime exhibited positive temporal correlation over forested regions. This study addresses an interesting subject for Atmospheric Chemistry and Physics journal. However, there are weaknesses and limitations in the present study, which raise doubts regarding the validity of the conclusions. Furthermore, the presentation is often difficult to follow, mostly due to insufficient mastery of the English language. To my view, the manuscript will need a major revision before it becomes suitable for publication. My main concerns are listed below:

Response: Thank you so much for your precious time and we really appreciate your comments. Since we haven't finished the revision of paper, right now we will only response the discussion of ACP and introduce our direction of revision. The final response will be submitted with the revised paper. We will try to address the problems you mentioned about uncertainty of this study by taking following measures:

1. We will further analyze the role of meteorology on the trend of BVOC emission.

2. Extra experiments will be added to illustrate the contribution of LAI on trends of BVOC emission. In addition, we will conduct some experiment to discuss the uncertainty induced by the PFT product and its classification system.

3. We will compare our estimations with the results by other studies to discuss the potential reasons of discrepancy between them.

4. We will invite some native speakers to improve the language of the revised paper.

(i) Important input datasets required for calculating BVOC emissions using MEGAN model (e.g. PFTs) are not shown. Annual maps of the MODIS PFTs and LAI should be provided, as well as their trends. Without such information, it is impossible to assess the driving factors for the changes and therefore for the validity of the claims. Furthermore, it is not clearly mentioned whether a unique emission factor per PFT has been used (Table 2 of Guenther et al. 2012) or if a map of standard emission factors has been used.

Response: Thank you so much for your comments. We will present the annual change of MODIS PFTs and LAI and their trends in the revised paper or the supplement. The emission factor is coming from the Table 2 of Guenther et al. 2012, and we will also mention that in the revised paper.

(ii) I have my doubts regarding the almost negligible isoprene trends due to meteorology suggested by Figure 3 (simulations S3 and S4). The scale in this figure does not allow to see any changes elsewhere than in the Tibetan Plateau. Elsewhere, the color (grey) corresponds to no value. In order to explain the emission trend in S3 and S4, trends of the main drivers of the BVOC emission trends, namely, air temperature, solar radiation and leaf area index should be analysed. In addition, the simulated trend in surface temperature and radiation should be compared to the corresponding trends of the in situ temperature and solar radiation data used for the evaluation of the WRF model

simulation in Section 2.3.

Response: Thank you so much for your comments. We will add more analysis regarding the meteorological impact on BVOC emission as you suggested. We will also analyze the trend of meteorological fields of WRF simulation and in-situ observation separately as you suggested.

(iii) There is not convincing evidence for the very low monoterpene emission derived in this study compared to previous work (Table 3). The invoked reasons, e.g. interannual variations, horizontal resolution, etc. (page 7, lines 81-20) are not convincing. The reasons of the discrepancy should be investigated through detailed comparisons e.g. with the MEGAN inventory and similar studies e.g. Sindelarova et al. (2014). These datasets are accessible via the ECCAD database (https://eccad.aeris-data.fr).

Response: Thank you so much for your comments. We agreed to your suggestion and will compare our results with other datasets of BVOC emission from ECCAD database. Another reviewer also mentioned it and asserted that the misclassification of PFTs of MODIS may be the potential reason of low monoterpene. Therefore, we will also do some experiments to investigate the impact of different land cover classification system of MODIS products.

(iv) The strong trends inferred over the Qinling mountains and over Southern China need further discussion. Can you put compare this result to past studies? What is the respective roles played by LAI and PFT cover trends?

Response: Thank you for your comments. We will compare our results with other studies to further investigate the strong trend over the Qinling Mountains. In addition, extra experiments will also be added to investigate the role played by LAI and PFT, respectively.

Specific comments/Language corrections
p.2, l.3-7: The sentence is too long, considering splitting into two and rephrasing.
Response: Thank you for your comments. This sentence has been re-written as:
"Besides the climatic factors, the land cover change also plays a key role in the variability of BVOC emission (Stavrakou et al., 2014; Unger, 201; Chen et al., 2018). For instance, cropland expansion has been estimated to dominate the reduction of isoprene, the dominant BVOC species, in last century (Lathière et al., 2010; Unger, 2013) although there are large uncertainties associated with these estimates."

p.2, l.5: add space between '2014' and 'Chen'
Response: Thank you. We have followed your advice.

p.2, l.12: 'a corresponding impact', replace by 'changes'
Response: Thank you. We have followed your advice.

p.3, l.10: remove 'observed'
Response: Thank you. We have followed your comments.

p.3, l.10: 'regional ecosystem isoprene emission', change to 'isoprene emission at regional to global scales'

Response: Thank you. We have followed your advice.

p.3, l.11: 'reported the', change to 'reported an'
Response: Thank you. We have followed your advice.

p.3, l.12: read 'detected by the Ozone'
Response: Thank you. We have followed your advice.

p.3, l.14-15: rephrases as follows: 'Here we used the long-term OMI 2005-2016 record to estimate the interannual isoprene variability in China'
Response: Thank you. We have followed your advice.

p.3, l.19: add reference Guenther et al.(2012)
Response: Thank you. We have followed your advice.

p.3, l.20: add more references, e.g. Bauwens et al.(2018) and Messina et al.(2016)
Response: Thank you. We have added these references.

p.3, l.23: read 'uses the fundamental'
Response: Thank you. We have followed your advice.

p.4, l.1: read 'the standard emissions factor, and the emission activity factor for the chemical species i'
Response: Thank you. We have followed your advice.

p.4, l.3: '(PFT) distribution from the Community Land...'
Response: Thank you. We have followed your advice.

p.4, l.5: replace 'expresses it as' by 'can be written as'
Response: Thank you. We have followed your advice.

p.4, l.8: 'equal to 1 at standard conditions (Guenther et al. (2006)'
Response: Thank you. We have followed your advice.

p.4, l.9: please specify the source of the LAI dataset
Response: Thank you. We have added the link of website of MODIS LAI products (https://lpdaac.usgs.gov/products/mcd15a2hv006/) in this sentence.

p.4, l.9: replace 'and the leaf age in MEGAN' by a new sentence: 'It is used to define the leaf age response function as described in Guenther et al.(2012).'
Response: Thank you. We have followed your advice.

p.4, l.10: the test should read 'Guenther et al. (1991, 1993, 2012)'
Response: Thank you. We have followed your advice.

p.4, l.14: remove 'factor'
Response: Thank you. We have followed your advice.

p.4, l.18: 'adopted', change to 'used'
Response: Thank you so much. We have followed your advice.

p.4, l.18: 'in this study', missing reference for the LAI datasets used
Response: Thank you so much. We have followed your advice and added the reference of MODIS LAI product.

p.4, l.20: missing reference for the dataset
Response: Thank you so much. We have followed your advice and added the reference of MODIS VCF product.

p.4, l.22: change 'data' to 'dataset'
Response: Thank you. We have followed your advice.

p.4, l.22: 'land cover product' missing reference.
Response: Thank you so much. We have followed your advice and added the reference of MODIS land cover product.

p.4, l.24: 'described in'
Response: Thank you. We have followed your advice.

p.4, l.24: using the climatology of ERA-interim dataset', change to 'using the ERAInterim climatology'
Response: Thank you. We have followed your advice.

p.5, l.1: 'during 2001-2016', change to 'over 2001-2016'
Response: Thank you. We have followed your advice.

p.5, l.5: 'The meteorological simulation is', change to 'The model was'
Response: Thank you. We have followed your advice.

p.5, l.10: 'using the in-situ', change to 'using in-situ'
Response: Thank you. We have followed your advice.

p.5, l.13: 'monthly averaged'
Response: Thank you. We have followed your advice.

p.5, l.15: -2 in Wm-2 should be superscript
Response: Thank you. We have followed your advice.

p.5, l.15: among 98 sites, and the overestimations', change to 'for 98 studied sites. The

overestimation'

Response: Thank you. We have followed your advice.

p.5, l.17: 'the lack of aerosol radiation effect and cloud simulation', not clear what is meant here

Response: Thank you so much for your comments. We have modified this sentence as:

"The overestimation of DSW simulation is a common issue in multiple simulation studies and may be induced by the lack of physical processes for aerosol radiation effect (Wang et al., 2011; Situ et al., 2013; Wang et al., 2018)."

p.5, l.23: 'Our', change to 'The'

Response: Thank you. We have followed your advice.

p.6, l.1: 'Observations'

Response: Thank you. We have followed your advice.

p.6, l.3: 'and was retrieved'

Response: Thank you. We have followed your advice.

p.6, l.4-5: 'The detailed...De Smedt et al. (2015)'. Please remove sentence (repetition)

Response: Thank you. We have followed your advice.

p.6, l.6: 'temporally stable', what about the row anomaly? This effect should be mentioned.

Response: Thank you. We will follow your advice and add the discussion about the row anomaly of OMI.

p.6, l.9: change 'anthropogenic source' to 'anthropogenic VOC'

Response: Thank you. We have followed your advice.

p.6, l.10: 'in the forest regions without obvious anthropogenic impact', replace by 'over forests in summertime'

Response: Thank you. We have followed your advice.

p.6, l.21: 'between 2001 to 2016', change to 'between 2001 and 2016'

Response: Thank you. We have followed your advice.

p.6, l.25: This has been already mentioned, please avoid repetitions

Response: Thank you. We have followed your advice and removed that sentence.

p.6, l.26-27: sentence not clear

Response: Thank you. We will rephrase or extend this sentence to let it clearer.

p.7, l.6-8: what do you mean by 'results' and corresponding results'? State clearly what you did

Response: Thank you. We have followed your advice.

p.7, l.15: 'S1...conditions', repetition
Response: Thank you. We have followed your advice and removed the sentence.

p.7, l.18: 'other estimations', missing references
Response: Thank you. We have followed your advice and added the reference in Table 3 to the sentence.

p.7, l.24: 'increasing rates of these species', replace by 'trends'
Response: Thank you. We have followed your advice.

p.7, l.25: 'despite the direct impact of meteorological conditions', not clear
Response: Thank you so much. We will rewrite this sentence in the revised paper.

p.8, l.11: Rewrite as 'The average annual total BVOC emission over 2009-2016 is by 50% higher than over 2001-2008.' Is that what you mean?
Response: Thank you so much for your comments. That's what we mean, and we have rewritten this sentence following your suggestion.

p.8, l.13: 'are by 11.3%'
Response: Thank you. We have followed your advice.

p.8, l.21: 'S4 is 23.5%', change to 'S4 is by 23.5%
Response: Thank you. We have followed your advice.

p.8, l.23: 'by 29.9%'
Response: Thank you. We have followed your advice.

p.8, l.25-26: poor language
Response: Thank you. We will rephrase this sentence.

p.9, l.15: 'landcover', change to 'land cover'
Response: Thank you. We have followed your advice.

p.9, l.15: read 'contribute up to 20%, and taken together more than 30% to the estimated...'
Response: Thank you. We have followed your advice.

p.10, l.6: 'driven', change to 'driven'
Response: Thank you. We have followed your advice.

p.10, l.10-12: Sentence could be removed
Response: Thank you. We have followed your advice.

p.10, l.15: superscripts for m-2 y-1
Response: Thank you. We have followed your advice.

p.10, l.20: read 'broadleaf trees, needleleaf trees and other vegetation'
Response: Thank you. We have followed your advice.

p.10, l.25: 'percent', replace by 'percentage'
Response: Thank you. We have followed your advice.

p.10, l.13-24: too many numbers in this paragraph make the reading difficult, consider removing some of the numbers and rewriting
Response: Thank you so much for your suggestion. We will rephrase this paragraph.

p.11, l.4-7: too many numbers in the text, consider introducing them in a table
Response: Thank you. We will consider your advice and add a suitable table or graph.

p.11, l.11: 'in (Figure 3)', change to 'in Figure 3'
Response: Thank you. We have followed your advice.

p.11, l.23: 'dominate factor', read 'dominant factor'
Response: Thank you. We have followed your advice.

p.12, l.2: 'suffering from poor air quality'
Response: Thank you. We have followed your advice.

p.12, l.2: add space between 'years' and 'Yang'
Response: Thank you. We have followed your advice.

p.12, 13: 'in rural regions with minimal anthropogenic influence', change to 'over forests'
Response: Thank you. We have followed your advice.

p.12, l.18: 'summer-average isoprene emission estimated in our study to evaluate our estimation of interannual variability of isoprene emission', poor wording.
Response: Thank you. We will rephrase this paragraph.

p.13, l.1: 'anthropogenic sources', missing reference
Response: Thank you. We will add the reference in the revised paper.

p.13, l.5: 'correlation can be found', change to 'correlation is found'
Response: Thank you. We have followed your advice.

p.13, l.10: 'anthropogenic sources', missing reference
Response: Thank you. We will add the reference in the revised paper.

p.13, l.20: 'greatest increasing trend', change to 'strongest positive trend'
Response: Thank you. We have followed your advice.

p.14, l.4: 'the mega-city areas', read 'in megacities'
Response: Thank you. We have followed your advice.

p.15, l.1: read 'from 2001 to 2016'
Response: Thank you. We have followed your advice.

p.15, l.1: read 'as inputs in the MEGAN'
Response: Thank you. We have followed your advice.

p.15, l.1: 'the long-term', remove 'the'
Response: Thank you. We have followed your advice.

p.15, l.11: here and elsewhere in the manuscript, use one instead of two decimals
Response: Thank you. We have followed your advice.

p.15, l.18: 'there'?
Response: Thank you so much for your comments. We will rephrase this sentence.

p.15, l.21: 'during 200-2010', missing reference
Response: Thank you. We will add the reference in the revised paper.

p.15, l.22: 'there has been in a increasing trend', do you mean 'showed an increasing trend'?
Response: Thank you for your comments. We have revised this sentence as your suggestion.

p.15, l.24: read 'assess'
Response: Thank you. We have followed your advice.

p.16, l.6: remove the references (they are already mentioned before)
Response: Thank you. We have followed your advice.

p.16, l.6-10: repetition of l.20-25 of page 14, not necessary
Response: Thank you. We have followed your advice and removed this sentence.

p.23: Table 3, the estimates reported in Li et al. are in TgC, not in Tg, please correct
Response: Thank you so much. We have corrected this in the revised paper.

p.26: Difficult to read, I suggest splitting into a figure with 4 panels (a, f, k, p) and another figure with the trends. The regions in panel (r) are barely visible. Please improve.
Response: Thank you. We will follow your suggestion and find a better way to present the results.

p.27: It is very difficult to distinguish the colors corresponding to broadleaf and needleleaf trees, please adapt. In the caption, please correct typos for the names of provinces.
Response: Thank you. We will follow your suggestion and improve the figure as well as caption.

---

## Author Response (AR1)

**Response to Editor:**

Dear editor,

We really appreciate the efforts you made for improving the quality of our manuscript and your patience for giving us enough time to revise our manuscript during this extremely hard time. We tried our best to revise our manuscript according to the comments from two anonymous reviewers. The following major changes were made in our revised paper:

1. We found some mistakes in our program for mapping the MODIS classification system to CLM PFTs, which will lead to missing or double counting some PFT categories during the mapping process. Therefore, we corrected the program and re-ran all experiments. In addition, we used the IGBP classification scheme this time instead of using the Leaf Area Index Classification Scheme in MCD12C1 product as the original classification scheme for mapping considering the more detailed descriptions of legends in IGBP scheme. Some conclusions were also corrected based on the new results.

2. We added one more experiment named S5 to illustrate the contribution of LAI on trends of BVOC emission. In S5, we used the annually updated LAIv and the fixed meteorological inputs and PFT dataset for the year 2001. The analysis for S5 was already added into the revised paper.

3. We further compared our results with other studies to discuss the uncertainties of our estimation. We downloaded some long-term BVOC estimations from ECCAD database (https://eccad.aeris-data.fr) and compared them with our results to analyses the potential reason that results in the discrepancies between our results and other estimations. In addition, we collected the flux measurements from some recent studies (Bai et al., 2015;Bai et al., 2016;Bai et al., 2017) to validate our model and discuss the uncertainties induced by emission factor. Corresponding content has been added into the revised paper.

4. We removed the Section 3.5 in the previous version paper about "Comparison of BVOC emission with Anthropogenic Emission China". Considering the uncertainties behind our estimations, we decided to concentrate on BVOC emission estimation and discuss more about uncertainties instead of extending to discuss anthropogenic emissions. Some lengthy and less informative paragraphs are also removed in the revised paper.

The point-by-point responses to two reviewers' comments are given below.

**Response to Referee #1**

General comments: This paper presented the MEGAN-simulated biogenic volatile organic compound (BVOC) emissions in China and analysed the modelled contributions from changes in land cover and climate to the BVOC emissions. The modelled variations in isoprene emissions were further linked to the HCHO vertical column. The paper is well-written and has delivered the message about the potential importance of land cover changes in BVOC emissions in China.

**Response:** Thank you so much for your comments, and we really appreciate it. In the revised paper, we

did the following measurements to address your concerns as well as the other reviewer's concerns:

1. We found some mistakes in our program for mapping the MODIS classification system to CLM PFTs, which will lead to missing or double counting some PFT categories during the mapping process. Therefore, we corrected the program and re-ran all experiments. In addition, we used the IGBP classification scheme this time instead of using the Leaf Area Index Classification Scheme in MCD12C1 product as the original classification scheme for mapping considering the more detailed descriptions of legends in IGBP scheme. Some conclusions were also corrected based on the new results.

2. We added one more experiment named S5 to illustrate the contribution of LAI on trends of BVOC emission. In S5, we used the annually updated LAIv and the fixed meteorological inputs and PFT dataset for the year 2001. The analysis for S5 was already added into the revised paper.

3. We further compared our results with other studies to discuss the uncertainties of our estimation. We downloaded some long-term BVOC estimations from ECCAD database (https://eccad.aeris-data.fr) and compared them with our results to analyses the potential reason that results in the discrepancies between our results and other estimations. In addition, we collected the flux measurements from some recent studies (Bai et al., 2015;Bai et al., 2016;Bai et al., 2017) to validate our model and discuss the uncertainties induced by emission factor. Corresponding content has been added into the revised paper.

4. We removed the Section 3.5 in the previous version paper about "Comparison of BVOC emission with Anthropogenic Emission China". Considering the uncertainties behind our estimations, we decided to concentrate on BVOC emission estimation and discuss more about uncertainties instead of extending to discuss anthropogenic emissions. Some lengthy and less informative paragraphs are also removed in the revised paper.

The current format of the manuscript has been much focused on analysing the patterns simulated from the four different scenarios, but rather limited in understanding the uncertainties (e.g., uncertainties from satellite products or assigned emission factor or missing PFT) associated with the model simulation.

**Response:** Thank you so much for your comments. We double-checked the program for mapping the MODIS PFT to CLM PFT classification, and we found some mistakes in the program that led to missing or double counting some PFTs during the mapping process. Therefore, we corrected the program and re-ran the all experiments. In addition, we used the IGBP classification scheme this time instead of using the Leaf Area Index Classification Scheme in MCD12C1 product as the original classification scheme for mapping considering the more detailed descriptions of legends in IGBP scheme. So, we redesigned the mapping method.

The mapping method is in two steps. As presented in Table R1, we firstly mapped the IGBP classification to eight main vegetation categories: needleleaf evergreen forests, broadleaf evergreen forests, needleleaf deciduous forests, broadleaf deciduous forests, mixed forests, shrub, grass and crop according to the description of the legends. Then, eight main categories were mapped to the

classification of CLM/MEGAN for boreal, temperate and boreal climatic zones using the definition from Bonan et al. (2002). The climatic criteria for mapping is presented in Table R2, and the climatic information for mapping was from the climatology of the ERA Interim during 2001-2016 (Berrisford et al., 2011). The final special distribution of the percentages of PFTs is presented in Figure R1. The emission factors in this study are coming from the PFT-level emission factors presented in Table 2 of Guenther et al. (2012). The corresponding description is added at P4, L9 in the revised paper as:

"The PFT was used to determine the canopy structure and standard emission factors in MEGAN (Guenther et al., 2012). We adopted the default emission factors for PFTs described in Table 2 in Guenther et al. (2012). The PFT dataset in this study is obtained from the MODIS MCD12C1 land cover product (https://lpdaac.usgs.gov/products/mcd12c1v006/, Friedl and Sulla-Menashe, 2015). MODIS IGBP classification were mapped to the PFT classification of MEGAN or the Community Land Model (CLM) (Lawrence et al., 2011) based on the description of the legends in the user guide (Sulla-Menashe and Friedl, 2018) and the climatic criteria described in Bonan et al. (2002). The spatial distribution of percentage of PFTs in model grids is presented in Figure 1. According to the description of the legends, we firstly mapped the IGBP classification to eight main vegetation categories: 1) needleleaf evergreen forests, 2) broadleaf evergreen forests, 3) needleleaf deciduous forests, 4) broadleaf deciduous forests, 5) mixed forests, 6) shrub, 7) grass and 8) crop. The mapping method is described in Table S1 in the supplement. Eight main categories then were mapped to the classification of MEGAN/CLM for boreal, temperate and boreal climatic zones using the definition in Bonan et al. (2002). Table S2 in the supplement presents the climatic criteria for mapping, and the climatic information for mapping was from the ERA Interim climatology (https://www.ecmwf.int/en/forecasts/datasets/reanalysis-datasets/era-interim, Berrisford et al., 2011) Reanalysis dataset over 2001-2016."

**Table R1. Look-up table for mapping the IGBP classification scheme to eight main vegetations categories.**

| Name | Value | Description | Percentages of Main Category |
|---|---|---|---|
| Needleleaf Evergreen Forest | 1 | Dominated by evergreen conifer trees (canopy >2m). | 100% Needleleaf Evergreen Tree Forest |
| Broadleaf Evergreen Forest | 2 | Dominated by evergreen broadleaf and palmate trees (canopy >2m). | 100% Broadleaf Evergreen Tree Forest |
| Needleleaf Deciduous Forest | 3 | Dominated by deciduous needleleaf (larch) trees (canopy >2m). | 100% Needleleaf Deciduous Tree Forest |
| Broadleaf Deciduous Forest | 4 | Dominated by deciduous broadleaf trees (canopy >2m). | 100% Broadleaf Deciduous Tree Forest |
| Mixed Forests | 5 | Dominated by neither deciduous nor evergreen (40-60% of each) tree type | 100% Mixed Forests |

| | | (canopy >2m). | |
|---|---|---|---|
| Closed Shrublands | 6 | Dominated by woody perennials (1-2m height) >60% cover. | 100% Shrub |
| Open Shrublands | 7 | Dominated by woody perennials (1-2m height) 10-60% cover. | 60% Shrub
40% Grass |
| Woody Savannas | 8 | Tree cover 30-60% (canopy >2m). | 60% Mixed Forest
20% Shrub
20% Grass |
| Savannas | 9 | Tree cover 10-30% (canopy >2m). | 30% Mixed Forest
35% Shrub
35% Grass |
| Grasslands | 10 | Dominated by herbaceous annuals (<2m). | 100% Grass |
| Permanent Wetlands | 11 | Permanently inundated lands with 30-60% water cover and >10% vegetated cover. | 40% Grass |
| Croplands | 12 | At least 60% of area is cultivated cropland. | 100% Crop |
| Urban and Built-up Lands | 13 | At least 30% impervious surface area including building materials, asphalt, and vehicles. | None |
| Cropland/Natural Vegetation Mosaics | 14 | Mosaics of small-scale cultivation 40-60% with natural tree, shrub, or herbaceous vegetation. | 60% Crop
20% Shrub
20% Grass |
| Permanent Snow and Ice | 15 | At least 60% of area is covered by snow and ice for at least 10 months of the year. | None |
| Barren | 16 | At least 60% of area is non-vegetated barren (sand, rock, soil) areas with less than 10% vegetation. | None |

**Table R2. The climatic criteria for mapping main vegetation categories to CLM PFTs [a].**

| Main Categories | Mapping Condition | Percentages of CLM PFTs |
|---|---|---|
| NET | $T_c$ >-19 °C and GDD > 1200 | 100% NET Temperate |
| | $T_c$≤-19 °C or GDD ≤ 1200 | 100% NET Boreal |
| BET | $T_c$ >15.5 °C | 100% BET Tropical |
| | $T_c$≤15.5 °C | 100% BET Temperate |
| NDT | None | 100% NDT |
| BDT | $T_c$ >15.5 °C | 100% BDT Tropical |
| | -15.5 °C <$T_c$≤15.5 °C or GDD>1200 | 100% BDT Temperate |
| | $T_c$≤-15.5 °C or GDD ≤ 1200 | 100% BDT Boreal |
| Mixed Forest | $T_c$ >15.5 °C | 50% BET Tropical |
| | | 50% BDT Tropical |
| | -15.5 °C<$T_c$≤15.5 °C and GDD>1200 | 33.33% NET Temperate |
| | | 33.33% BET Temperate |
| | | 33.33% BDT Temperate |
| | $T_c$≤-15.5 °C or GDD ≤ 1200 | 33.33% NDT |
| | | 33.33% NET Boreal |
| | | 33.33% BDT Boreal |
| Shrub | $T_c$ >-19 °C and GDD > 1200 | 100% BDS Temperate |
| | $T_c$≤-19 °C or GDD ≤ 1200 | 100% BDS Boreal |
| Grass | GDD<1000 | 100% C3 Arctic |
| | GDD>1000 and (Tc ≤ 22°C or Pmon≤25 mm) | 100% C3 |
| | GDD>1000 and Tc > 22°C and Pmon >25 mm | 100% C4 |
| Crop | None | 100% Crop |

[a] NET, Needleleaf Evergreen Trees; BET, Broadleaf Evergreen Trees; NDT, Needleleaf Evergreen Trees; BDT, Broadleaf Deciduous Trees; Tc, Temperature in the coldest month; GDD, growing-degree days above 5°C; Pmon, monthly precipitation.

[Figure]

**Figure R1. The percentage of different PFTs for the year 2016.**

Then when the authors linked their simulated isoprene emission with the HCHO vertical column, the disagreement of these two has been mainly attributed to the AVOC, but I would think there could be also contributions from the uncertainties in the simulated BVOCs. From the maps with simulated BVOCs, I am a bit surprised to see that the north part of China with high LAI showed very low simulated emissions, especially monoterpene. Could this be linked to the misclassification of forest type?

**Response:** Thank you so much for comments. Firstly, we have added one more section to discuss the uncertainties by comparing our results with the flux measurements and other estimations from previous studies. Secondly, we updated the figure by presenting the annual averaged LAIv instead of growing season LAIv (May-Sep). As shown in the Figure R2, the annual averaged LAIv is not as high as the growing season averaged LAIv in northeast China. In addition, we also mapped the IGBP classification to PFTs with the new rules we designed and the distribution of different PFTs has been given in Figure R1. The main reason why the BVOC emission in northeastern China is low is the impact of local climate in this region. In the revised paper, we added northeastern China as one of the sub-regions for analyzing. As shown in Table 3 in the revised paper, the simulated growing season averaged temperature is about 13.74 °C in northeastern China, which is much lower than other regions, e.g. the simulated growing season averaged temperature is about 20.78 °C in the Qinling mountains. As shown in Figure R1, the tree cover fraction is not low in northeastern China, however, the unfavorable meteorological conditions lead to the low emission in this region.

[Figure]

**Figure R2. Comparison of estimated isoprene annual emission with the satellite derived tropospheric HCHO vertical column concentration by OMI during 2005-2016. (a), (b) and (c) illustrate the spatial distributions of annual mean LAI, isoprene emission and HCHO vertical columns (VC) by OMI respectively. (d) presents the**

**spatial distribution of the correlation coefficient between summertime isoprene emission and HCHO VC. (e) and (f) shows the increasing trend of isoprene and HCHO VC during 2005-2016.**

Then in the east and/or at least North China Plain area, there is wide distribution of crops. Are crops specifically considered in MEGAN?

**Response:** Yeah, as shown in Figure R1, there is wide distribution of crops in North China Plain. The crops are considered as only one kind of PFT in the MEGAN, therefore, emission factors for all species of crops are same in our simulation.

In general, a map showing the spatial distribution of PFTs could be very useful for readers.

**Response:** Thank you so much for your comments. The spatial distribution of different PFTs has been given in Figure R1.

I also think it is crucial to compare the modelled emissions with a few sites' measurement data to illustrate the performance of the model before digging into analysing the changes of the emission patterns at the national scale and further linking to the HCHO column data.

**Response:** Thank you so much for your precious time and your comments. We collected the flux measurements in China from some recent studies (Bai et al., 2015;Bai et al., 2016;Bai et al., 2017) and use them to validate and analyze the uncertainties of our estimation. The details about the flux measurements has been given in Table R3. In addition, we also compared our results with other similar studies to discuss the source of uncertainties in this study. The discussion about the uncertainties in this study has been added at P11, L11 in the revised paper as:

[revised manuscript text omitted]

| | | 28 June -9 July 2010; | | | |
| | | 19 July -30 July 2010; | | | |
| | | 12 Aug.- 25 Aug. 2010; | | | |
| Bai et al. (2015) | Changbai Mountain (42°24′ N, 128°6′) | 19 June - 30 June 2011; | Mixed forest | 4.3 | 0.32 |
| | | 10 July -16 July 2011; | | | |
| | | 22 July - 29 July 2011; | | | |
| | | 5 Sep. - 8 Sep. 2011. | | | |
| Bai et al. (2016) | An Ji, Zhejiang (30°40′15″ N, 119°40′15″) | 7 July-13 July 2012; 20 Aug.-26 Aug. 2012; 25 Sep.-1 Oct. 2012; 28 Oct.- 5 Nov. 2012. | Moso bamboo forest | 3.3 | 0.008 |
| Bai et al. (2017) | Taihe, Jiangxi (26°44′48″ N, 115°04′13″) | 22 May -28 May 2013; 29 June - 6 July 2013; 6 Aug. -13 Aug. 2013; 7 Sep. -11 Sep. 2013; 18 Jan. -19 Jan. 2014; 23 July - 27 July 2014; 2 Nov. - 7 Nov. 2015; 31 Dec. 2015 -4 Jan. 2016. | Subtropical Pinus forest | 0.71 | 1.65 |

Specific comments:

P2 L5-6, please indicate at which spatial scale we can see cropland dominates the reduction of isoprene.

**Response:** Thank you for your comments. We have revised this sentence as:

"For instance, the global cropland expansion has been estimated to dominate the reduction of isoprene, the dominant BVOC species, in last century (Lathière et al., 2010; Unger, 2013) although there are large uncertainties associated with these estimates."

P2 L10, the authors mentioned that the greening in China has been linked to "maintain and expand forests". Did they change plant species when expanding forest? And can you see this level of land use change in the MODIS PFT product?

**Response:** Thank you for your comments. Currently, it is not possible to distinguish the specific species of trees using MODIS since the spatial resolution of MODIS sensor is not high enough to do so. So, we can't see the species-level change through the MODIS PFTs. Our estimation is mainly based on the PFT level change.

P3 L2, suggest to delete "accurately". You have not evaluated the modelled BVOC against the measurements.

**Response:** Thank you so much for your advice, and the word "accurately" has been deleted in the revised paper.

P4L2-4, here you might need to specify where these emission factors are from? How much of these emission factors covered the measurements from China? I did a quick google search and could already see some measurement data available for different ecosystems in China. https://www.sciencedirect.com/science/article/pii/S1352231017302947 https://www.sciencedirect.com/science/article/pii/S1352231015305173 https://www.sciencedirect.com/science/article/pii/S0269749119346081?via%3Dihub

**Response:** Thank you so much for your comments. The emission factors in this study are the default values of the MEGAN 2.1 provided by Guenther et al. 2012. Since we didn't have an ability to distinguish the species of the trees using the MODIS images, we didn't consider using the species-based emission factors. It is true that this will induce the uncertainty of emission amount, and we have added some discussion for this in the revised paper. As mentioned above, we used the flux measurements of BVOC from some recent studies to validate our model and discuss the uncertainties induced by emission factors. The performance of model can be improved by updating the emission factors according to our results. When we adopted the emission factor retrieved from observations (Bai et al., 2015; Bai et al., 2016;Bai et al., 2017), the simulated isoprene and monoterpenes fluxes showed relatively good consistence with the observations by using the same activity factors from this study shown in (c) and (b) in Figure R3. However, these studies only covered very limited numbers of ecosystems in China. Since our work is focusing on the impact of land cover change and vegetation biomass change on BVOC emission, so using the default emission factor is also able to discuss the change of BVOC induced by vegetation development.

[Figure]

**Figure R3. Validation of the model with flux measurements in China. (a) and (b) show the performance of the MEGAN model with the default emission factors. (c) and (d) show the performance of the MEGAN model with the emission factors derived from observations.**

**Response:** As described by Guenther et al. (2006), the $C_{ce}$ is a parameter in MEGAN model that sets the emission factor to unity at the standard conditions. It has no physical meaning and was used to normalize the emission factors.

**Response:** Thank you so much for your comments. The leaf-age factor,$\gamma_{age}$, in MEGAN is described in detail in Guenther et al. (2006). For the evergreen canopies, $\gamma_{age}$ is constant. For the deciduous canopies, the leaves are divided into four stages of new leaf, growing leaf, mature leaf and old leaf since the emission capacity of leaf is diverse with leaf age(Guenther et al., 1991;Monson et al., 1994;Guenther et al., 2006). According to Guenther et al. (2006), the $\gamma_{age}$ is defined as:

$$\gamma_{age} = F_{new}A_{new} + F_{gro}A_{gro} + F_{mat}A_{mat} + F_{old}A_{old}$$

where $A_{new}$, $A_{gro}$, $A_{mat}$ and $A_{old}$ are the relative emission rates for new, growing, mature and old foliages. $F_{new}$, $F_{gro}$, $F_{mat}$ and $F_{old}$ are the fractions of different sorts of leaves and are defined by the change of LAI between the current time step (LAIc) and the previous time step (LAIp). $F_{new}$=0, $F_{gro}$=0.1, $F_{mat}$=0.8 and $F_{old}$=0.1 when LAIc equals LAIp. When LAIp> LAIc, the fractions in different stages are as:

$$\begin{cases} F_{new} = 0 \\ F_{gro} = 0 \\ F_{old} = [(LAIp - LAIc)/LAIp] \\ F_{old} = 1 - F_{old} \end{cases}$$

In the cases of LAIp<LAIc, the fractions are calculated as:

$$\begin{cases} F_{new} = 1 - (LAIp/LAIc) \\ F_{gro} = 1 - F_{new} - F_{mat} \\ F_{mat} = LAIp/LAIc \\ F_{old} = 0 \end{cases}$$

.

P4 L13, Is soil moisture used as inputs for model? If so, please clarify.

**Response:** Thanks for your comments. The soil moisture is simulated by the WRF model and will be considered in the calculation. We have followed your comments and clarify this part in the revised paper in P4, L20:

"The hourly meteorological fields including temperature, downward shortwave radiation (DSW), wind speed, surface pressure, precipitation and water vapor mixing ratio were provided by the Weather Research and Forecast (WRF) Model V3.9 (Skamarock et al., 2008) simulations."

P4 L17-18, LAI is a 'modelled' product from other satellite products and potentially has large uncertainty in itself. I wonder if the LAI has been filtered by the quality flags before using as inputs for MEGAN and how the model deals with the LAI gap if there is no data for many 8-days?

**Response:** Thank you so much for your comments. We used all available values in MODIS LAI products, and we didn't use the quality filter at the first place to ensure the model can be driven by continued LAI field. The model didn't have ability to deal with the LAI gap, but this problem can be solved by using some interpolation technics when preparing the inputs. In this study, we didn't use interpolation method to fill the gaps to avoid introducing artificial uncertainties especially for trend analysis.

P4 L21-23, Could you list what PFTs you have in your simulations (or showing a map), and also how MODIS PFTs were reclassified to the CLM group? I think this information is important for readers to understand the spatial pattern.

**Response:** Thank you so much for your comments. We already provided the method we adopted to reclassify the MODIS IGBP classification to the CLM group in the revised paper. The spatial distribution of different PFTs has been given in Figure R1.

P7 L18-19, the reasons why the simulated MT is so much lower than the previous estimations needs to dig in-depth. Like I mentioned early, could it be linked to the misclassification of PFTs or very different emission factors assigned? In Table 3, the modelled isoprene is very low than Li et al., 2013, can the authors describe a bit about why?

Response: Thank so much for your comments. As mentioned above, we re-mapped the IGBP classification to PFTs with the new rules we designed and the distribution of different PFTs has been given in Figure R1. Currently, our estimation of 33.99 Tg is relatively moderate comparing to other studies (Table R4). In addition, the studies by Li et al. (2013, 2020) showed the highest amounts of isoprene and monoterpenes emissions comparing to other studies. Therefore, in the revised paper, we listed the inputs of different studies to analyze the potential reasons for the discrepancies among these studies. As shown in Table R5, the estimations by Li et al. (2013, 2020) used the species level emission factors and Vegetation Atlas of China for 2007 to describe the spatial distribution of BVOC emission potentials, which is quite different from other studies adopting the PFT-level emission factors and satellite PFT products. They themselves concluded the reason why their estimations were far higher than other studies was because of the high emission factors they adopted. Therefore, the same validations by using canopy-scale BVOC flux measurements are also needed for these studies to validate and constrain the emission factors they used.

**Table R4. Comparison of isoprene and monoterpene emissions (Tg) in China with previous studies.**

| Data Source | Isoprene | Monoterpene | Study period | Method or Model |
|---|---|---|---|---|
| This study | 15.94 (±1.12) | 3.99 (±0.17) | 2001-2016 | MEGAN |
| Stavrakou et al. (2014) | 7.17 (±0.30) | - | 2007-2012 | MEGAN-MOHYCAN |
| Li et al. (2013) | 23.4 | 5.6 | 2003 | MEGAN |
| Li et al. (2020) | 33.21 | 6.35 | 2008-2018 | MEGAN |
| CAMS-GLOB-BIO v1.1 | 7.67 | 3.04 | 2001-2016 | MEGEN |

| | | | | |
|---|---|---|---|---|
| (Sindelarova et al., 2014) | | | | |
| CAMS-GLOB-BIO v3.1 (Sindelarova et al., 2014) | 8.54 | 3.23 | 2001-2016 | MEGAN |
| Fu and Liao (2012) | 10.87 | 3.21 | 2001-2006 | GEOS-Chem-MEGAN |
| Tie et al. (2006) | 7.7 | 3.16 | 2004 | Guenther et al. (1993) |
| Klinger et al. (2002) | 4.65 | 3.97 | 2000 | Guenther et al. (1995) |
| Guenther et al. (1995) | 17 | 4.87 | 1990 | Guenther et al. (1995) |

**Table R5. Comparison of inputs for BVOC estimation with previous studies.**

| Reference | Emission Factor Type | Emission Factor Reference | PFT/Land use | LAI/Biomass | Meteorology | Model/Algorithms |
|---|---|---|---|---|---|---|
| This study | PFT level emission factors | Guenther et al. (2012) | MODIS MCD12C1 v6 | MODIS MCD15A2H v5 | WRF Model v3.9 | MEGANv2.1 |
| Stavrakou et al. (2014) | PFT level emission factors | Guenther et al. (2006) | Ramankutty and Foley (1999) | MODIS MOD15A2 v5 | ERA-Interim Dataset | MEGAN-MOHYCAN |
| Li et al. (2013) | Vegetation genera/species level emission factors | Li et al. (2013) | Vegetation Atlas of China for year 2007 | MEGAN database for 2003 | MM5 Model v3.7 | MEGAN |
| Li et al. (2020) | Vegetation genera/species level emission factors | Li et al. (2013) | Vegetation Atlas of China for year 2007 | Estimations based on surveys and statistics | WRF Model v3.8 | MEGAN |
| CAMS-GLOB-BIO v1.1 (Sindelarova et al., 2014) | PFT level emission factors | Guenther et al. (2012) | 16 plant functional types consistent with the Community Land Model | MODIS MOD15A2 v5 | ERA-Interim Dataset | MEGAN |

| | | | | | | |
|---|---|---|---|---|---|---|
| CAMS-GLOB-BIO v3.1 (Sindelarova et al., 2014) | PFT level emission factors | Guenther et al. (2012) | 16 plant functional types consistent with the Community Land Model | MODIS MOD15A2 v5 | ERA-5 Dataset | MEGAN |
| Fu and Liao (2012) | PFT level emission factors | Guenther et al. (1995) Lathière et al. (2006) Levis et al. (2003) Bai et al. (2006) | MODIS MCD12Q1 v5 | MODIS MOD15A2 v5 | GEOS-4 Meteorology | GEOS-Chem-MEGAN |
| Tie et al. (2006) | Landuse level emission factors | Landuse-based emission rates | USGS 1km land use data | / | WRF model | Guenther et al. (1993) |
| Klinger et al. (2002) | Vegetation genera/species level emission factors | Klinger et al. (2002) | Province-level Forest Inventory | / | Monthly meteorology database by (Leemans and Cramer, 1991) | Guenther et al. (1995) |
| Guenther et al. (1995) | PFT level emission factors | Guenther et al. (1995) | Grided Global Ecosystem Types | Estimations from NPP | Monthly meteorology database by (Leemans and Cramer, 1991) | Guenther et al. (1995) |

P9 L23, might need to add one or two sentences in the method section why p > 0.9 is statistically significant. I did not get it here.

**Response:** Thank you so much for comments. The probability we used here is defined as:

probability = 1 – p,

where p is the 2-sided p value after MK test (https://mailman.ucar.edu/pipermail/ncl-talk/2015-May/002594.html). Since this may confuse the readers, we adopted the original 2-sided p value from MK tests in the revised paper.

P12 L11-12, "The lack of long-term in-situ observations of BVOC in China..." I think this might be the case for most of countries where we don't have dataset being representative at the whole country level,

but I think the authors should definitely compare the modelled with in-situ data for a few representative sites to evaluate the model performance. In China, there are some sites where you can find the ecosystem-level BVOC measurement data for comparison, like some links I provided in the previous comments.

**Response:** Thank you so much for comments. Luckily, some flux measurements were conducted in China and published in recent years. We collected theses flux measurements from some recent studies (Bai et al., 2015;Bai et al., 2016;Bai et al., 2017) and use them to validate and analyze the uncertainties of our estimation. The details about the flux measurements has been given in **Table R3**.

According to our validation, the performance of model can be improved by updating the emission factors. When we adopted the emission factor retrieved from observations (Bai et al., 2015; Bai et al., 2016;Bai et al., 2017), the simulated isoprene and monoterpenes fluxes showed relatively good consistence with the observations by using the same activity factors from this study shown in (c) and (b) in Figure R3. This indicates that emission factors are an important source of uncertainties in this study, on the other hand, it also demonstrates our calculation of activity factor in the model is in a relatively reasonable range. However, these studies only covered very limited numbers of ecosystems in China. Our work is focusing on the impact of land cover change and vegetation biomass change on BVOC emission. The increasing trend of tree cover fraction will increase the BVOC emission with the reasonable activity factors, and the role of emission factors is to decide how strong the trend can be. So, using the default emission factor is also able to discuss the change of BVOC induced by vegetation development.

P12 L12-18, this part should be in the method section.

Response: Thanks for your comments. This part has been introduced in the Section 2.4, so we removed the repeated information here and rephrased this paragraph as:

"The OMI HCHO VC product from 2005-2016 developed by BIRA-IASB (De Smedt et al., 2015) was used in this study. The interannual variability of isoprene emission estimated in this study was evaluated by comparing the isoprene emission with the summer (June-August) averaged HCHO VC."

P13 L5, ". . . are marked with black dots" it is difficult to see these dots though.

Response: Thanks for your comments. As shown in Figure R2, we used relatively sparser and more conspicuous dots to illustrate the grids that passed the t test in the revised paper.

Conclusion, it is rather lengthy at this moment and includes large section of discussion as well. Please make it more concise.

Response: Thanks for your comments. We have removed some lengthy paragraphs in the revised paper,

and we were more focused on discussing the detail of methods and uncertainties.

**Response to Referee #2**

The study by Wang and co-workers investigates the impact of satellite-based land use changes on biogenic VOC emissions in China over 16 years (2001-2016). They report positive emission trends of 1-1.5% per year over the whole country, which are attributed, for a major part, to changes in vegetation. The strongest BVOC trends are reported in Qianling mountains and in south China, where the BVOC emissions increased by more than ~60% in 2016 relative to 2001. Further comparison of BVOC interannual variability with HCHO columns from the OMI instrument over the studied period in summertime exhibited positive temporal correlation over forested regions. This study addresses an interesting subject for Atmospheric Chemistry and Physics journal. However, there are weaknesses and limitations in the present study, which raise doubts regarding the validity of the conclusions. Furthermore, the presentation is often difficult to follow, mostly due to insufficient mastery of the English language. To my view, the manuscript will need a major revision before it becomes suitable for publication. My main concerns are listed below:

**Response:** Thank you so much for your precious time and we really appreciate your comments. We have tried to address your concerns by taking the following measures:

1. We found some mistakes in our program for mapping the MODIS classification system to CLM PFTs, which will lead to missing or double counting some PFT categories during the mapping process. Therefore, we corrected the program and re-ran the all experiments. In addition, we used the IGBP classification scheme this time instead of using the Leaf Area Index Classification Scheme in MCD12C1 product as the original classification scheme for mapping considering the more detailed descriptions of legends in IGBP scheme. Some conclusions were also corrected based on the new results.

2. We added one more experiment named S5 to illustrate the contribution of LAI on trends of BVOC emission. In S5, we used the annually updated LAIv and the fixed meteorological inputs and PFT dataset for the year 2001. The analysis for S5 has been added into the revised paper.

3. We further compared our results with other studies to discuss the uncertainties of our estimation. We downloaded some long-term BVOC estimations from ECCAD database (https://eccad.aeris-data.fr) and compared them with our results to analyses the potential reason that results in the discrepancies between our results and other estimations. In addition, we collected the flux measurements from some recent studies (Bai et al., 2015;Bai et al., 2016;Bai et al., 2017) to validate our model and discuss the uncertainties induced by emission factor. Corresponding content has been added into the revised paper.

4. We removed the Section 3.5 in the previous version paper about "Comparison of BVOC emission with Anthropogenic Emission China". Considering the uncertainties behind our estimations, we decided to concentrate on BVOC emission estimation and discuss more about uncertainties instead of extending to discuss anthropogenic emissions. Some lengthy and less informative paragraphs are also removed in

the revised paper.

(i) Important input datasets required for calculating BVOC emissions using MEGAN model (e.g. PFTs) are not shown. Annual maps of the MODIS PFTs and LAI should be provided, as well as their trends. Without such information, it is impossible to assess the driving factors for the changes and therefore for the validity of the claims. Furthermore, it is not clearly mentioned whether a unique emission factor per PFT has been used (Table 2 of Guenther et al. 2012) or if a map of standard emission factors has been used.

**Response:** Thank you so much for your comments. The spatial distribution of different PFTs has been shown in Figure R1, which is also provided in the revised paper. Besides the spatial distribution of PFTs, the trend of main PFTs and LAIv are also provided here (Figure R4) as well as in the revised paper.

The method for converting MODIS classification system to CLM PFTs is added at P4, L8 as:

"The PFT was used to determine the canopy structure and standard emission factors in MEGAN (Guenther et al., 2012). We adopted the default emission factors for PFTs described in Table 2 in Guenther et al. (2012). The PFT data source in this study is obtained from the MODIS MCD12C1 land cover product (https://lpdaac.usgs.gov/products/mcd12c1v006/, Friedl and Sulla-Menashe, 2015). MODIS IGBP classification were mapped to the PFT classification of MEGAN or the Community Land Model (CLM) (Lawrence et al., 2011) based on the description of the legends in the user guide (Sulla-Menashe and Friedl, 2018) and the climatic criteria described in Bonan et al. (2002). The spatial distribution of percentage of PFTs in model grids is presented in Figure 1. According to the description of the legends, we firstly mapped the IGBP classification to eight main vegetation categories: 1) needleleaf evergreen forests, 2) broadleaf evergreen forests, 3) needleleaf deciduous forests, 4) broadleaf deciduous forests, 5) mixed forests, 6) shrub, 7) grass and 8) crop. The mapping method is described in Table S1 in the supplement. Eight main categories then were mapped to the classification of MEGAN/CLM for boreal, temperate and boreal climatic zones using the definition by Bonan et al. (2002). Table S2 in the supplement presents the climatic criteria for mapping, and the climatic information for mapping was from the climatology of the ERA Interim (https://www.ecmwf.int/en/forecasts/datasets/reanalysis-datasets/era-interim, Berrisford et al., 2011) Reanalysis dataset over 2001-2016."

[Figure]

**Figure R4. Spatial distribution of BVOC emission in 2001 (b) and the changing trends of annual emission rate (S1, S2 and S5), cover fractions of main PFTs and LAIv.**

(ii) I have my doubts regarding the almost negligible isoprene trends due to meteorology suggested by Figure 3 (simulations S3 and S4). The scale in this figure does not allow to see any changes elsewhere than in the Tibetan Plateau. Elsewhere, the color (grey) corresponds to no value. In order to explain the emission trend in S3 and S4, trends of the main drivers of the BVOC emission trends, namely, air temperature, solar radiation and leaf area index should be analysed. In addition, the simulated trend in surface temperature and radiation should be compared to the corresponding trends of the in situ

temperature and solar radiation data used for the evaluation of the WRF model simulation in Section 2.3.

**Response:** Thank you so much for your comments. We have adopted your suggestions and took some measures to improve the way to convey information. As shown in Figure R5, we changed the way we presented the spatial patterns of trends, and we used the black dots to mark the regions with statistically significant trends and keep the non-significant trends for other the regions. For the meteorological drivers, we also gave the trends of growing season 2-meter temperature (T2) and downward shortwave radiation (DSW) (Figure R6). Furthermore, the details of land cover changes, LAI and meteorological conditions were also presented and analyzed for the regional analysis.

We also added the following description in P5, L12:

"The trends of growing season averaged T2 and DSW from model results as well as in-situ measurements are presented in Figure 3. The model and the in-situ measurements show similar patterns for T2. For instance, the model and observations both show an increasing trend in regions like the Tibetan Plateau, southern China and a decreasing trend in eastern and northeastern China. For DSW, the model presented a dimming trend in northeastern and eastern China and a brightening trend in southeastern and central China, and the limited number of radiation observation sites show a similar pattern of trend with model results. In general, the WRF simulation successfully captured the long-term meteorological variabilities and is reasonable to use for estimating the impact of climatic variability on BVOC emission in China for this study."

[Figure]

**Figure R5. The horizontal distributions of isoprene, monoterpenes, sesquiterpenes and total BVOCs emissions of China in 2001 are showed in figure (a), (g), (m) and (s), respectively. The rest columns of figures present the changing trend of isoprene (b-f), monoterpenes (h-l), sesquiterpenes (n-r) and total BVOCs (t-x) in S1, S2, S3, S4 and S5, respectively. The Mann-Kendall test were used to mark the grids where the p is smaller than 0.1.**

[Figure]

**Figure R6. The trend of growing season averaged 2-meter temperature (T2) and downward shortwave radiation (DSW). (a) and (b) are for in-situ T2 and DSW, respectively, and the sites with statistically significant trend are marked by black circles. (c) and (d) are for the WRF simulated T2 and DSW, respectively, and the regions with statistically significant trend are illustrated by shadow.**

(iii) There is not convincing evidence for the very low monoterpene emission derived in this study compared to previous work (Table 3). The invoked reasons, e.g. interannual variations, horizontal resolution, etc. (page 7, lines 81-20) are not convincing. The reasons of the discrepancy should be investigated through detailed comparisons e.g. with the MEGAN inventory and similar studies e.g. Sindelarova et al. (2014). These datasets are accessible via the ECCAD database (https://eccad.aeris-data.fr).

**Response:** Thank you so much for your comments. After re-runing our experiments, our estimation of monoterpenes emission is about 3.99 Tg, which is close to or even higher than other studies. In addition, we have compared our results with the CAMS-GLOB-BIO inventories of BVOC emission from ECCAD database as shown in Figure R7 and Figure R8, and we concluded that the discrepancy between our estimation with CAMS-GLOB-BIO inventories is the PFT and LAI inputs. The meteorological

[revised manuscript text omitted]

(iv) The strong trends inferred over the Qinling mountains and over Southern China need further discussion. Can you put compare this result to past studies? What is the respective roles played by LAI and PFT cover trends?

**Response:** Thank you for your comments. As mentioned above, we compared our results with the CAMS-GLB-BIO inventories. As shown in Figure 5 above, an increasing trend of isoprene emission can be found in CAMS-GLOB-BIO v 1.1 inventory but with relative low magnitude comparing with our estimation. For further discuss the trends of BVOC emission in these regions, we listed the change of annual emission amount in S1, S2 and S5 scenarios, cover fractions of main PFTs, LAIv, growing season temperature and DSW in these regions in Table 3 and Table 4 in the revised paper. In addition, we also added one more experiment named S5 with annually updated LAIv inputs and fixed the meteorological conditions as well as PFT input to investigate the contribution of LAI trend on BVOC emission trend. The results of S5 has been added into the revised paper.

Specific comments/Language corrections

p.2, l.3-7: The sentence is too long, considering splitting into two and rephrasing.

Response: Thank you for your comments. This sentence has been re-written as:

"Besides the climatic factors, the land cover change also plays a key role in the variability of BVOC emission (Stavrakou et al., 2014; Unger, 2013; Chen et al., 2018). For instance, cropland expansion has been estimated to dominate the reduction of isoprene, the dominant BVOC species, in last century (Lathière et al., 2010; Unger, 2013) although there are large uncertainties associated with these estimates."

p.2, l.5: add space between '2014' and 'Chen'

Response: Thank you. We have followed your advice.

p.2, l.12: 'a corresponding impact', replace by 'changes'

Response: Thank you. We have followed your advice.

p.3, l.10: remove 'observed'

Response: Thank you. We have followed your comments.

p.3, l.10: 'regional ecosystem isoprene emission', change to 'isoprene emission at regional to global scales'

Response: Thank you. We have followed your advice.

p.3, l.11: 'reported the', change to 'reported an'

Response: Thank you. We have followed your advice.

p.3, l.12: read 'detected by the Ozone'

Response: Thank you. We have followed your advice.

p.3, l.14-15: rephrases as follows: 'Here we used the long-term OMI 2005-2016 record to estimate the interannual isoprene variability in China'

Response: Thank you. We have followed your advice.

p.3, l.19: add reference Guenther et al.(2012)

Response: Thank you. We have followed your advice.

p.3, l.20: add more references, e.g. Bauwens et al.(2018) and Messina et al.(2016)

Response: Thank you. We have added these references.

p.3, l.23: read 'uses the fundamental'

Response: Thank you. We have followed your advice.

p.4, l.1: read 'the standard emissions factor, and the emission activity factor for the chemical species i'

Response: Thank you. We have followed your advice.

p.4, l.3: '(PFT) distribution from the Community Land...'

Response: Thank you. We have followed your advice.

p.4, l.5: replace 'expresses it as' by 'can be written as'

Response: Thank you. We have followed your advice.

p.4, l.8: 'equal to 1 at standard conditions (Guenther et al. (2006)'

Response: Thank you. We have followed your advice.

p.4, l.9: please specify the source of the LAI dataset

Response: Thank you. We have added the link of website of MODIS LAI products (https://lpdaac.usgs.gov/products/mcd15a2hv006/) in this sentence.

p.4, l.9: replace 'and the leaf age in MEGAN' by a new sentence: 'It is used to define the leaf age response function as described in Guenther et al.(2012).'

Response: Thank you. We have followed your advice.

p.4, l.10: the test should read 'Guenther et al. (1991, 1993, 2012)'

Response: Thank you. We have followed your advice.

Response: Thank you. We have followed your advice.

Response: Thank you so much. We have followed your advice.

p.4, l.18: 'in this study', missing reference for the LAI datasets used

Response: Thank you so much. We have followed your advice and added the reference of MODIS LAI product.

Response: Thank you so much. We have followed your advice and added the reference of MODIS VCF product.

Response: Thank you. We have followed your advice.

Response: Thank you so much. We have followed your advice and added the reference of MODIS land cover product.

Response: Thank you. We have followed your advice.

Response: Thank you. We have followed your advice.

Response: Thank you. We have followed your advice.

p.5, l.5: 'The meteorological simulation is', change to 'The model was'

Response: Thank you. We have followed your advice.

p.5, l.10: 'using the in-situ', change to 'using in-situ'

Response: Thank you. We have followed your advice.

p.5, l.13: 'monthly averaged'

Response: Thank you. We have followed your advice.

p.5, l.15: -2 in Wm-2 should be superscript

Response: Thank you. We have followed your advice.

p.5, l.15: among 98 sites, and the overestimations', change to 'for 98 studied sites. The overestimation'

Response: Thank you. We have followed your advice.

p.5, l.17: 'the lack of aerosol radiation effect and cloud simulation', not clear what is meant here

Response: Thank you so much for your comments. We have modified this sentence as:

"The overestimation of DSW simulation is a common issue in multiple simulation studies and may be induced by the lack of physical processes for aerosol radiation effect (Wang et al., 2011; Situ et al., 2013; Wang et al., 2018)."

p.5, l.23: 'Our', change to 'The'

Response: Thank you. We have followed your advice.

p.6, l.1: 'Observations'

Response: Thank you. We have followed your advice.

p.6, l.3: 'and was retrieved'

Response: Thank you. We have followed your advice.

p.6, l.4-5: 'The detailed...De Smedt et al. (2015)'. Please remove sentence (repetition)

Response: Thank you. We have followed your advice.

p.6, l.6: 'temporally stable', what about the row anomaly? This effect should be mentioned.

Response: Thank you. We have added the following description in the revised paper:

"We used the monthly Level-3 HCHO VC product with 0.25° × 0.25° spatial resolution, and the rows affected by the row anomaly since June 2007 have been filtered in this product (De Smedt et al., 2015; Jin and Holloway, 2015). Since the OMI instrument is temporally stable (Dobber et al., 2008; De Smedt et al., 2015), the OMI HCHO VC product is suitable for long-term analysis (Jin and Holloway, 2015) and was used to primarily validate our estimation of isoprene emission variability."

p.6, l.9: change 'anthropogenic source' to 'anthropogenic VOC'

Response: Thank you. We have followed your advice.

p.6, l.10: 'in the forest regions without obvious anthropogenic impact', replace by 'over forests in summertime'

Response: Thank you. We have followed your advice.

p.6, l.21: 'between 2001 to 2016', change to 'between 2001 and 2016'

Response: Thank you. We have followed your advice.

p.6, l.25: This has been already mentioned, please avoid repetitions

Response: Thank you. We have followed your advice and removed that sentence.

p.6, l.26-27: sentence not clear

Response: Thank you. We have rephrased this sentence as:

"Therefore, the indirect impact of meteorological conditions on BVOC emission through affecting biomass and phenology was not considered in this study."

p.7, l.6-8: what do you mean by 'results' and corresponding results'? State clearly what you did

Response: Thank you. We have followed your advice and rephase this sentence as:

"The trend analysis and the MK tests in this study were implemented using the trend_manken (https://www.ncl.ucar.edu/Document/Functions/Built-in/trend_manken.shtml) function of the NCAR Command Language (NCL, https://www.ncl.ucar.edu/)."

p.7, l.15: 'S1...conditions', repetition

Response: Thank you. We have followed your advice and removed the sentence.

p.7, l.18: 'other estimations', missing references

Response: Thank you. About this part, we have moved the comparison with other studies to section 3.4 as an independent section. The references of studies we used for comparison has been listed in Table 5 as.

"**Table 5. Comparison of isoprene and monoterpene emissions (Tg) in China with previous studies.**

| Data Source | Isoprene | Monoterpene | Study period | Method or Model |
|---|---|---|---|---|
| This study | 15.94 (±1.12) | 3.99 (±0.17) | 2001-2016 | MEGAN |
| Stavrakou et al. (2014) | 7.17 (±0.30) | - | 2007-2012 | MEGAN-MOHYCAN |
| Li et al. (2013) | 23.4 | 5.6 | 2003 | MEGAN |
| Li et al. (2020) | 33.21 | 6.35 | 2008-2018 | MEGAN |
| CAMS-GLOB-BIO v1.1 (Sindelarova et al., 2014) | 7.67 | 3.04 | 2001-2016 | MEGEN |
| CAMS-GLOB-BIO v3.1 (Sindelarova et al., 2014) | 8.54 | 3.23 | 2001-2016 | MEGAN |
| Fu and Liao (2012) | 10.87 | 3.21 | 2001-2006 | GEOS-Chem-MEGAN |
| Tie et al. (2006) | 7.7 | 3.16 | 2004 | Guenther et al. (1993) |
| Klinger et al. (2002) | 4.65 | 3.97 | 2000 | Guenther et al. (1995) |

| | | | | |
|---|---|---|---|---|
| Guenther et al. (1995) | 17 | 4.87 | 1990 | Guenther et al. (1995) |

,,

p.7, l.24: 'increasing rates of these species', replace by 'trends'

Response: Thank you. We have followed your advice.

p.7, l.25: 'despite the direct impact of meteorological conditions', not clear

Response: Thank you so much. We have removed this sentence in the revised paper.

p.8, l.11: Rewrite as 'The average annual total BVOC emission over 2009-2016 is by 50% higher than over 2001-2008.' Is that what you mean?

Response: Thank you so much for your comments. That's what we mean, and we have rewritten this sentence following your suggestion.

p.8, l.13: 'are by 11.3%'

Response: Thank you. We have followed your advice.

p.8, l.21: 'S4 is 23.5%', change to 'S4 is by 23.5%

Response: Thank you. We have followed your advice.

p.8, l.23: 'by 29.9%'

Response: Thank you. We have followed your advice.

p.8, l.25-26: poor language

Response: Thank you. We will rephrase this sentence.

p.9, l.15: 'landcover', change to 'land cover'

Response: Thank you. We have followed your advice.

p.9, l.15: read 'contribute up to 20%, and taken together more than 30% to the estimated...'

Response: Thank you. We have followed your advice.

p.10, l.6: 'driven', change to 'driven'

Response: Thank you. We have followed your advice.

p.10, l.10-12: Sentence could be removed

Response: Thank you. We have followed your advice.

p.10, l.15: superscripts for m-2 y-1

Response: Thank you. We have followed your advice.

p.10, l.20: read 'broadleaf trees, needleleaf trees and other vegetation'

Response: Thank you. We have followed your advice.

p.10, l.25: 'percent', replace by 'percentage'

Response: Thank you. We have followed your advice.

p.10, l.13-24: too many numbers in this paragraph make the reading difficult, consider removing some of the numbers and rewriting

Response: Thank you so much for your suggestion. We will rephrase this paragraph.

p.11, l.4-7: too many numbers in the text, consider introducing them in a table

Response: Thank you. We will consider your advice and add a suitable table or graph.

p.11, l.11: 'in (Figure 3)', change to 'in Figure 3'

Response: Thank you. We have followed your advice.

p.11, l.23: 'dominate factor', read 'dominant factor'

Response: Thank you. We have followed your advice.

p.12, l.2: 'suffering from poor air quality'

Response: Thank you. We have followed your advice.

p.12, l.2: add space between 'years' and 'Yang'

Response: Thank you. We have followed your advice.

p.12, 13: 'in rural regions with minimal anthropogenic influence', change to 'over forests'

Response: Thank you. We have followed your advice.

p.12, l.18: 'summer-average isoprene emission estimated in our study to evaluate our estimation of interannual variability of isoprene emission', poor wording.

Response: Thank you. We have rephrased this sentence as:

"The interannual variability of isoprene emission estimated in this study was evaluated by comparing the isoprene emission with the summer (June-August) averaged HCHO VC."

p.13, l.1: 'anthropogenic sources', missing reference

Response: Thank you. We have added the reference in the revised paper.

p.13, l.5: 'correlation can be found', change to 'correlation is found'

Response: Thank you. We have followed your advice.

p.13, l.10: 'anthropogenic sources', missing reference

Response: Thank you. We have removed this sentence from the revised paper.

p.13, l.20: 'greatest increasing trend', change to 'strongest positive trend'

Response: Thank you. We have followed your advice.

p.14, l.4: 'the mega-city areas', read 'in megacities'

Response: Thank you. We have removed this section.

p.15, l.1: read 'from 2001 to 2016'

Response: Thank you. We have followed your advice.

p.15, l.1: read 'as inputs in the MEGAN'

Response: Thank you. We have followed your advice.

p.15, l.1: 'the long-term', remove 'the'

Response: Thank you. We have followed your advice.

p.15, l.11: here and elsewhere in the manuscript, use one instead of two decimals

Response: Thank you. We have followed your advice.

p.15, l.18: 'there'?

Response: Thank you so much for your comments. We have removed this sentence from the revised paper.

p.15, l.21: 'during 200-2010', missing reference

Response: Thank you so much for your comments. We have removed this sentence from the revised paper.

p.15, l.22: 'there has been in a increasing trend', do you mean 'showed an increasing trend'?

Response: Thank you so much for your comments. We have removed this sentence from the revised paper.

p.15, l.24: read 'assess'

Response: Thank you. We have followed your advice.

p.16, l.6: remove the references (they are already mentioned before)

Response: Thank you. We have followed your advice.

p.16, l.6-10: repetition of l.20-25 of page 14, not necessary

Response: Thank you. We have followed your advice and removed this sentence.

p.23: Table 3, the estimates reported in Li et al. are in TgC, not in Tg, please correct

Response: Thank you so much. We have corrected this in the revised paper.

p.26: Difficult to read, I suggest splitting into a figure with 4 panels (a, f, k, p) and another figure with the trends. The regions in panel (r) are barely visible. Please improve.

Response: Thank you. We added one more figure to illustrate the interest regions and present the trend of BVOC emission.

p.27: It is very difficult to distinguish the colors corresponding to broadleaf and needleleaf trees, please adapt. In the caption, please correct typos for the names of provinces.

Response: Thank you. We have added one figure to present the spatial distribution of broadleaf trees and needle leaf trees. The typos in the caption have been corrected.

[revised manuscript text omitted]

Font color: Text 1

| Page 25: [1] Formatted | Author | 23/10/2020 20:37:00 |

Font color: Text 1

| Page 25: [1] Formatted | Author | 23/10/2020 20:37:00 |

Font color: Text 1

| Page 27: [2] Formatted | Author | 23/10/2020 20:37:00 |

Font color: Text 1

| Page 27: [2] Formatted | Author | 23/10/2020 20:37:00 |

Font color: Text 1

| Page 27: [2] Formatted | Author | 23/10/2020 20:37:00 |

Font color: Text 1

| Page 27: [3] Formatted | Author | 23/10/2020 20:37:00 |

Font color: Text 1

| Page 27: [3] Formatted | Author | 23/10/2020 20:37:00 |

Font color: Text 1

| Page 27: [4] Formatted | Author | 23/10/2020 20:37:00 |

Font color: Text 1

| Page 27: [4] Formatted | Author | 23/10/2020 20:37:00 |

Font color: Text 1

| Page 27: [5] Formatted | Author | 23/10/2020 20:37:00 |

Font color: Text 1

| Page 27: [5] Formatted | Author | 23/10/2020 20:37:00 |

Font color: Text 1

| Page 27: [6] Formatted | Author | 23/10/2020 20:37:00 |

Font color: Text 1

| Page 27: [6] Formatted | Author | 23/10/2020 20:37:00 |

Font color: Text 1

| Page 27: [7] Formatted | Author | 23/10/2020 20:37:00 |

Font color: Text 1

| Page 27: [7] Formatted | Author | 23/10/2020 20:37:00 |

Font color: Text 1

| Page 27: [8] Formatted | Author | 23/10/2020 20:37:00 |

Font color: Text 1

| Page 27: [8] Formatted | Author | 23/10/2020 20:37:00 |

Font color: Text 1

| Page 27: [9] Formatted | Author | 23/10/2020 20:37:00 |

Font color: Text 1

| Page 27: [9] Formatted | Author | 23/10/2020 20:37:00 |

Font color: Text 1

| Page 27: [10] Formatted | Author | 23/10/2020 20:37:00 |

Font color: Text 1

| Page 27: [10] Formatted | Author | 23/10/2020 20:37:00 |

Font color: Text 1

| Page 27: [11] Formatted | Author | 23/10/2020 20:37:00 |

Font color: Text 1

| Page 27: [11] Formatted | Author | 23/10/2020 20:37:00 |

Font color: Text 1

| Page 27: [12] Formatted | Author | 23/10/2020 20:37:00 |

Font color: Text 1

| Page 27: [12] Formatted | Author | 23/10/2020 20:37:00 |

Font color: Text 1

| Page 27: [13] Formatted | Author | 23/10/2020 20:37:00 |

Font color: Text 1

| Page 27: [13] Formatted | Author | 23/10/2020 20:37:00 |

Font color: Text 1

| Page 27: [14] Formatted | Author | 23/10/2020 20:37:00 |

Font color: Text 1

| Page 27: [14] Formatted | Author | 23/10/2020 20:37:00 |

Font color: Text 1

| Page 27: [15] Formatted | Author | 23/10/2020 20:37:00 |

Font color: Text 1

| Page 27: [15] Formatted | Author | 23/10/2020 20:37:00 |

Font color: Text 1

| Page 27: [16] Formatted | Author | 23/10/2020 20:37:00 |

Font color: Text 1

| Page 27: [16] Formatted | Author | 23/10/2020 20:37:00 |

Font color: Text 1

| Page 27: [17] Formatted | Author | 23/10/2020 20:37:00 |

Font color: Text 1

| Page 27: [17] Formatted | Author | 23/10/2020 20:37:00 |

Font color: Text 1

| Page 27: [17] Formatted | Author | 23/10/2020 20:37:00 |

Font color: Text 1

| Page 27: [18] Formatted | Author | 23/10/2020 20:37:00 |

Font color: Text 1

| Page 27: [18] Formatted | Author | 23/10/2020 20:37:00 |

Font color: Text 1

| Page 27: [19] Formatted | Author | 23/10/2020 20:37:00 |

Font color: Text 1

| Page 27: [19] Formatted | Author | 23/10/2020 20:37:00 |

Font color: Text 1

| Page 27: [20] Formatted | Author | 23/10/2020 20:37:00 |

Space After:    0 pt, No widow/orphan control, Keep with next

| Page 27: [21] Formatted | Author | 23/10/2020 20:37:00 |

Font color: Text 1

| Page 27: [21] Formatted | Author | 23/10/2020 20:37:00 |

Font color: Text 1

| Page 27: [21] Formatted | Author | 23/10/2020 20:37:00 |

Font color: Text 1

| Page 27: [21] Formatted | Author | 23/10/2020 20:37:00 |

Font color: Text 1

| Page 28: [22] Formatted | Author | 23/10/2020 20:37:00 |

Caption, Centered, Space After:    10 pt

| Page 28: [23] Formatted | Author | 23/10/2020 20:37:00 |

Font: Times New Roman, 7.5 pt, Bold, Font color: Text 1, English (United Kingdom)

| Page 28: [24] Formatted | Author | 23/10/2020 20:37:00 |

Caption, Centered, Space After:    10 pt

| Page 28: [25] Formatted | Author | 23/10/2020 20:37:00 |

Font: Times New Roman, 7.5 pt, Bold, Font color: Text 1, English (United Kingdom)

| Page 28: [26] Formatted | Author | 23/10/2020 20:37:00 |

Font: Times New Roman, 7.5 pt, Font color: Text 1

| Page 28: [27] Formatted | Author | 23/10/2020 20:37:00 |

Caption, Centered, Space After:    10 pt

| Page 28: [28] Formatted | Author | 23/10/2020 20:37:00 |

Font: Times New Roman, 7.5 pt, Font color: Text 1

| Page 28: [29] Formatted | Author | 23/10/2020 20:37:00 |

Font: Times New Roman, 7.5 pt, Font color: Text 1

| Page 28: [30] Formatted | Author | 23/10/2020 20:37:00 |

Font: Times New Roman, 7.5 pt, Font color: Text 1

| Page 28: [31] Formatted | Author | 23/10/2020 20:37:00 |

Caption, Centered, Space After:    10 pt

| Page 28: [32] Formatted | Author | 23/10/2020 20:37:00 |

Font: Times New Roman, 7.5 pt, Font color: Text 1

| Page 28: [33] Formatted | Author | 23/10/2020 20:37:00 |

Caption, Centered, Space After:    10 pt

| Page 28: [34] Formatted | Author | 23/10/2020 20:37:00 |
|---|---|---|

Font: Times New Roman, 7.5 pt, Bold, Font color: Text 1, English (United Kingdom)

| Page 28: [35] Formatted | Author | 23/10/2020 20:37:00 |
|---|---|---|

Caption, Centered, Space After:    10 pt

| Page 28: [36] Formatted | Author | 23/10/2020 20:37:00 |
|---|---|---|

Font: Times New Roman, 7.5 pt, Bold, Font color: Text 1, English (United Kingdom)

| Page 28: [37] Formatted | Author | 23/10/2020 20:37:00 |
|---|---|---|

Caption, Centered, Space After:    10 pt

| Page 28: [38] Formatted | Author | 23/10/2020 20:37:00 |
|---|---|---|

Font: Times New Roman, 7.5 pt, Bold, Font color: Text 1, English (United Kingdom)

| Page 28: [39] Formatted | Author | 23/10/2020 20:37:00 |
|---|---|---|

Font: Times New Roman, 7.5 pt, Font color: Text 1

| Page 28: [40] Formatted | Author | 23/10/2020 20:37:00 |
|---|---|---|

Caption, Centered, Space After:    10 pt

| Page 28: [41] Formatted | Author | 23/10/2020 20:37:00 |
|---|---|---|

Font: Times New Roman, 7.5 pt, Font color: Text 1

| Page 28: [42] Formatted | Author | 23/10/2020 20:37:00 |
|---|---|---|

Caption, Centered, Space After:    10 pt

| Page 28: [43] Formatted | Author | 23/10/2020 20:37:00 |
|---|---|---|

Font: Times New Roman, 7.5 pt, Bold, Font color: Text 1, English (United Kingdom)

| Page 28: [44] Formatted | Author | 23/10/2020 20:37:00 |
|---|---|---|

Caption, Centered, Space After:    10 pt

| Page 28: [45] Formatted | Author | 23/10/2020 20:37:00 |
|---|---|---|

Font: Times New Roman, 7.5 pt, Font color: Text 1

| Page 28: [46] Formatted | Author | 23/10/2020 20:37:00 |
|---|---|---|

Caption, Centered, Space After:    10 pt

| Page 28: [47] Formatted | Author | 23/10/2020 20:37:00 |
|---|---|---|

Font: Times New Roman, 7.5 pt, Bold, Font color: Text 1, English (United Kingdom)

**Table S1.** Look-up table for mapping the IGBP legend to eight main vegetations categories.

| Name | Value | Description | Main Category Percentage |
|---|---|---|---|
| Needleleaf Evergreen Forest | 1 | Dominated by evergreen conifer trees (canopy >2m). | 100% NET |
| Broadleaf Evergreen Forest | 2 | Dominated by evergreen broadleaf and palmate trees (canopy >2m). | 100% BET |
| Needleleaf Deciduous Forest | 3 | Dominated by deciduous needleleaf (larch) trees (canopy >2m). | 100% NDT |
| Broadleaf Deciduous Forest | 4 | Dominated by deciduous broadleaf trees (canopy >2m). | 100% BDT |
| Mixed Forests | 5 | Dominated by neither deciduous nor evergreen (40-60% of each) tree type (canopy >2m). | 100% Mixed Forests |
| Closed Shrublands | 6 | Dominated by woody perennials (1-2m height) >60% cover. | 100% Shrub |
| Open Shrublands | 7 | Dominated by woody perennials (1-2m height) 10-60% cover. | 60% Shrub 40% Grass |
| Woody Savannas | 8 | Tree cover 30-60% (canopy >2m). | 60% Mixed Forest 20% Shrub 20% Grass |
| Savannas | 9 | Tree cover 10-30% (canopy >2m). | 30% Mixed Forest 35% Shrub 35% Grass |
| Grasslands | 10 | Dominated by herbaceous annuals (<2m). | 100% Grass |
| Permanent Wetlands | 11 | Permanently inundated lands with 30-60% water cover and >10% vegetated cover. | 40% Grass |
| Croplands | 12 | At least 60% of area is cultivated cropland. | 100% Crop |

| | | | |
|---|---|---|---|
| Urban and Built-up Lands | 13 | At least 30% impervious surface area including building materials, asphalt, and vehicles. | None |
| Cropland/Natural Vegetation Mosaics | 14 | Mosaics of small-scale cultivation 40-60% with natural tree, shrub, or herbaceous vegetation. | 60% Crop 20% Shrub 20% Grass |
| Permanent Snow and Ice | 15 | At least 60% of area is covered by snow and ice for at least 10 months of the year. | None |
| Barren | 16 | At least 60% of area is non-vegetated barren (sand, rock, soil) areas with less than 10% vegetation. | None |

**Table 2.** The climatic criteria for mapping main vegetation categories to CLM PFTs.

| Main Category | Mapping Condition | CLM PFT |
|---|---|---|
| NET | $T_c$ >-19 °C and GDD > 1200 | 100% NET Temperate |
| | $T_c$ ≤-19 °C or GDD ≤ 1200 | 100% NET Boreal |
| BET | $T_c$ >15.5 °C | 100% BET Tropical |
| | $T_c$ ≤15.5 °C | 100% BET Temperate |
| NDT | None | 100% NDT |
| BDT | $T_c$ >15.5 °C | 100% BDT Tropical |
| | -15.5 °C <$T_c$≤15.5 °C or GDD>1200 | 100% BDT Temperate |
| | $T_c$ ≤-15.5 °C or GDD ≤ 1200 | 100% BDT Boreal |
| Mixed Forest | $T_c$ >15.5 °C | 50% BET Tropical |
| | | 50% BDT Tropical |
| | -15.5 °C<$T_c$≤15.5 °C and GDD>1200 | 33.33% NET Temperate |
| | | 33.33% BET Temperate |
| | | 33.33% BDT Temperate |
| | $T_c$ ≤-15.5 °C or GDD ≤ 1200 | 33.33% NDT |
| | | 33.33% NET Boreal |
| | | 33.33% BDT Boreal |

| Shrub | $T_c$ >-19 °C and GDD > 1200 | 100% BDS Temperate |
| | $T_c$ ≤-19 °C or GDD ≤ 1200 | 100% BDS Boreal |
| Grass | GDD<1000 | 100% C3 Arctic |
| | GDD>1000 and (Tc ≤ 22°C or Pmon≤25 mm) | 100% C3 |
| | GDD>1000 and Tc > 22°C and Pmon >25 mm | 100% C4 |
| Crop | None | 100% Crop |

**Table S3.** The physical schemes for the WRF simulation.

| Physical mechanism | Scheme |
|---|---|
| Microphysics | WSM 3-class simple ice scheme |
| Long-wave radiation | RRTM scheme |
| Short-wave radiation | Duhbia scheme |
| Land Surface | Noah Land Surface Model |
| PBL Scheme | YSU scheme |
| Cumulus parameter | Kain-Fritsch (new Eta) scheme |

---

## Author Response (AR2)

**Response to Editor**

Thank you for your careful consideration of the previous round of referee comments. The manuscript is substantially improved. However, I agree with the referee reports that further revision is required. Please consider the comments raised by the referee's in this latest round of reports. In particular, please address the comments related to the uncertainty in the trends.

Response: Dear editor, thank you so much for your precious time and great works. In this round, we provided plenty of additional analysis to address the reviewers' concerns. Some analysis is only provided in this response file not in the revised paper since these contents are mostly for addressing the reviewers' questions. In addition, we followed the reviewers' comments and modified the title and the conclusions of the paper, which can better conclude the content of the current manuscript.

We also changed the name of the authors' institution #1 in the paper from "College of Global Change and Earth System Science, Joint Center for Global Changes Studies, Beijing Normal University, Beijing 100875, China" to "College of Global Change and Earth System Science, Beijing Normal University, Beijing 100875, China" because the "Joint Center for Global Changes Studies" has been canceled recently.

**Response to Referee #1**

First, thanks to the authors for the great efforts in revising the manuscript. I saw many places that have been changed, including newly-added tables and uncertainty discussion etc.

Response: Thank you so much for you precious time and constructive comments. We already revised the paper following your comment, and the point-by-point responses have been given as below.

The title of this study is "Land cover change dominates decadal trends of biogenic volatile organic compound (BVOC) emission in China". However, in this new version of figure 4, there is no clear trend in the total emissions (see S1 in Figure 4d), so for me, then the current title is misleading. This also goes to the descriptions in the abstract (Line 20-21). The significant trend found in S2 is with the fixed climate inputs to 2001, but cannot represent the modelled total emissions (i.e., S1).

Response: Thank you so much for your comment. We agreed with your suggestion, and the new title as "A long-term estimation of biogenic volatile organic compound (BVOC) emission in China during 2001-2016: the roles of land cover change and climate variability" according to the current content of manuscript.

Then, my another concern with Figure 4 is that if we remove the first two years (i.e., 2001 and 2002), can we still see the increase trend for S2 and S5? If not, what does that mean? Does that mean the land use management mainly occurs before the year 2003 nationalwide? And are these two years dominating the trend you actually detect for the whole time series? Or can you see the gradual changes from 2003 also contribute to the trend? Response: Thanks for your comment. The reviewer raised a very interesting question, so we analyzed the results without considering the first two years. As shown in Figure R1, after removing the first two years, the scenarios other than S2 didn't show statistically significant trends for most of species as expected by the reviewer, however, the scenario S2 with the fixed climate inputs of 2001 and the annually updated land cover still showed statistically significant increasing trends for all species, which means the change of land cover is not dominated by the first two years. We can further take a look at the horizontal distribution of changing rate. Figure R2 presented the horizontal distributions of BVOC emission trends in different scenarios since 2003. In the revised manuscript, we presented the same figure but with the data starting from 2001. A nationwide significantly increasing trend caused by the vegetation development can be found in scenario S2 (c, i, o and u in Figure R2), and it is very close to the situation we found in the original figure with the data starting from 2001. In addition, the analyses for hot-spot regions in the revised paper also indicate that the first two years are not the dominate factor of increasing trend in these regions. For instance, as shown in Figure R3 (Figure 7 in the revised manuscript), we found that the LAIv as well as the tree cover fraction (broadleaf + needle leaf) in these regions are still in an increasing trend if we ignored the first two years. Therefore, we can't say that this land cover change occurred before 2003, and it is long-term change between 2001 and 2016.

Figure R1. Annual BVOC emissions in China during 2003 to 2016 for five scenarios. The increasing trends and the probabilities (p) using the Mann-Kendall test are shown in the legend.

---

## Author Response (AR3)

**Response to Editor**

**Thank you for your consideration of the referees' comments. I find the manuscript improved and that the referee comments have been largely addressed. I think that after attention to the following points, the manuscript will likely be suitable for publication.**

Response: Dear editor, thank you so much for your precious time and great works. We have revised the manuscript following your comments. The point-to-point response and revised part in the manuscript has been given in the

**1) Please add some additional text (no additional figures are required), that address Referee 1's comment regarding removing the first two years of data and the resulting impact on the trends. The text in the author response document contains important information that should be included in the main text.**

Response: Thank you so much for your comments. We added the corresponding text in the page 7, line 23 in Section 3.1 as:

"We also tested the impact of first two years on the trend estimation of the national scale BVOC emission. After removing the first two years, the scenarios other than S2 didn't show statistically significant trends for most of species, however, the scenario S2 with the fixed climate inputs of 2001 and the annually updated land cover still showed statistically significant increasing trends for all species (p<0.05), which means the change of land cover is not dominated by the first two years."

**2) Please add the information included in response to Referee 1's comments about the main reason for selecting the six regions to the main text. With some minor editing, the text in the author response document is sufficient.**

Response: Thanks for your comments. We have added the information about how we chose 6 regions at page 8, line 32 in the Section 3.2 as:

"Our selection is based on the changing trends we found from Figure 5 and Figure 6. We didn't use the geographical boundaries as the criteria to select the regions of interest, and we chose the hotspots with positive trends and investigated the drivers of trends in these regions."

**3) Please add to the main text a version of discussion in the author response document regarding Referee 2's point about the trends not adding up to the full model simulation trend.**

Response: Thank you so much for your comments. The discussion we provided in the author response document has been added at page 7, line 18 in Section 3.1 in the revised paper as:

"The changing trend of BVOC in the full scenario cannot be treated by the linear summation of trends in other one factor scenarios. On the one hand, the response of isoprene emission to meteorological conditions are nonlinear (Guenther et al. 1993). On the other hand, the calculation of national scale total emission amount is affected by the spatial variabilities of vegetation types as well as climatic conditions, and it should not be a linear combination of the two aspects."

**4) In the abstract line 30 (track changes version) the wording "since the absence of chemical and physical processes" is unclear (perhaps words are missing). Please rephrase.**

Response: Thanks for your comments. This part has been revised as:

"This result may support our estimation of the variability and trends of BVOC emission in this region, however, the comparison still has large uncertainties since the chemical and physical processes, including transportation, diffusion, and chemical reactions, were not considered."

**5) I disagree with the addition of the word "nationwide on page 8 lines 11 and 14 (track changes version) since the trends decrease in some areas. Do you mean that only the statistically significant trends are increasing? The wording needs to be clarified.**

Response: Thank you for your comments. We agreed that the current expression is not very accurate, and it is true that some regions are showing a decreasing trend. Therefore, we summarized the regions with statistically significant positive trends in c, i, o and u of Figure 5 and revised this part at page 8, line 17 as:

"The spatial distributions of trends of different species in S2 all show a significantly increasing trend in the regions including the northeast, central and south of China since the vegetation development is the main driver of the increasing trend of BVOC emission (c, i, o and u in Figure 5)."

**6) Figure 6 caption – add "total" before "BVOC".**

Response: Thanks for your comments. "total" has been added before "BVOC" in the caption of Figure 6.